# Score-based Generative Modeling in Latent Space

**Arash Vahdat**[*]
NVIDIA
avahdat@nvidia.com

**Karsten Kreis**[*]
NVIDIA
kkreis@nvidia.com

**Jan Kautz**
NVIDIA
jkautz@nvidia.com

## Abstract

Score-based generative models (SGMs) have recently demonstrated impressive results in terms of both sample quality and distribution coverage. However, they are usually applied directly in data space and often require thousands of network evaluations for sampling. Here, we propose the *Latent Score-based Generative Model* (LSGM), a novel approach that trains SGMs in a latent space, relying on the variational autoencoder framework. Moving from data to latent space allows us to train more expressive generative models, apply SGMs to non-continuous data, and learn smoother SGMs in a smaller space, resulting in fewer network evaluations and faster sampling. To enable training LSGMs end-to-end in a scalable and stable manner, we (i) introduce a new score-matching objective suitable to the LSGM setting, (ii) propose a novel parameterization of the score function that allows SGM to focus on the mismatch of the target distribution with respect to a simple Normal one, and (iii) analytically derive multiple techniques for variance reduction of the training objective. LSGM obtains a state-of-the-art FID score of 2.10 on CIFAR-10, outperforming all existing generative results on this dataset. On CelebA-HQ-256, LSGM is on a par with previous SGMs in sample quality while outperforming them in sampling time by two orders of magnitude. In modeling binary images, LSGM achieves state-of-the-art likelihood on the binarized OMNIGLOT dataset. Our implementation is available at https://github.com/NVlabs/LSGM.

## 1 Introduction

The long-standing goal of likelihood-based generative learning is to faithfully learn a data distribution, while also generating high-quality samples. Achieving these two goals simultaneously is a tremendous challenge, which has led to the development of a plethora of different generative models. Recently, score-based generative models (SGMs) demonstrated astonishing results in terms of both high sample quality and likelihood [1, 2]. These models define a forward diffusion process that maps data to noise by gradually perturbing the input data. Generation corresponds to a reverse process that synthesizes novel data via iterative denoising, starting from random noise. The problem then reduces to learning *the score function*—the gradient of the log-density—of the perturbed data [3]. In a seminal work, Song et al. [2] show how this modeling approach is described with a stochastic differential equation (SDE) framework which can be converted to maximum likelihood training [4]. Variants of SGMs have been applied to images [1, 2, 5, 6], audio [7, 8, 9, 10], graphs [11] and point clouds [12, 13].

Albeit high quality, sampling from SGMs is computationally expensive. This is because generation amounts to solving a complex SDE, or equivalently ordinary differential equation (ODE) (denoted as the *probability flow ODE* in [2]), that maps a simple base distribution to the complex data distribution. The resulting differential equations are typically complex and solving them accurately requires numerical integration with very small step sizes, which results in thousands of neural network evaluations [1, 2, 6]. Furthermore, generation complexity is uniquely defined by the underlying data distribution and the forward SDE for data perturbation, implying that synthesis speed cannot be

---

[*]Equal contribution.

35th Conference on Neural Information Processing Systems (NeurIPS 2021).

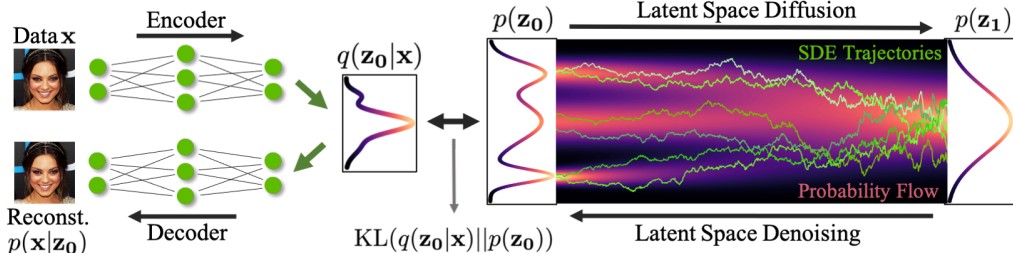

Figure 1: In our latent score-based generative model (LSGM), data is mapped to latent space via an encoder $q(\mathbf{z_0}|\mathbf{x})$ and a diffusion process is applied in the latent space ($\mathbf{z_0} \rightarrow \mathbf{z_1}$). Synthesis starts from the base distribution $p(\mathbf{z_1})$ and generates samples in latent space via denoising ($\mathbf{z_0} \leftarrow \mathbf{z_1}$). Then, the samples are mapped from latent to data space using a decoder $p(\mathbf{x}|\mathbf{z_0})$. The model is trained end-to-end.

increased easily without sacrifices. Moreover, SDE-based generative models are currently defined for continuous data and cannot be applied effortlessly to binary, categorical, or graph-structured data.

Here, we propose the *Latent Score-based Generative Model* (LSGM), a new approach for learning SGMs in latent space, leveraging a variational autoencoder (VAE) framework [14, 15]. We map the input data to latent space and apply the score-based generative model there. The score-based model is then tasked with modeling the distribution over the embeddings of the data set. Novel data synthesis is achieved by first generating embeddings via drawing from a simple base distribution followed by iterative denoising, and then transforming this embedding via a decoder to data space (see Fig. 1). We can consider this model a VAE with an SGM prior. Our approach has several key advantages:

**Synthesis Speed:** By pretraining the VAE with a Normal prior first, we can bring the marginal distribution over encodings (the *aggregate posterior*) close to the Normal prior, which is also the SGM's base distribution. Consequently, the SGM only needs to model the remaining mismatch, resulting in a less complex model from which sampling becomes easier. Furthermore, we can tailor the latent space according to our needs. For example, we can use hierarchical latent variables and apply the diffusion model only over a subset of them, further improving synthesis speed.

**Expressivity:** Training a regular SGM can be considered as training a neural ODE directly on the data [2]. However, previous works found that augmenting neural ODEs [16, 17] and more generally generative models [18, 19, 20, 21] with latent variables improves their expressivity. Consequently, we expect similar performance gains from combining SGMs with a latent variable framework.

**Tailored Encoders and Decoders:** Since we use the SGM in latent space, we can utilize carefully designed encoders and decoders mapping between latent and data space, further improving expressivity. Additionally, the LSGM method can therefore be naturally applied to non-continuous data.

LSGMs can be trained end-to-end by maximizing the variational lower bound on the data likelihood. Compared to regular score matching, our approach comes with additional challenges, since both the score-based denoising model and its target distribution, formed by the latent space encodings, are learnt simultaneously. To this end, we make the following technical contributions: **(i)** We derive a new denoising score matching objective that allows us to efficiently learn the VAE model and the latent SGM prior at the same time. **(ii)** We introduce a new parameterization of the latent space score function, which mixes a Normal distribution with a learnable SGM, allowing the SGM to model only the mismatch between the distribution of latent variables and the Normal prior. **(iii)** We propose techniques for variance reduction of the training objective by designing a new SDE and by analytically deriving importance sampling schemes, allowing us to stably train deep LSGMs. Experimentally, we achieve state-of-the-art 2.10 FID on CIFAR-10 and 7.22 FID on CelebA-HQ-256, and significantly improve upon likelihoods of previous SGMs. On CelebA-HQ-256, we outperform previous SGMs in synthesis speed by two orders of magnitude. We also model binarized images, MNIST and OMNIGLOT, achieving state-of-the-art likelihood on the latter.

## 2 Background

Here, we review continuous-time score-based generative models (see [2] for an in-depth discussion). Consider a forward diffusion process $\{\mathbf{z}_t\}_{t=0}^{t=1}$ for continuous time variable $t \in [0, 1]$, where $\mathbf{z}_0$ is the starting variable and $\mathbf{z}_t$ its perturbation at time $t$. The diffusion process is defined by an Itô SDE:

$$d\mathbf{z} = f(t)\mathbf{z}\, dt + g(t)\, d\mathbf{w} \tag{1}$$

where $f : \mathbb{R} \rightarrow \mathbb{R}$ and $g : \mathbb{R} \rightarrow \mathbb{R}$ are scalar drift and diffusion coefficients, respectively, and $\mathbf{w}$ is the standard Wiener process. $f(t)$ and $g(t)$ can be designed such that $\mathbf{z}_1 \sim \mathcal{N}(\mathbf{z}_1; \mathbf{0}, \mathbf{I})$ follows a

Normal distribution at the end of the diffusion process.[2] Song et al. [2] show that the SDE in Eq. 1 can be converted to a generative model by first sampling from $\mathbf{z}_1 \sim \mathcal{N}(\mathbf{z}_1; \mathbf{0}, \mathbf{I})$ and then running the reverse-time SDE $d\mathbf{z} = [f(t)\mathbf{z} - g(t)^2 \nabla_\mathbf{z} \log q_t(\mathbf{z})] dt + g(t) d\bar{\mathbf{w}}$, where $\bar{\mathbf{w}}$ is a reverse-time standard Wiener process and $dt$ is an infinitesimal negative time step. The reverse SDE requires knowledge of $\nabla_{\mathbf{z}_t} \log q_t(\mathbf{z}_t)$, the score function of the marginal distribution under the forward diffusion at time $t$. One approach for estimating it is via the score matching objective[3]:

$$\min_{\boldsymbol{\theta}} \mathbb{E}_{t \sim \mathcal{U}[0,1]} \left[ \lambda(t) \mathbb{E}_{q(\mathbf{z}_0)} \mathbb{E}_{q(\mathbf{z}_t|\mathbf{z}_0)} [||\nabla_{\mathbf{z}_t} \log q(\mathbf{z}_t) - \nabla_{\mathbf{z}_t} \log p_{\boldsymbol{\theta}}(\mathbf{z}_t)||_2^2] \right] \tag{2}$$

that trains the parameteric score function $\nabla_{\mathbf{z}_t} \log p_{\boldsymbol{\theta}}(\mathbf{z}_t)$ at time $t \sim \mathcal{U}[0,1]$ for a given weighting coefficient $\lambda(t)$. $q(\mathbf{z}_0)$ is the $\mathbf{z}_0$-generating distribution and $q(\mathbf{z}_t|\mathbf{z}_0)$ is the diffusion kernel, which is available in closed form for certain $f(t)$ and $g(t)$. Since $\nabla_{\mathbf{z}_t} \log q(\mathbf{z}_t)$ is not analytically available, Song et al. [2] rely on denoising score matching [22] that converts the objective in Eq. 2 to:

$$\min_{\boldsymbol{\theta}} \mathbb{E}_{t \sim \mathcal{U}[0,1]} \left[ \lambda(t) \mathbb{E}_{q(\mathbf{z}_0)} \mathbb{E}_{q(\mathbf{z}_t|\mathbf{z}_0)} [||\nabla_{\mathbf{z}_t} \log q(\mathbf{z}_t|\mathbf{z}_0) - \nabla_{\mathbf{z}_t} \log p_{\boldsymbol{\theta}}(\mathbf{z}_t)||_2^2] \right] + C \tag{3}$$

Vincent [22] shows $C = \mathbb{E}_{t \sim \mathcal{U}[0,1]}[\lambda(t) \mathbb{E}_{q(\mathbf{z}_0)} \mathbb{E}_{q(\mathbf{z}_t|\mathbf{z}_0)} [||\nabla_{\mathbf{z}_t} \log q(\mathbf{z}_t)||_2^2 - ||\nabla_{\mathbf{z}_t} \log q(\mathbf{z}_t|\mathbf{z}_0)||_2^2]]$ is independent of $\boldsymbol{\theta}$, making the minimizations in Eq. 3 and Eq. 2 equivalent. Song et al. [4] show that for $\lambda(t) = g(t)^2/2$, the minimizations correspond to approximate maximum likelihood training based on an upper on the Kullback-Leibler (KL) divergence between the target distribution and the distribution defined by the reverse-time generative SDE with the learnt score function. In particular, the objective of Eq. 2 can then be written:

$$\mathrm{KL}\big(q(\mathbf{z}_0)||p_{\boldsymbol{\theta}}(\mathbf{z}_0)\big) \leq \mathbb{E}_{t \sim \mathcal{U}[0,1]} \left[ \frac{g(t)^2}{2} \mathbb{E}_{q(\mathbf{z}_0)} \mathbb{E}_{q(\mathbf{z}_t|\mathbf{z}_0)} \left[ ||\nabla_{\mathbf{z}_t} \log q(\mathbf{z}_t) - \nabla_{\mathbf{z}_t} \log p_{\boldsymbol{\theta}}(\mathbf{z}_t)||_2^2 \right] \right] \tag{4}$$

which can again be transformed into denoising score matching (Eq. 3) following Vincent [22].

## 3  Score-based Generative Modeling in Latent Space

The LSGM framework in Fig. 1 consists of the encoder $q_{\boldsymbol{\phi}}(\mathbf{z}_0|\mathbf{x})$, SGM prior $p_{\boldsymbol{\theta}}(\mathbf{z}_0)$, and decoder $p_{\boldsymbol{\psi}}(\mathbf{x}|\mathbf{z}_0)$. The SGM prior leverages a diffusion process as defined in Eq. 1 and diffuses $\mathbf{z}_0 \sim q_{\boldsymbol{\phi}}(\mathbf{z}_0|\mathbf{x})$ samples in latent space to the standard Normal distribution $p(\mathbf{z}_1) = \mathcal{N}(\mathbf{z}_1; \mathbf{0}, \mathbf{I})$. Generation uses the reverse SDE to sample from $p_{\boldsymbol{\theta}}(\mathbf{z}_0)$ with time-dependent score function $\nabla_{\mathbf{z}_t} \log p_{\boldsymbol{\theta}}(\mathbf{z}_t)$, and the decoder $p_{\boldsymbol{\psi}}(\mathbf{x}|\mathbf{z}_0)$ to map the synthesized encodings $\mathbf{z}_0$ to data space. Formally, the generative process is written as $p(\mathbf{z}_0, \mathbf{x}) = p_{\boldsymbol{\theta}}(\mathbf{z}_0) p_{\boldsymbol{\psi}}(\mathbf{x}|\mathbf{z}_0)$. The goal of training is to learn $\{\boldsymbol{\phi}, \boldsymbol{\theta}, \boldsymbol{\psi}\}$, the parameters of the encoder $q_{\boldsymbol{\phi}}(\mathbf{z}_0|\mathbf{x})$, score function $\nabla_{\mathbf{z}_t} \log p_{\boldsymbol{\theta}}(\mathbf{z}_t)$, and decoder $p_{\boldsymbol{\psi}}(\mathbf{x}|\mathbf{z}_0)$, respectively.

We train LSGM by minimizing the variational upper bound on negative data log-likelihood $\log p(\mathbf{x})$:

$$\mathcal{L}(\mathbf{x}, \boldsymbol{\phi}, \boldsymbol{\theta}, \boldsymbol{\psi}) = \mathbb{E}_{q_{\boldsymbol{\phi}}(\mathbf{z}_0|\mathbf{x})} \big[ -\log p_{\boldsymbol{\psi}}(\mathbf{x}|\mathbf{z}_0) \big] + \mathrm{KL}\big(q_{\boldsymbol{\phi}}(\mathbf{z}_0|\mathbf{x})||p_{\boldsymbol{\theta}}(\mathbf{z}_0)\big) \tag{5}$$

$$= \underbrace{\mathbb{E}_{q_{\boldsymbol{\phi}}(\mathbf{z}_0|\mathbf{x})} \big[ -\log p_{\boldsymbol{\psi}}(\mathbf{x}|\mathbf{z}_0) \big]}_{\text{reconstruction term}} + \underbrace{\mathbb{E}_{q_{\boldsymbol{\phi}}(\mathbf{z}_0|\mathbf{x})} \big[ \log q_{\boldsymbol{\phi}}(\mathbf{z}_0|\mathbf{x}) \big]}_{\text{negative encoder entropy}} + \underbrace{\mathbb{E}_{q_{\boldsymbol{\phi}}(\mathbf{z}_0|\mathbf{x})} \big[ -\log p_{\boldsymbol{\theta}}(\mathbf{z}_0) \big]}_{\text{cross entropy}} \tag{6}$$

following a VAE approach [14, 15], where $q_{\boldsymbol{\phi}}(\mathbf{z}_0|\mathbf{x})$ approximates the true posterior $p(\mathbf{z}_0|\mathbf{x})$.

In this paper, we use Eq. 6 with decomposed KL divergence into its entropy and cross entropy terms. The reconstruction and entropy terms are estimated easily for any explicit encoder as long as the reparameterization trick is available [14]. The challenging part in training LSGM is to train the cross entropy term that involves the SGM prior. We motivate and present our expression for the cross-entropy term in Sec. 3.1, the parameterization of the SGM prior in Sec. 3.2, different weighting mechanisms for the training objective in Sec. 3.3, and variance reduction techniques in Sec. 3.4.

### 3.1  The Cross Entropy Term

One may ask, why not train LSGM with Eq. 5 and rely on the KL in Eq. 4. Directly using the KL expression in Eq. 4 is not possible, as it involves the marginal score $\nabla_{\mathbf{z}_t} \log q(\mathbf{z}_t)$, which is unavailable analytically for common non-Normal distributions $q(\mathbf{z}_0)$ such as Normalizing flows.

---

[2]Other distributions at $t = 1$ are possible; for instance, see the "variance-exploding" SDE in [2]. In this paper, however, we use only SDEs converging towards $\mathcal{N}(\mathbf{z}_1; \mathbf{0}, \mathbf{I})$ at $t = 1$.

[3]We omit the $t$-subscript of the diffused distributions $q_t$ in all score functions of the form $\nabla_{\mathbf{z}_t} \log q_t(\mathbf{z}_t)$.

Transforming into denoising score matching does not help either, since in that case the problematic $\nabla_{\mathbf{z}_t} \log q(\mathbf{z}_t)$ term appears in the $C$ term (see Eq. 3). In contrast to previous works [2, 22], we cannot simply drop $C$, since it is, in fact, not constant but depends on $q(\mathbf{z}_t)$, which is trainable in our setup.

To circumvent this problem, we instead decompose the KL in Eq. 5 and rather work directly with the cross entropy between the encoder distribution $q(\mathbf{z}_0|\mathbf{x})$ and the SGM prior $p(\mathbf{z}_0)$. We show:

**Theorem 1.** *Given two distributions $q(\mathbf{z}_0|\mathbf{x})$ and $p(\mathbf{z}_0)$, defined in the continuous space $\mathbb{R}^D$, denote the marginal distributions of diffused samples under the SDE in Eq. 1 at time $t$ with $q(\mathbf{z}_t|\mathbf{x})$ and $p(\mathbf{z}_t)$. Assuming mild smoothness conditions on $\log q(\mathbf{z}_t|\mathbf{x})$ and $\log p(\mathbf{z}_t)$, the cross entropy is:*

$$CE(q(\mathbf{z}_0|\mathbf{x})||p(\mathbf{z}_0)) = \mathbb{E}_{t\sim\mathcal{U}[0,1]}\left[\frac{g(t)^2}{2}\mathbb{E}_{q(\mathbf{z}_t,\mathbf{z}_0|\mathbf{x})}\left[||\nabla_{\mathbf{z}_t}\log q(\mathbf{z}_t|\mathbf{z}_0)-\nabla_{\mathbf{z}_t}\log p(\mathbf{z}_t)||_2^2\right]\right]+\frac{D}{2}\log\left(2\pi e\sigma_0^2\right),$$

*with $q(\mathbf{z}_t,\mathbf{z}_0|\mathbf{x}) = q(\mathbf{z}_t|\mathbf{z}_0)q(\mathbf{z}_0|\mathbf{x})$ and a Normal transition kernel $q(\mathbf{z}_t|\mathbf{z}_0) = \mathcal{N}(\mathbf{z}_t; \boldsymbol{\mu}_t(\mathbf{z}_0), \sigma_t^2\mathbf{I})$, where $\boldsymbol{\mu}_t$ and $\sigma_t^2$ are obtained from $f(t)$ and $g(t)$ for a fixed initial variance $\sigma_0^2$ at $t=0$.*

A proof with generic expressions for $\boldsymbol{\mu}_t$ and $\sigma_t^2$ as well as an intuitive interpretation are in App. A.

Importantly, unlike for the KL objective of Eq. 4, no problematic terms depending on the marginal score $\nabla_{\mathbf{z}_t} \log q(\mathbf{z}_t|\mathbf{x})$ arise. This allows us to use this denoising score matching objective for the cross entropy term in Theorem 1 not only for optimizing $p(\mathbf{z}_0)$ (which is commonly done in the score matching literature), but also for the $q(\mathbf{z}_0|\mathbf{x})$ encoding distribution. It can be used even with complex $q(\mathbf{z}_0|\mathbf{x})$ distributions, defined, for example, in a hierarchical fashion [20, 21] or via Normalizing flows [23, 24]. Our novel analysis shows that, for diffusion SDEs following Eq. 1, only the cross entropy can be expressed purely with $\nabla_{\mathbf{z}_t} \log q(\mathbf{z}_t|\mathbf{z}_0)$. Neither KL nor entropy in [4] can be expressed without the problematic term $\nabla_{\mathbf{z}_t} \log q(\mathbf{z}_t|\mathbf{x})$ (details in the Appendix).

Note that in Theorem 1, the term $\nabla_{\mathbf{z}_t} \log p(\mathbf{z}_t)$ in the score matching expression corresponds to the score that originates from diffusing an initial $p(\mathbf{z}_0)$ distribution. In practice, we use the expression to learn an SGM prior $p_{\boldsymbol{\theta}}(\mathbf{z}_0)$, which models $\nabla_{\mathbf{z}_t} \log p(\mathbf{z}_t)$ by a neural network. With the learnt score $\nabla_{\mathbf{z}_t} \log p_{\boldsymbol{\theta}}(\mathbf{z}_t)$ (here we explicitly indicate the parameters $\boldsymbol{\theta}$ to clarify that this is the learnt model), the actual SGM prior is defined via the generative reverse-time SDE (or, alternatively, a closely-connected ODE, see Sec. 2 and App. D), which generally defines its own, separate marginal distribution $p_{\boldsymbol{\theta}}(\mathbf{z}_0)$ at $t=0$. Importantly, the learnt, approximate score $\nabla_{\mathbf{z}_t} \log p_{\boldsymbol{\theta}}(\mathbf{z}_t)$ is not necessarily the same as one would obtain when diffusing $p_{\boldsymbol{\theta}}(\mathbf{z}_0)$. Hence, when considering the learnt score $\nabla_{\mathbf{z}_t} \log p_{\boldsymbol{\theta}}(\mathbf{z}_t)$, the score matching expression in our Theorem only corresponds to an upper bound on the cross entropy between $q(\mathbf{z}_0|\mathbf{x})$ and $p_{\boldsymbol{\theta}}(\mathbf{z}_0)$ defined by the generative reverse-time SDE. This is discussed in detail in concurrent works [4, 25]. Hence, from the perspective of the learnt SGM prior, we are training with an upper bound on the cross entropy (similar to the bound on the KL in Eq. 4), which can also be considered as the continuous version of the discretized variational objective derived by Ho et al. [1].

### 3.2 Mixing Normal and Neural Score Functions

In VAEs [14], $p(\mathbf{z}_0)$ is often chosen as a standard Normal $\mathcal{N}(\mathbf{z}_0; \mathbf{0}, \mathbf{I})$. For recent hierarchical VAEs [20, 21], using the reparameterization trick, the prior can be converted to $\mathcal{N}(\mathbf{z}_0; \mathbf{0}, \mathbf{I})$ (App. E).

Considering a single dimensional latent space, we can assume that the prior at time $t$ is in the form of a geometric mixture $p(z_t) \propto \mathcal{N}(z_t; 0, 1)^{1-\alpha} p'_{\boldsymbol{\theta}}(z_t)^{\alpha}$ where $p'_{\boldsymbol{\theta}}(z_t)$ is a trainable SGM prior and $\alpha \in [0, 1]$ is a learnable scalar mixing coefficient. Formulating the prior this way has crucial advantages: (i) We can pretrain LSGM's autoencoder networks assuming $\alpha=0$, which corresponds to training the VAE with a standard Normal prior. This pretraining step will bring the distribution of latent variable close to $\mathcal{N}(z_0; 0, 1)$, allowing the SGM prior to learn a much simpler distribution in the following end-to-end training stage. (ii) The score function for this mixture is of the form $\nabla_{z_t} \log p(z_t) = -(1-\alpha)z_t + \alpha\nabla_{z_t} \log p'_{\boldsymbol{\theta}}(z_t)$. When the score function is dominated by the linear term, we expect that the reverse SDE can be solved faster, as its drift is dominated by this linear term.

For our multivariate latent space, we obtain diffused samples at time $t$ by sampling $\mathbf{z}_t \sim q(\mathbf{z}_t|\mathbf{z}_0)$ with $\mathbf{z}_t = \boldsymbol{\mu}_t(\mathbf{z}_0) + \sigma_t\boldsymbol{\epsilon}$, where $\boldsymbol{\epsilon} \sim \mathcal{N}(\boldsymbol{\epsilon}; \mathbf{0}, \mathbf{I})$. Since we have $\nabla_{\mathbf{z}_t} \log q(\mathbf{z}_t|\mathbf{z}_0) = -\boldsymbol{\epsilon}/\sigma_t$, similar to [1], we parameterize the score function by $\nabla_{\mathbf{z}_t} \log p(\mathbf{z}_t) := -\boldsymbol{\epsilon}_{\theta}(\mathbf{z}_t, t)/\sigma_t$, where $\boldsymbol{\epsilon}_{\theta}(\mathbf{z}_t, t) := \sigma_t(1-\boldsymbol{\alpha}) \odot \mathbf{z}_t + \boldsymbol{\alpha} \odot \boldsymbol{\epsilon}'_{\theta}(\mathbf{z}_t, t)$ is defined by our *mixed score parameterization* that is applied elementwise to the components of the score. With this, we simplify the cross entropy expression to:

$$CE(q_{\boldsymbol{\phi}}(\mathbf{z}_0|\mathbf{x})||p_{\boldsymbol{\theta}}(\mathbf{z}_0)) = \mathbb{E}_{t\sim\mathcal{U}[0,1]}\left[\frac{w(t)}{2}\mathbb{E}_{q_{\boldsymbol{\phi}}(\mathbf{z}_t,\mathbf{z}_0|\mathbf{x}),\boldsymbol{\epsilon}}\left[||\boldsymbol{\epsilon}-\boldsymbol{\epsilon}_{\boldsymbol{\theta}}(\mathbf{z}_t, t)||_2^2\right]\right]+\frac{D}{2}\log\left(2\pi e\sigma_0^2\right), \quad (7)$$

where $w(t) = g(t)^2/\sigma_t^2$ is a time-dependent weighting scalar.

### 3.3 Training with Different Weighting Mechanisms

The weighting term $w(t)$ in Eq. 7 trains the prior with maximum likelihood. Similar to [1, 2], we observe that when $w(t)$ is dropped while training the SGM prior (i.e., $w(t) = 1$), LSGM often yields higher quality samples at a small cost in likelihood. However, in our case, we can only drop the weighting when training the prior. When

Table 1: Weighting mechanisms

| Mechanism | Weights |
|-----------|---------|
| Weighted | $w_{\text{ll}}(t) = g(t)^2/\sigma_t^2$ |
| Unweighted | $w_{\text{un}}(t) = 1$ |
| Reweighted | $w_{\text{re}}(t) = g(t)^2$ |

updating the encoder parameters, we still need to use the maximum likelihood weighting to ensure that the encoder $q(\mathbf{z}_0|\mathbf{x})$ is brought closer to the true posterior $p(\mathbf{z}_0|\mathbf{x})$[4]. Tab. 1 summarizes three weighting mechanisms we consider in this paper: $w_{\text{ll}}(t)$ corresponds to maximum likelihood, $w_{\text{un}}(t)$ is the unweighted objective used by [1, 2], and $w_{\text{re}}(t)$ is a variant obtained by dropping only $1/\sigma_t^2$. This weighting mechanism has a similar affect on the sample quality as $w_{\text{un}}(t) = 1$; however, in Sec. 3.4, we show that it is easier to define a variance reduction scheme for this weighting mechanism.

The following summarizes our training objectives (with $t \sim \mathcal{U}[0, 1]$ and $\boldsymbol{\epsilon} \sim \mathcal{N}(\boldsymbol{\epsilon}; \mathbf{0}, \mathbf{I})$):

$$\min_{\boldsymbol{\phi}, \boldsymbol{\psi}} \mathbb{E}_{q_{\boldsymbol{\phi}}(\mathbf{z}_0|\mathbf{x})} \left[ -\log p_{\boldsymbol{\psi}}(\mathbf{x}|\mathbf{z}_0) \right] + \mathbb{E}_{q_{\boldsymbol{\phi}}(\mathbf{z}_0|\mathbf{x})} \left[ \log q_{\boldsymbol{\phi}}(\mathbf{z}_0|\mathbf{x}) \right] + \mathbb{E}_{t, \boldsymbol{\epsilon}, q(\mathbf{z}_t|\mathbf{z}_0), q_{\boldsymbol{\phi}}(\mathbf{z}_0|\mathbf{x})} \left[ \frac{w_{\text{ll}}(t)}{2} ||\boldsymbol{\epsilon} - \boldsymbol{\epsilon}_{\boldsymbol{\theta}}(\mathbf{z}_t, t)||_2^2 \right] \quad (8)$$

$$\min_{\boldsymbol{\theta}} \mathbb{E}_{t, \boldsymbol{\epsilon}, q(\mathbf{z}_t|\mathbf{z}_0), q_{\boldsymbol{\phi}}(\mathbf{z}_0|\mathbf{x})} \left[ \frac{w_{\text{ll/un/re}}(t)}{2} ||\boldsymbol{\epsilon} - \boldsymbol{\epsilon}_{\boldsymbol{\theta}}(\mathbf{z}_t, t)||_2^2 \right] \quad \text{with} \quad q(\mathbf{z}_t|\mathbf{z}_0) = \mathcal{N}(\mathbf{z}_t; \boldsymbol{\mu}_t(\mathbf{z}_0), \sigma_t^2 \mathbf{I}), \quad (9)$$

where Eq. 8 trains the VAE encoder and decoder parameters $\{\boldsymbol{\phi}, \boldsymbol{\psi}\}$ using the variational bound $\mathcal{L}(\mathbf{x}, \boldsymbol{\phi}, \boldsymbol{\theta}, \boldsymbol{\psi})$ from Eq. 6. Eq. 9 trains the prior with one of the three weighting mechanisms. Since the SGM prior participates in the objective only in the cross entropy term, we only consider this term when training the prior. Efficient algorithms for training with the objectives are presented in App. G.

### 3.4 Variance Reduction

The objectives in Eqs. 8 and 9 involve sampling of the time variable $t$, which has high variance [26]. We introduce several techniques for reducing this variance for all three objective weightings. We focus on the "variance preserving" SDEs (VPSDEs) [2, 1, 27], defined by $d\mathbf{z} = -\frac{1}{2}\beta(t)\mathbf{z}\,dt + \sqrt{\beta(t)}\,d\mathbf{w}$ where $\beta(t) = \beta_0 + (\beta_1 - \beta_0)t$ linearly interpolates in $[\beta_0, \beta_1]$ (other SDEs discussed in App. B).

We denote the marginal distribution of latent variables by $q(\mathbf{z}_0) := \mathbb{E}_{p_{\text{data}}(\mathbf{x})}[q(\mathbf{z}_0|\mathbf{x})]$. Here, we derive variance reduction techniques for $\text{CE}(q(\mathbf{z}_0)||p(\mathbf{z}_0))$, assuming that both $q(\mathbf{z}_0) = p(\mathbf{z}_0) = \mathcal{N}(\mathbf{z}_0; \mathbf{0}, \mathbf{I})$. This is a reasonable simplification for our analysis because pretraining our LSGM model with a $\mathcal{N}(\mathbf{z}_0; \mathbf{0}, \mathbf{I})$ prior brings $q(\mathbf{z}_0)$ close to $\mathcal{N}(\mathbf{z}_0; \mathbf{0}, \mathbf{I})$ and our SGM prior is often dominated by the fixed Normal mixture component. We empirically observe that the variance reduction techniques developed with this assumption still work well when $q(\mathbf{z}_0)$ and $p(\mathbf{z}_0)$ are not exactly $\mathcal{N}(\mathbf{z}_0; \mathbf{0}, \mathbf{I})$.

**Variance reduction for likelihood weighting:** In App. B, for $q(\mathbf{z}_0) = p(\mathbf{z}_0) = \mathcal{N}(\mathbf{z}_0; \mathbf{0}, \mathbf{I})$, we show $\text{CE}(q(\mathbf{z}_0)||p(\mathbf{z}_0))$ is given by $\frac{D}{2}\mathbb{E}_{t \sim \mathcal{U}[0,1]}[\mathrm{d}\log\sigma_t^2/\mathrm{d}t] + \text{const}$. We consider two approaches:

**(1)** *Geometric VPSDE*: To reduce the variance sampling uniformly from $t$, we can design the SDE such that $\mathrm{d}\log\sigma_t^2/\mathrm{d}t$ is constant for $t \in [0, 1]$. We show in App. B that a $\beta(t) = \log(\sigma_{\max}^2/\sigma_{\min}^2)\frac{\sigma_t^2}{(1-\sigma_t^2)}$ with geometric variance $\sigma_t^2 = \sigma_{\min}^2(\sigma_{\max}^2/\sigma_{\min}^2)^t$ satisfies this condition. We call a VPSDE with this $\beta(t)$ a geometric VPSDE. $\sigma_{\min}^2$ and $\sigma_{\max}^2$ are the hyperparameters of the SDE, with $0 < \sigma_{\min}^2 < \sigma_{\max}^2 < 1$. Although our geometric VPSDE has a geometric variance progression similar to the "variance exploding" SDE (VESDE) [2], it still enjoys the "variance preserving" property of the VPSDE. In App. B, we show that the VESDE does not come with a reduced variance for $t$-sampling by default.

**(2)** *Importance sampling (IS):* We can keep $\beta(t)$ and $\sigma_t^2$ unchanged for the original linear VPSDE, and instead use IS to minimize variance. The theory of IS shows that the proposal $r(t) \propto \mathrm{d}\log\sigma_t^2/\mathrm{d}t$ has minimum variance [28]. In App. B, we show that we can sample from $r(t)$ using inverse transform sampling $t = \text{var}^{-1}((\sigma_1^2)^\rho(\sigma_0^2)^{1-\rho})$ where $\text{var}^{-1}$ is the inverse of $\sigma_t^2$ and $\rho \sim \mathcal{U}[0, 1]$. This variance reduction technique is available for any VPSDE with arbitrary $\beta(t)$.

In Fig. 2, we train a small LSGM on CIFAR-10 with $w_{\text{ll}}$ weighting using (i) the original VPSDE with uniform $t$ sampling, (ii) the same SDE but with our IS from $t$, and (iii) the proposed geometric

---

[4] Minimizing $\mathcal{L}(\mathbf{x}, \boldsymbol{\phi}, \boldsymbol{\theta}, \boldsymbol{\psi})$ w.r.t $\boldsymbol{\phi}$ is equivalent to minimizing $\text{KL}\big(q(\mathbf{z}_0|\mathbf{x})||p(\mathbf{z}_0|\mathbf{x})\big)$ w.r.t $q(\mathbf{z}_0|\mathbf{x})$.

VPSDE. Note how both (ii) and (iii) significantly reduce the variance and allow us to monitor the progress of the training objective. In this case, (i) has difficulty minimizing the objective due to the high variance. In App. B, we show how IS proposals can be formed for other SDEs, including the VESDE and Sub-VPSDE from [2].

**Variance reduction for unweighted and reweighted objectives:** When training with $w_{\text{un}}$, analytically deriving IS proposal distributions for arbitrary $\beta(t)$ is challenging. For linear VPSDEs, we provide a derivation in App. B to obtain the optimal IS distribution. In contrast, defining IS proposal distributions is easier when training with $w_{\text{re}}$. In App. B, we show that the optimal distribution is in the form $r(t) \propto \mathrm{d}\sigma_t^2/\mathrm{d}t$ which is sampled by $t=\mathrm{var}^{-1}((1-\rho)\sigma_0^2+\rho\sigma_1^2)$ with $\rho \sim \mathcal{U}[0,1]$. In Fig. 3, we visualize the IS distributions for the three weighting mechanisms for the linear VPSDE with the original $[\beta_0, \beta_1]$ parameters from [2]. $r(t)$ for the likelihood weighting is more tilted towards $t = 0$ due to the $1/\sigma_t^2$ term in $w_{\text{ll}}$.

When using differently weighted objectives for training, we can either sample separate $t$ with different IS distributions for each objective, or use IS for the SGM objective (Eq. 9) and reweight the samples according to the likelihood objective for encoder training (Eq. 8). See App. G for details.

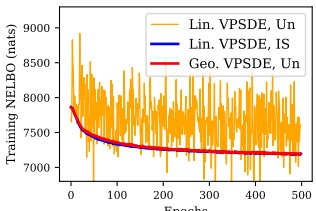

Figure 2: Variance reduction

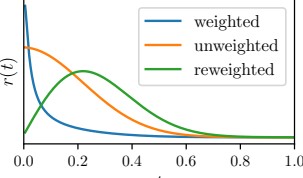

Figure 3: IS distributions

# 4 Related Work

Our work builds on score-matching [29, 30, 31, 32, 33, 34, 35, 36, 37], specifically denoising score matching [22], which makes our work related to recent generative models using denoising score matching- and denoising diffusion-based objectives [3, 38, 1, 2, 6]. Among those, [1, 6] use a discretized diffusion process with many noise scales, building on [27], while Song et al. [2] introduce the continuous time framework using SDEs. Experimentally, these works focus on image modeling and, contrary to us, work directly in pixel space. Various works recently tried to address the slow sampling of these types of models and further improve output quality. [39] add an adversarial objective, [5] introduce non-Markovian diffusion processes that allow to trade off synthesis speed, quality, and sample diversity, [40] learn a sequence of conditional energy-based models for denoising, [41] distill the iterative sampling process into single shot synthesis, and [42] learn an adaptive noise schedule, which is adjusted during synthesis to accelerate sampling. Further, [26] propose empirical variance reduction techniques for discretized diffusions and introduce a new, heuristically motivated, noise schedule. In contrast, our proposed noise schedule and our variance reduction techniques are analytically derived and directly tailored to our learning setting in the continuous time setup.

Recently, [11] presented a method to generate graphs using score-based models, relaxing the entries of adjacency matrices to continuous values. LSGM would allow to model graph data more naturally using encoders and decoders tailored to graphs [43, 44, 45, 46].

Since our model can be considered a VAE [14, 15] with score-based prior, it is related to approaches that improve VAE priors. For example, Normalizing flows and hierarchical distributions [23, 24, 47, 48, 20, 21], as well as energy-based models [49, 50, 51, 52, 53] have been proposed as VAE priors. Furthermore, classifiers [54, 55, 56], adversarial methods [57], and other techniques [58, 59] have been used to define prior distributions implicitly. In two-stage training, a separate generative model is trained in latent space as a new prior after training the VAE itself [60, 61, 62, 63, 64, 10]. Our work also bears a resemblance to recent methods on improving the sampling quality in generative adversarial networks using gradient flows in the latent space [65, 66, 67, 68], with the main difference that these prior works use a discriminator to update the latent variables, whereas we train an SGM.

**Concurrent works:** [10] proposed to learn a denoising diffusion model in the latent space of a VAE for symbolic music generation. This work does not introduce an end-to-end training framework of the combined VAE and denoising diffusion model and instead trains them in two separate stages. In contrast, concurrently with us [69] proposed an end-to-end training approach, and [70] combines contrastive learning with diffusion models in the latent space of VAEs for controllable generation. However, [10, 69, 70] consider the discretized diffusion objective [1], while we build on the continuous time framework. Also, these models are not equipped with the mixed score parameterization and variance reduction techniques, which we found crucial for the successful training of SGM priors.

Additionally, [71, 4, 25] concurrently with us proposed likelihood-based training of SGMs in data space[5]. [4] developed a bound for the data likelihood in their Theorem 3 of their second version, using a denoising score matching objective, closely related to our cross entropy expression. However, our cross entropy expression is much simpler as we show how several terms can be marginalized out analytically for the diffusion SDEs employed by us (see our proof in App. A). The same marginalization can be applied to Theorem 3 in [4] when the drift coefficient takes a special affine form (i.e., $\mathbf{f}(\mathbf{z}, t) = f(t)\mathbf{z}$). Moreover, [25] discusses the likelihood-based training of SGMs from a fundamental perspective and shows how several score matching objectives become a variational bound on the data likelihood. [71] introduced a notion of signal-to-noise ratio (SNR) that results in a noise-invariant parameterization of time that depends only on the initial and final noise. Interestingly, our importance sampling distribution in Sec. 3.4 has a similar noise-invariant parameterization of time via $t = \text{var}^{-1}((\sigma_1^2)^\rho (\sigma_0^2)^{1-\rho})$, which also depends only on the initial and final diffusion process variances. We additionally show that this time parameterization results in the optimal minimum-variance objective, if the distribution of latent variables follows a standard Normal distribution. Finally, [72] proposed a modified time parameterization that allows modeling unbounded data scores.

## 5 Experiments

Here, we examine the efficacy of LSGM in learning generative models for images.

**Implementation details:** We implement LSGM using the NVAE [20] architecture as VAE backbone and NCSN++ [2] as SGM backbone. NVAE has a hierarchical latent structure. The diffusion process input $\mathbf{z}_0$ is constructed by concatenating the latent variables from all groups in the channel dimension. For NVAEs with multiple spatial resolutions in latent groups, we only feed the smallest resolution groups to the SGM prior and assume that the remaining groups have a standard Normal distribution.

**Sampling:** To generate samples from LSGM at test time, we use a black-box ODE solver [73] to sample from the prior. Prior samples are then passed to the decoder to generate samples in data space.

**Evaluation:** We measure NELBO, an upper bound on negative log-likelihood (NLL), using Eq. 6. For estimating $\log p(\mathbf{z}_0)$, we rely on the probability flow ODE [2], which provides an unbiased but stochastic estimation of $\log p(\mathbf{z}_0)$. This stochasticity prevents us from performing an importance weighted estimation of NLL [74] (see App. F for details). For measuring sample quality, Fréchet inception distance (FID) [75] is evaluated with 50K samples. Implementation details in App. G.

### 5.1 Main Results

**Unconditional color image generation:** Here, we present our main results for unconditional image generation on CIFAR-10 [89] (Tab. 2) and CelebA-HQ-256 (5-bit quantized) [88] (Tab. 3). For CIFAR-10, we train 3 different models: *LSGM (FID)* and *LSGM (balanced)* both use the VPSDE with linear $\beta(t)$ and $w_{\text{un}}$-weighting for the SGM prior in Eq. 9, while performing IS as derived in Sec. 3.4. They only differ in how the backbone VAE is trained. *LSGM (NLL)* is a model that is trained with our novel geometric VPSDE, using $w_{\text{ll}}$-weighting in the prior objective (further details in App. G). When set up for high image quality, LSGM achieves a new state-of-the-art FID of 2.10. When tuned towards NLL, we achieve a NELBO of $2.87$, which is significantly better than previous score-based models. Only autoregressive models, which come with very slow synthesis, and VDVAE [21] reach similar or higher likelihoods, but they usually have much poorer image quality.

For CelebA-HQ-256, we observe that when LSGM is trained with different SDE types and weighting mechanisms, it often obtains similar NELBO potentially due to applying the SGM prior only to small latent variable groups and using Normal priors at the larger groups. With $w_{\text{re}}$-weighting and linear VPSDE, LSGM obtains the state-of-the-art FID score of 7.22 on a par with the original SGM [2].

For both datasets, we also report results for the VAE backbone used in our LSGM. Although this baseline achieves competitive NLL, its sample quality is behind our LSGM and the original SGM.

**Modeling binarized images:** Next, we examine LSGM on dynamically binarized MNIST [93] and OMNIGLOT [74]. We apply LSGM to binary images using a decoder with pixel-wise independent Bernoulli distributions. For these datasets, we report both NELBO and NLL in nats in Tab. 4 and Tab. 5. On OMNIGLOT, LSGM achieves state-of-the-art likelihood of $\leq 87.79$ nat, outperforming previous models including VAEs with autoregressive decoders, and even when comparing its NELBO

---

[5]We build on the V1 version of [4], which was substantially updated after the NeurIPS submission deadline.

Table 2: Generative performance on CIFAR-10.

| | Method | NLL↓ | FID↓ |
|---|---|---|---|
| **Ours** | LSGM (FID) | ≤3.43 | **2.10** |
| | LSGM (NLL) | ≤**2.87** | 6.89 |
| | LSGM (balanced) | ≤2.95 | 2.17 |
| | VAE Backbone | 2.96 | 43.18 |
| **VAEs** | VDVAE [21] | **2.87** | - |
| | NVAE [20] | 2.91 | 23.49 |
| | VAEBM [76] | - | 12.19 |
| | NCP-VAE [56] | - | 24.08 |
| | BIVA [48] | 3.08 | - |
| | DC-VAE [77] | - | 17.90 |
| **Score** | NCSN [3] | - | 25.32 |
| | Rec. Likelihood [40] | 3.18 | 9.36 |
| | DSM-ALS [39] | 3.65 | - |
| | DDPM [1] | 3.75 | 3.17 |
| | Improved DDPM [26] | 2.94 | 11.47 |
| | SDE (DDPM++) [2] | 2.99 | 2.92 |
| | SDE (NCSN++) [2] | - | 2.20 |
| **Flows** | VFlow [19] | 2.98 | - |
| | ANF [18] | 3.05 | - |
| **Aut. Reg.** | DistAug aug [78] | 2.53 | 42.90 |
| | Sp. Transformers [79] | 2.80 | - |
| | $\delta$-VAE [80] | 2.83 | - |
| | PixelSNAIL [81] | 2.85 | - |
| | PixelCNN++ [82] | 2.92 | - |
| **GANs** | AutoGAN [83] | - | 12.42 |
| | StyleGAN2-ADA [84] | - | 2.92 |

Table 3: Generative results on CelebA-HQ-256.

| | Method | NLL↓ | FID↓ |
|---|---|---|---|
| **Ours** | LSGM | ≤**0.70** | **7.22** |
| | VAE Backbone | **0.70** | 30.87 |
| **VAEs** | NVAE [20] | **0.70** | 29.76 |
| | VAEBM [76] | - | 20.38 |
| | NCP-VAE [56] | - | 24.79 |
| | DC-VAE [77] | - | 15.80 |
| **Score** | SDE [2] | - | **7.23** |
| **Flows** | GLOW [85] | 1.03 | 68.93 |
| **Aut. Reg.** | SPN [86] | 0.61 | - |
| **GANs** | Adv. LAE [87] | - | 19.21 |
| | VQ-GAN [64] | - | 10.70 |
| | PGGAN [88] | - | 8.03 |

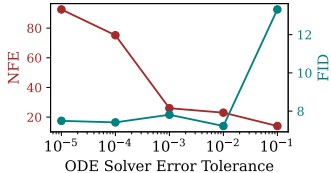

Figure 4: FID and number of function evaluations (NFEs) for different ODE solver error tolerances on CelebA-HQ-256. LSGM takes 4.15 sec. for sampling while the original SGM [2] takes 45 min. with PC and 3.9 min. with ODE-based sampling.

Table 4: Dyn. binarized OMNIGLOT results.

| | Method | NELBO↓ | NLL↓ |
|---|---|---|---|
| **Ours** | LSGM | **87.79** | ≤**87.79** |
| **VAEs** | NVAE [20] | 93.92 | 90.75 |
| | BIVA [48] | 93.54 | 91.34 |
| | DVAE++ [51] | - | 92.38 |
| | Ladder VAE [90] | - | 102.11 |
| **Aut. Reg.** | VLVAE [47] | - | 89.83 |
| | VampPrior [59] | - | 89.76 |
| | PixelVAE++ [91] | - | 88.29 |

Table 5: Dynamically binarized MNIST results.

| | Method | NELBO↓ | NLL↓ |
|---|---|---|---|
| **Ours** | LSGM | **78.47** | ≤78.47 |
| **VAEs** | NVAE [20] | 79.56 | **78.01** |
| | BIVA [48] | 80.06 | 78.41 |
| | IAF-VAE [24] | 80.80 | 79.10 |
| | DVAE++ [51] | - | 78.49 |
| **Aut. Reg.** | PixelVAE++ [91] | - | 78.00 |
| | VampPrior [59] | - | 78.45 |
| | MAE [92] | - | 77.98 |

against importance weighted estimation of NLL for other methods. On MNIST, LSGM outperforms previous VAEs in NELBO, reaching a NELBO 1.09 nat lower than the state-of-the-art NVAE.

**Qualitative results**: We visualize qualitative results for all datasets in Fig. 5. On the complex multimodal CIFAR-10 dataset, LSGM generates sharp and high-quality images. On CelebA-HQ-256, LSGM generates diverse samples from different ethnicity and age groups with varying head poses and facial expressions. On MNIST and OMNIGLOT, the generated characters are sharp and high-contrast.

**Sampling time**: We compare LSGM against the original SGM [2] trained on the CelebA-HQ-256 dataset in terms of sampling time and number of function evaluations (NFEs) of the ODE solver. Song et al. [2] propose two main sampling techniques including predictor-corrector (PC) and probability flow ODE. PC sampling involves 4000 NFEs and takes 44.6 min. on a Titan V for a batch of 16 images. It yields 7.23 FID score (see Tab. 3). ODE-based sampling from SGM takes 3.91 min. with 335 NFEs, but it obtains a poor FID score of 128.13 with $10^{-5}$ as ODE solver error tolerance[6].

In a stark contrast, ODE-based sampling from our LSGM takes 0.07 min. with average of 23 NFEs, yielding 7.22 FID score. LSGM is 637× and 56× faster than original SGM's [2] PC and ODE

---

[6]We use the VESDE checkpoint at https://github.com/yang-song/score_sde_pytorch. Song et al. [2] report that ODE-based sampling yields worse FID scores for their models (see D.4 in [2]). The problem is more severe for VESDEs. Unfortunately, at submission time only a VESDE model was released.

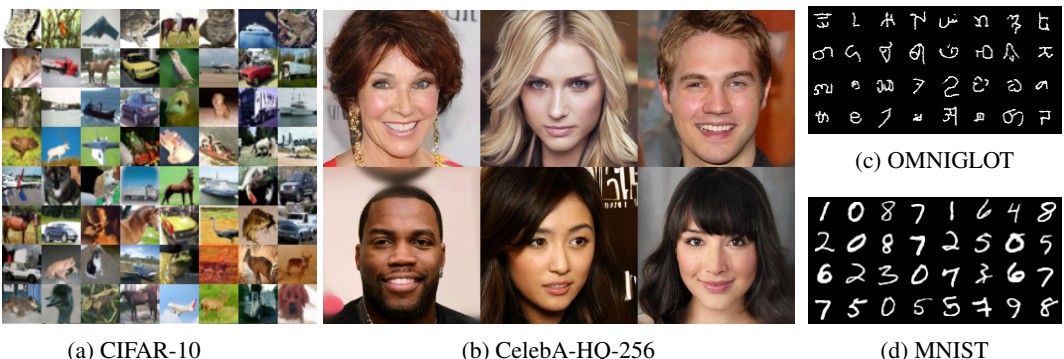

(a) CIFAR-10            (b) CelebA-HQ-256          (c) OMNIGLOT        (d) MNIST

Figure 5: Generated samples for different datasets. For binary datasets, we visualize the decoder mean. LSGM successfully generates sharp, high-quality, and diverse samples (additional samples in appendix).

Table 6: Ablations on SDEs, objectives, weighting mechanisms, and variance reduction. Details in App. G.

| SGM-obj.-weighting | | $w_{\text{ll}}$ | | $w_{\text{un}}$ | | | | $w_{\text{re}}$ | | | |
|---|---|---|---|---|---|---|---|---|---|---|---|
| $t$-sampling (SGM-obj.) | | $\mathcal{U}[0,1]$ | $r_{\text{ll}}(t)$ | $\mathcal{U}[0,1]$ | | $r_{\text{un}}(t)$ | | $\mathcal{U}[0,1]$ | | $r_{\text{re}}(t)$ | |
| $t$-sampling (q-obj.) | | rew. | rew. | rew. | $r_{\text{ll}}(t)$ | rew. | $r_{\text{ll}}(t)$ | rew. | $r_{\text{ll}}(t)$ | rew. | $r_{\text{ll}}(t)$ |
| **Geom.-** | **FID↓** | 10.18 | n/a | NaN | NaN | n/a | n/a | 22.21 | NaN | 7.29 | 7.18 |
| **VPSDE** | **NELBO↓** | 2.96 | n/a | NaN | NaN | n/a | n/a | 3.04 | NaN | 2.99 | 2.99 |
| **VPSDE** | **FID↓** | 6.15 | 8.00 | NaN | NaN | 5.39 | 5.39 | NaN | 4.99 | 15.12 | 6.19 |
| | **NELBO↓** | 2.97 | 2.97 | NaN | NaN | 2.98 | 2.98 | NaN | 2.99 | 3.03 | 2.99 |

sampling, respectively. In Fig. 4, we visualize FID scores and NFEs for different ODE solver error tolerances. Our LSGM achieves low FID scores for relatively large error tolerances.

We identify three main reasons for this significantly faster sampling from LSGM: (i) The SGM prior in our LSGM models latent variables with 32×32 spatial dim., whereas the original SGM [2] directly models 256×256 images. The larger spatial dimensions require a deeper network to achieve a large receptive field. (ii) Inspecting the SGM prior in our model suggests that the score function is heavily dominated by the linear term at the end of training, as the mixing coefficients $\boldsymbol{\alpha}$ are all $< 0.02$. This makes our SGM prior smooth and numerically faster to solve. (iii) Since SGM is formed in the latent space in our model, errors from solving the ODE can be corrected to some degree using the VAE decoder, while in the original SGM [2] errors directly translate to artifacts in pixel space.

## 5.2 Ablation Studies

**SDEs, objective weighting mechanisms and variance reduction.** In Tab. 6, we analyze the different weighting mechanisms and variance reduction techniques and compare the geometric VPSDE with the regular VPSDE with linear $\beta(t)$ [1, 2]. In the table, *SGM-obj.-weighting* denotes the weighting mechanism used when training the SGM prior (via Eq. 9). $t$-sampling (SGM-obj.) indicates the sampling approach for $t$, where $r_{\text{ll}}(t)$, $r_{\text{un}}(t)$ and $r_{\text{re}}(t)$ denote the IS distributions for the weighted (likelihood), the unweighted, and the reweighted objective, respectively. For training the VAE encoder $q_\phi(\mathbf{z}_0|\mathbf{x})$ (last term in Eq. 8), we either sample a separate batch $t$ with importance sampling following $r_{\text{ll}}(t)$ (only necessary when the SGM prior is not trained with $w_{\text{ll}}$ itself), or we *reweight* the samples drawn for training the prior according to the likelihood objective (denoted by *rew.*). *n/a* indicates fields that do not apply: The geometric VPSDE has optimal variance for the weighted (likelihood) objective already with uniform sampling; there is no additional IS distribution. Also, we did not derive IS distributions for the geometric VPSDE for $w_{\text{un}}$. *NaN* indicates experiments that failed due to training instabilities. Previous work [20, 21] have reported instability in training large VAEs. We find that our method inherits similar instabilities from VAEs; however, importance sampling often stabilizes training our LSGM. As expected, we obtain the best NELBOs (red) when training with the weighted, maximum likelihood objective ($w_{\text{ll}}$). Importantly, our new geometric VPSDE achieves the best NELBO. Furthermore, the best FIDs (blue) are obtained either by unweighted ($w_{\text{un}}$) or reweighted ($w_{\text{re}}$) SGM prior training, with only slightly worse NELBOs. These experiments were run on the CIFAR10 dataset, using a smaller model than for our main results above (details in App. G).

**End-to-end training.** We proposed to train LSGM end-to-end, in contrast to [10]. Using a similar setup as above we compare end-to-end training of LSGM during the second stage with freezing the VAE encoder and decoder and only training the SGM prior in latent space during the second stage. When training the model end-to-end, we achieve an FID of 5.19 and NELBO of 2.98; when freezing the VAE networks during the second stage, we only get an FID of 9.00 and NELBO of 3.03. These results clearly motivate our end-to-end training strategy.

**Mixing Normal and neural score functions.** We generally found training LSGM without our proposed "mixed score" formulation (Sec. 3.2) to be unstable during end-to-end training, highlighting its importance. To quantify the contribution of the mixed score parametrization for a stable model, we train a small LSGM with only one latent variable group. In this case, without the mixed score, we reached an FID of 34.71 and NELBO of 3.39; with it, we got an FID of 7.60 and NELBO of 3.29. Without the inductive bias provided by the mixed score, learning that the marginal distribution is close to a Normal one for large $t$ purely from samples can be very hard in the high-dimensional latent space, where our diffusion is run. Furthermore, due to our importance sampling schemes, we tend to oversample small, rather than large $t$. However, synthesizing high-quality images requires an accurate score function estimate for all $t$. On the other hand, the log-likelihood of samples is highly sensitive to local image statistics and primarily determined at small $t$. It is plausible that we are still able to learn a reasonable estimate of the score function for these small $t$ even without the mixed score formulation. That may explain why log-likelihood suffers much less than sample quality, as estimated by FID, when we remove the mixed score parameterization.

Additional experiments and model samples are presented in App. H.

## 6    Conclusions

We proposed the *Latent Score-based Generative Model*, a novel framework for end-to-end training of score-based generative models in the latent space of a variational autoencoder. Moving from data to latent space allows us to form more expressive generative models, model non-continuous data, and reduce sampling time using smoother SGMs. To enable training latent SGMs, we made three core contributions: (i) we derived a simple expression for the cross entropy term in the variational objective, (ii) we parameterized the SGM prior by mixing Normal and neural score functions, and (iii) we proposed several techniques for variance reduction in the estimation of the training objective. Experimental results show that latent SGMs outperform recent pixel-space SGMs in terms of both data likelihood and sample quality, and they can also be applied to binary datasets. In large image generation, LSGM generates data several orders of magnitude faster than recent SGMs. Nevertheless, LSGM's synthesis speed does not yet permit sampling at interactive rates, and our implementation of LSGM is currently limited to image generation. Therefore, future work includes further accelerating sampling, applying LSGMs to other data types, and designing efficient networks for LSGMs.

## 7    Broader Impact

Generating high-quality samples while fully covering the data distribution has been a long-standing challenge in generative learning. A solution to this problem will likely help reduce biases in generative models and lead to improving overall representation of minorities in the data distribution. SGMs are perhaps one of the first deep models that excel at both sample quality and distribution coverage. However, the high computational cost of sampling limits their widespread use. Our proposed LSGM reduces the sampling complexity of SGMs by a large margin and improves their expressivity further. Thus, in the long term, it can enable the usage of SGMs in practical applications.

Here, LSGM is examined on the image generation task which has potential benefits and risks discussed in [94, 95]. However, LSGM can be considered a generic framework that extends SGMs to non-continuous data types. In principle LSGM could be used to model, for example, language [96, 97], music [98, 10], or molecules [99, 100]. Furthermore, like other deep generative models, it can potentially be used also for non-generative tasks such as semi-supervised and representation learning [101, 102, 103]. This makes the long-term social impacts of LSGM dependent on the downstream applications.

## Funding Statement

All authors were funded by NVIDIA through full-time employment.

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
