# Appendix

# A  Proof for Theorem 1

Without loss of generality, we state the theorem in general form without conditioning on $\mathbf{x}$.

**Theorem 1.** *Given two distributions $q(\mathbf{z}_0)$ and $p(\mathbf{z}_0)$ defined in the continuous space $\mathbb{R}^D$, denote the marginal distributions of diffused samples under the SDE $d\mathbf{z} = f(t)\mathbf{z}\,dt + g(t)\,d\mathbf{w}$ at time $t \in [0,1]$ with $q(\mathbf{z}_t)$ and $p(\mathbf{z}_t)$. Assuming that $\log q(\mathbf{z}_t)$ and $\log p(\mathbf{z}_t)$ are smooth with at most polynomial growth at $\mathbf{z}_t \to \pm\infty$, and also assuming that $f(t)$ and $g(t)$ are chosen such that $q(\mathbf{z}_1) = p(\mathbf{z}_1)$ at $t = 1$, the cross entropy is given by:*

$$CE(q(\mathbf{z}_0)||p(\mathbf{z}_0)) = \mathbb{E}_{t\sim\mathcal{U}[0,1]} \left[ \frac{g(t)^2}{2} \mathbb{E}_{q(\mathbf{z}_t,\mathbf{z}_0|\mathbf{x})} \left[ ||\nabla_{\mathbf{z}_t} \log q(\mathbf{z}_t|\mathbf{z}_0) - \nabla_{\mathbf{z}_t} \log p(\mathbf{z}_t)||_2^2 \right] \right] + \frac{D}{2} \log\left( 2\pi e \sigma_0^2 \right),$$

*with $q(\mathbf{z}_t, \mathbf{z}_0) = q(\mathbf{z}_t|\mathbf{z}_0)q(\mathbf{z}_0)$ and a Normal transition kernel $q(\mathbf{z}_t|\mathbf{z}_0) = \mathcal{N}(\mathbf{z}_t; \boldsymbol{\mu}_t(\mathbf{z}_0), \sigma_t^2 \mathbf{I})$ where $\boldsymbol{\mu}_t$ and $\sigma_t^2$ are obtained from $f(t)$ and $g(t)$ for a fixed initial variance $\sigma_0^2$ at $t = 0$.*

Theorem 1 amounts to estimating the cross entropy between $q(\mathbf{z}_0)$ and $p(\mathbf{z}_0)$ with denoising score matching and can be understood intuitively in the context of LSGM: We are drawing samples from a potentially complex encoding distribution $q(\mathbf{z}_0)$, add Gaussian noise with small initial variance $\sigma_0^2$ to obtain a well-defined initial distribution, and then smoothly perturb the sampled encodings using a diffusion process, while learning a denoising model, the SGM prior. Note that from the perspective of the learnt SGM prior, which is defined by the separate reverse-time generative SDE with the learnt score function model (see Sec. 2), the expression in our theorem becomes an upper bound (see discussion in Sec. 3.1).

*Proof.* The first part of our proof follows a similar proof strategy as was used by Song et al. [4]. We start the proof with a more generic diffusion process in the form:

$$d\mathbf{z} = \mathbf{f}(\mathbf{z},t)dt + g(t)d\mathbf{w}$$

The time-evolution of probability densities $q(\mathbf{z}_t)$ and $p(\mathbf{z}_t)$ under this SDE is described by the Fokker-Planck equation [104] (note that we follow the same notation as in the main paper: We omit the $t$-subscript of the diffused distributions $q_t$, indicating the time dependence at the variable, i.e. $q(\mathbf{z}_t) \equiv q_t(\mathbf{z}_t)$):

$$\begin{aligned}
\frac{\partial q(\mathbf{z}_t)}{\partial t} &= \nabla_{\mathbf{z}_t}\left( \frac{1}{2}g^2(t)q(\mathbf{z}_t)\nabla_{\mathbf{z}_t}\log q(\mathbf{z}_t) - \mathbf{f}(\mathbf{z},t)q(\mathbf{z}_t) \right) \\
&= \nabla_{\mathbf{z}_t}\left( \mathbf{h}_q(\mathbf{z}_t,t)q(\mathbf{z}_t) \right)
\end{aligned} \tag{10}$$

with

$$\mathbf{h}_q(\mathbf{z}_t,t) := \frac{1}{2}g^2(t)\nabla_{\mathbf{z}_t}\log q(\mathbf{z}_t) - \mathbf{f}(\mathbf{z},t) \tag{11}$$

and analogously for $p(\mathbf{z}_t)$.

The cross entropy can be written as

$$\begin{aligned}
\mathrm{CE}(q(\mathbf{z}_0)||p(\mathbf{z}_0)) &= \mathrm{CE}(q(\mathbf{z}_1)||p(\mathbf{z}_1)) + \int_1^0 \frac{\partial}{\partial t}\mathrm{CE}(q(\mathbf{z}_t)||p(\mathbf{z}_t))dt \\
&= \mathrm{H}\big(q(\mathbf{z}_1)\big) - \int_0^1 \frac{\partial}{\partial t}\mathrm{CE}(q(\mathbf{z}_t)||p(\mathbf{z}_t))dt
\end{aligned}$$

since $q(\mathbf{z}_1) = p(\mathbf{z}_1)$, as assumed in the Theorem (in practice, the used SDEs are designed such that $q(\mathbf{z}_1) = p(\mathbf{z}_1)$).

Furthermore, we have

$$
\begin{aligned}
\frac{\partial}{\partial t}\mathrm{CE}(q(\mathbf{z}_t)||p(\mathbf{z}_t)) &= -\int \left[ \frac{\partial q(\mathbf{z}_t)}{\partial t} \log p(\mathbf{z}_t) + \frac{q(\mathbf{z}_t)}{p(\mathbf{z}_t)} \frac{\partial p(\mathbf{z}_t)}{\partial t} \right] d\mathbf{z} \\
&\overset{(i)}{=} -\int \left[ \nabla_{\mathbf{z}_t}(\mathbf{h}_q(\mathbf{z}_t, t) q(\mathbf{z}_t)) \log p(\mathbf{z}_t) + \frac{q(\mathbf{z}_t)}{p(\mathbf{z}_t)} \nabla_{\mathbf{z}_t}(\mathbf{h}_p(\mathbf{z}_t, t) p(\mathbf{z}_t)) \right] d\mathbf{z} \\
&\overset{(ii)}{=} \int \left[ \mathbf{h}_q(\mathbf{z}_t, t)^\top q(\mathbf{z}_t) \nabla_{\mathbf{z}_t} \log p(\mathbf{z}_t) + \mathbf{h}_p(\mathbf{z}_t, t)^\top p(\mathbf{z}_t) \nabla_{\mathbf{z}_t} \frac{q(\mathbf{z}_t)}{p(\mathbf{z}_t)} \right] d\mathbf{z} \\
&\overset{(iii)}{=} \int q(\mathbf{z}_t) \big[ \mathbf{h}_q(\mathbf{z}_t, t)^\top \nabla_{\mathbf{z}_t} \log p(\mathbf{z}_t) \\
&\qquad\qquad + \mathbf{h}_p(\mathbf{z}_t, t)^\top \nabla_{\mathbf{z}_t} \log q(\mathbf{z}_t) \\
&\qquad\qquad - \mathbf{h}_p(\mathbf{z}_t, t)^\top \nabla_{\mathbf{z}_t} \log p(\mathbf{z}_t) \big] d\mathbf{z} \\
&\overset{(iv)}{=} \int q(\mathbf{z}_t) \Big[ -\frac{1}{2} g^2(t) ||\nabla_{\mathbf{z}_t} \log p(\mathbf{z}_t)||^2 - \mathbf{f}(\mathbf{z}_t, t)^\top \nabla_{\mathbf{z}_t} \log q(\mathbf{z}_t) \\
&\qquad\qquad + g^2(t) \nabla_{\mathbf{z}_t} \log q(\mathbf{z}_t)^\top \nabla_{\mathbf{z}_t} \log p(\mathbf{z}_t) \Big] d\mathbf{z}
\end{aligned}
$$

where $(i)$ inserts the Fokker Planck equations for $q(\mathbf{z}_t)$ and $p(\mathbf{z}_t)$, respectively. Furthermore, $(ii)$ is integration by parts assuming similar limiting behavior of $q(\mathbf{z}_t)$ and $p(\mathbf{z}_t)$ at $\mathbf{z}_t \to \pm\infty$ as Song et al. [4]. Specifically, we know that $q(\mathbf{z}_t)$ and $p(\mathbf{z}_t)$ must decay towards zero at $\mathbf{z}_t \to \pm\infty$ to be normalized. Furthermore, we assumed $\log q(\mathbf{z}_t)$ and $\log p(\mathbf{z}_t)$ to have at most polynomial growth (or decay, when looking at it from the other direction) at $\mathbf{z}_t \to \pm\infty$, which implies faster exponential growth/decay of $q(\mathbf{z}_t)$ and $p(\mathbf{z}_t)$. Also, $\nabla_{\mathbf{z}_t} \log q(\mathbf{z}_t)$ and $\nabla_{\mathbf{z}_t} \log p(\mathbf{z}_t)$ grow/decay at most polynomially, too, since the gradient of a polynomial is still a polynomial. Hence, one can work out that all terms to be evaluated at $\mathbf{z}_t \to \pm\infty$ after integration by parts vanish. Finally, $(iii)$ uses the log derivative trick and some rearrangements, and $(iv)$ is obtained by inserting $\mathbf{h}_q$ and $\mathbf{h}_p$.

Hence, we obtain

$$
\begin{aligned}
\mathrm{CE}(q(\mathbf{z}_0)||p(\mathbf{z}_0)) &= \mathrm{H}\big(q(\mathbf{z}_1)\big) + \int_0^1 \mathbb{E}_{q(\mathbf{z}_t)} \Big[ \frac{1}{2} g^2(t) ||\nabla_{\mathbf{z}_t} \log p(\mathbf{z}_t)||_2^2 + \mathbf{f}(\mathbf{z}_t, t) \nabla_{\mathbf{z}_t} \log q(\mathbf{z}_t) \\
&\qquad\qquad - g^2(t) \nabla_{\mathbf{z}_t} \log q(\mathbf{z}_t)^\top \nabla_{\mathbf{z}_t} \log p(\mathbf{z}_t) \Big] dt,
\end{aligned}
$$

which we can interpret as a general score matching-based expression for calculating the cross entropy, analogous to the expressions for the Kullback-Leibler divergence and entropy derived by Song et al. [4].

However, as discussed in the main paper, dealing with the marginal score $\nabla_{\mathbf{z}_t} \log q(\mathbf{z}_t)$ is problematic for complex "input" distributions $q(\mathbf{z}_0)$. Hence, we further transform the cross entropy expression into a denoising score matching-based expression:

$$\text{CE}(q(\mathbf{z}_0)||p(\mathbf{z}_0)) = \text{H}\big(q(\mathbf{z}_1)\big) + \int_0^1 \mathbb{E}_{q(\mathbf{z}_t)}\left[\frac{1}{2}g^2(t)||\nabla_{\mathbf{z}_t}\log p(\mathbf{z}_t)||_2^2 + \mathbf{f}(\mathbf{z}_t,t)\nabla_{\mathbf{z}_t}\log q(\mathbf{z}_t)\right.$$

$$\left. - g^2(t)\nabla_{\mathbf{z}_t}\log q(\mathbf{z}_t)^\top \nabla_{\mathbf{z}_t}\log p(\mathbf{z}_t)\right]dt$$

$$\overset{(i)}{=} \frac{1}{2}\int_0^1 g(t)^2 \mathbb{E}_{q(\mathbf{z}_0,\mathbf{z}_t)}\left[-2\nabla_{\mathbf{z}}\log q(\mathbf{z}_t|\mathbf{z}_0)^\top \nabla_{\mathbf{z}_t}\log p(\mathbf{z}_t) + ||\nabla_{\mathbf{z}_t}\log p(\mathbf{z}_t)||_2^2\right]dt$$

$$+ \frac{1}{2}\int_0^1 \mathbb{E}_{q(\mathbf{z}_0,\mathbf{z}_t)}\left[2\mathbf{f}(\mathbf{z},t)^\top \nabla_{\mathbf{z}_t}\log q(\mathbf{z}_t|\mathbf{z}_0)\right]dt + \text{H}\big(q(\mathbf{z}_1)\big)$$

$$\overset{(ii)}{=} \frac{1}{2}\int_0^1 g(t)^2 \mathbb{E}_{q(\mathbf{z}_0,\mathbf{z}_t)}\left[||\nabla_{\mathbf{z}_t}\log q(\mathbf{z}_t|\mathbf{z}_0)||_2^2 - 2\nabla_{\mathbf{z}_t}\log q(\mathbf{z}_t|\mathbf{z}_0)^\top \nabla_{\mathbf{z}_t}\log p(\mathbf{z}_t) + ||\nabla_{\mathbf{z}_t}\log p(\mathbf{z}_t)||_2^2\right]dt$$

$$+ \frac{1}{2}\int_0^1 \mathbb{E}_{q(\mathbf{z}_t)}\left[2\mathbf{f}(\mathbf{z},t)^\top \nabla_{\mathbf{z}_t}\log q(\mathbf{z}_t|\mathbf{z}_0) - g(t)^2||\nabla_{\mathbf{z}_t}\log q(\mathbf{z}_t|\mathbf{z}_0)||_2^2\right]dt + \text{H}\big(q(\mathbf{z}_1)\big)$$

$$\overset{(iii)}{=} \frac{1}{2}\int_0^1 g(t)^2 \mathbb{E}_{q(\mathbf{z}_0,\mathbf{z}_t)}\left[||\nabla_{\mathbf{z}_t}\log q(\mathbf{z}_t|\mathbf{z}_0) - \nabla_{\mathbf{z}_t}\log p(\mathbf{z}_t)||_2^2\right]dt$$

$$+ \underbrace{\frac{1}{2}\int_0^1 \mathbb{E}_{q(\mathbf{z}_0,\mathbf{z}_t)}\left[\left(2\mathbf{f}(\mathbf{z},t) - g(t)^2\nabla_{\mathbf{z}_t}\log q(\mathbf{z}_t|\mathbf{z}_0)\right)^\top \nabla_{\mathbf{z}_t}\log q(\mathbf{z}_t|\mathbf{z}_0)\right]dt}_{\text{(I): Model-independent term}} + \text{H}\big(q(\mathbf{z}_1)\big)$$

with $q(\mathbf{z}_0,\mathbf{z}_t) = q(\mathbf{z}_t|\mathbf{z}_0)q(\mathbf{z}_0)$ and where in $(i)$ we have used the following identity from Vincent [22]:

$$\mathbb{E}_{q(\mathbf{z}_t)}\left[\nabla_{\mathbf{z}_t}\log q(\mathbf{z}_t)\right] = \mathbb{E}_{q(\mathbf{z}_t)}\left[\mathbb{E}_{q(\mathbf{z}_0|\mathbf{z}_t)}\left[\nabla_{\mathbf{z}_t}\log q(\mathbf{z}_t|\mathbf{z}_0)\right]\right] = \mathbb{E}_{q(\mathbf{z}_0)q(\mathbf{z}_t|\mathbf{z}_0)}\left[\nabla_{\mathbf{z}_t}\log q(\mathbf{z}_t|\mathbf{z}_0)\right].$$

In $(ii)$, we have added and subtracted $g(t)^2||\nabla_{\mathbf{z}_t}\log q(\mathbf{z}_t|\mathbf{z}_0)||_2^2$ and in $(iii)$ we rearrange the terms into denoising score matching. In the following, we show that the term marked by (I) depends only on the diffusion parameters and does not depend on $q(\mathbf{z}_0)$ when $\mathbf{f}(\mathbf{z},t)$ takes a special affine (linear) form $\mathbf{f}(\mathbf{z},t) := f(t)\mathbf{z}$, which is often used for training SGMs and which we assume in our Theorem.

Note that for linear $\mathbf{f}(\mathbf{z},t) := f(t)\mathbf{z}$, we can derive the mean and variance (there are no "off-diagonal" co-variance terms here, since all dimensions undergo diffusion independently) of the distribution $q(\mathbf{z}_t|\mathbf{z}_0)$ at any time $t$ in closed form, essentially solving the Fokker-Planck equation for this special case analytically. In that case, if the initial distribution at $t=0$ is Normal then the distribution stays Normal and the mean and variance completely describe the distribution, i.e. $q(\mathbf{z}_t|\mathbf{z}_0) = \mathcal{N}(\mathbf{z}_t; \boldsymbol{\mu}_t(\mathbf{z}_0), \sigma_t^2\mathbf{I})$. The mean and variance are given by the differential equations and their solutions [104]:

$$\frac{d\boldsymbol{\mu}}{dt} = f(t)\boldsymbol{\mu} \qquad \rightarrow \boldsymbol{\mu}_t = \mathbf{z}_0 e^{\int_0^t f(s)ds} \tag{12}$$

$$\frac{d\sigma^2}{dt} = 2f(t)\sigma^2 + g^2(t) \rightarrow \sigma_t^2 = \frac{1}{\tilde{F}(t)}\left(\int_0^t \tilde{F}(s)g^2(s)ds + \sigma_0^2\right), \quad \tilde{F}(t) := e^{-2\int_0^t f(s)ds} \tag{13}$$

Here, $\mathbf{z}_0$ denotes the mean of the distribution at $t=0$ and $\sigma_0^2$ the component-wise variance at $t=0$. After transforming into the denoising score matching expression above, what we are doing is essentially drawing samples $\mathbf{z}_0$ from the potentially complex $q(\mathbf{z}_0)$, then placing simple Normal distributions with variance $\sigma_0^2$ at those samples, and then letting those distributions evolve according to the SDE. $\sigma_0^2$ acts as a hyperparameter of the model.

In this case, i.e. when the distribution $q(\mathbf{z}_t|\mathbf{z}_0)$ is Normal at all $t$, we can represent samples $\mathbf{z}_t$ from the intermediate distributions in reparameterized from $\mathbf{z}_t = \boldsymbol{\mu}_t(\mathbf{z}_0) + \sigma_t\boldsymbol{\epsilon}$ where $\boldsymbol{\epsilon} \sim \mathcal{N}(\boldsymbol{\epsilon}; \mathbf{0}, \mathbf{I})$. We

also know that $\nabla_{\mathbf{z}} \log q(\mathbf{z}_t|\mathbf{z}_0) = -\frac{\boldsymbol{\epsilon}}{\sigma_t}$ With this we can write down (i) as:

$$(I) = \frac{1}{2} \int_0^1 \mathbb{E}_{q(\mathbf{z}_0),\boldsymbol{\epsilon}} \left[ \left( 2f(t)(\boldsymbol{\mu}_t(\mathbf{z}_0) + \sigma_t\boldsymbol{\epsilon}) + g(t)^2 \frac{\boldsymbol{\epsilon}}{\sigma_t} \right)^T \left( -\frac{\boldsymbol{\epsilon}}{\sigma_t} \right) \right] dt \tag{14}$$

$$= \int_0^1 -\frac{f(t)}{\sigma_t} \underbrace{\mathbb{E}_{q(\mathbf{z}_0),\boldsymbol{\epsilon}} \left[ \boldsymbol{\mu}_t(\mathbf{z}_0)^T \boldsymbol{\epsilon} \right]}_{=0} - \frac{2f(t)\sigma_t^2 + g(t)^2}{2\sigma_t^2} \underbrace{\mathbb{E}_{\boldsymbol{\epsilon}}[\boldsymbol{\epsilon}^T \boldsymbol{\epsilon}]}_{=D} dt \tag{15}$$

$$= -\frac{D}{2} \int_0^1 \frac{2f(t)\sigma_t^2 + g(t)^2}{\sigma_t^2} dt \tag{16}$$

$$= -\frac{D}{2} \int_{\sigma_0^2}^{\sigma_1^2} \frac{1}{\sigma_t^2} d\sigma_t^2 = \frac{D}{2} (\log \sigma_0^2 - \log \sigma_1^2), \tag{17}$$

where we have used Eq. 13.

Furthermore, since $q(\mathbf{z}_T) \to \mathcal{N}(\mathbf{z}_T, \mathbf{0}, \sigma_1^2 \mathbf{I})$ at $t = 1$, its entropy is $\mathrm{H}(q(\mathbf{z}_T)) = \frac{D}{2} \log(2\pi e \sigma_1^2)$. With this, we get the following simple expression for the cross-entropy:

$$\mathrm{CE}(q(\mathbf{z}_0)||p(\mathbf{z}_0)) = \frac{1}{2} \int_0^1 g(t)^2 \mathbb{E}_{q(\mathbf{z}_0,\mathbf{z}_t)} \left[ ||\nabla_{\mathbf{z}} \log q(\mathbf{z}_t|\mathbf{z}_0) - \nabla_{\mathbf{z}} \log p(\mathbf{z}_t)||_2^2 \right] dt + D \log(\sqrt{2\pi e \sigma_0^2})$$

Expressing the integral as an expectation completes the proof:

$$\mathrm{CE}(q(\mathbf{z}_0)||p(\mathbf{z}_0)) = \mathbb{E}_{t \sim \mathcal{U}[0,1]} \left[ \frac{g(t)^2}{2} \mathbb{E}_{q(\mathbf{z}_t,\mathbf{z}_0)} \left[ ||\nabla_{\mathbf{z}_t} \log q(\mathbf{z}_t|\mathbf{z}_0) - \nabla_{\mathbf{z}_t} \log p(\mathbf{z}_t)||_2^2 \right] \right] + \frac{D}{2} \log \left( 2\pi e \sigma_0^2 \right)$$

$$\square$$

The expression in Theorem 1 measures the cross entropy between $q$ and $p$ at $t = 0$. However, one should consider practical implications of the choice of initial variance $\sigma_0^2$ when estimating the cross entropy between two distributions using our expression, as we discuss below.

Consider two arbitrary distributions $q'(\mathbf{z})$ and $p'(\mathbf{z})$. If the forward diffusion process has a non-zero initial variance (i.e., $\sigma_0^2 > 0$), the actual distributions $q$ and $p$ at $t = 0$ in the score matching expression are defined by $q(\mathbf{z}_0) := \int q'(\mathbf{z})\mathcal{N}(\mathbf{z}_0, \mathbf{z}, \sigma_0^2 \mathbf{I})d\mathbf{z}$ and $p(\mathbf{z}_0) := \int p'(\mathbf{z})\mathcal{N}(\mathbf{z}_0, \mathbf{z}, \sigma_0^2 \mathbf{I})d\mathbf{z}$, which correspond to convolving $q'(\mathbf{z})$ and $p'(\mathbf{z})$ each with a Normal distribution with variance $\sigma_0^2 \mathbf{I}$. In this case, $q'(\mathbf{z})$ and $p'(\mathbf{z})$ are not identical to $q(\mathbf{z}_0)$ and $p(\mathbf{z}_0)$, respectively, in general. However, we can approximate $q'(\mathbf{z})$ and $p'(\mathbf{z})$ using $p(\mathbf{z}_0)$ and $q(\mathbf{z}_0)$, respectively, when $\sigma_0^2$ is small. That is why our expression in Theorem 1 that measures $\mathrm{CE}(q(\mathbf{z}_0)||p(\mathbf{z}_0))$, can be considered as an approximation of $\mathrm{CE}(q'(\mathbf{z})||p'(\mathbf{z}))$ when $\sigma_0^2$ takes a positive small value. Note that in practice, our $\sigma_0^2$ is indeed generally very small (see Tab. 7).

On the other hand, when $\sigma_0^2 = 0$ (e.g., when using the VPSDE from Song et al. [2]), we know that $q'(\mathbf{z})$ and $p'(\mathbf{z})$ are identical to $q(\mathbf{z}_0)$ and $p(\mathbf{z}_0)$. However, in this case, the initial distribution at $t = 0$ is essentially an infinitely sharp Normal and we cannot evaluate the integral over the full interval $t \in [0, 1]$. Hence, we limit its range to $t \in [\epsilon, 1]$, where $\epsilon$ is another hyperparameter. In this case, we can approximate the cross entropy $\mathrm{CE}(q'(\mathbf{z})||p'(\mathbf{z}))$ using:

$$\mathrm{CE}(q(\mathbf{z}_0)||p(\mathbf{z}_0)) \approx \frac{1}{2} \int_\epsilon^1 g(t)^2 \mathbb{E}_{q(\mathbf{z}_0,\mathbf{z}_t)} \left[ ||\nabla_{\mathbf{z}} \log q(\mathbf{z}_t|\mathbf{z}_0) - \nabla_{\mathbf{z}} \log p(\mathbf{z}_t)||_2^2 \right] dt + D \log(\sqrt{2\pi e \sigma_\epsilon^2})$$

$$= \mathbb{E}_{t \sim \mathcal{U}[\epsilon,1]} \left[ \frac{g(t)^2}{2} \mathbb{E}_{q(\mathbf{z}_t,\mathbf{z}_0)} \left[ ||\nabla_{\mathbf{z}_t} \log q(\mathbf{z}_t|\mathbf{z}_0) - \nabla_{\mathbf{z}_t} \log p(\mathbf{z}_t)||_2^2 \right] \right] + \frac{D}{2} \log \left( 2\pi e \sigma_\epsilon^2 \right)$$

## B  Variance Reduction

The variance of the cross entropy in a mini-batch update depends on the variance of $\mathrm{CE}(q(\mathbf{z}_0)||p(\mathbf{z}_0))$ where $q(\mathbf{z}_0) := \mathbb{E}_{p_{\mathrm{data}}(\mathbf{x})}[q(\mathbf{z}_0|\mathbf{x})]$ is the aggregate posterior (i.e., the distribution of latent variables) and $p_{\mathrm{data}}$ is the data distribution. This is because, for training, we use a mini-batch

estimation of $\mathbb{E}_{p_{\text{data}}(\mathbf{x})}[\mathcal{L}(\mathbf{x}, \phi, \boldsymbol{\theta}, \psi)]$. For the cross entropy term in $\mathcal{L}(\mathbf{x}, \phi, \boldsymbol{\theta}, \psi)$, we have $\mathbb{E}_{p_{\text{data}}(\mathbf{x})}[\text{CE}(q(\mathbf{z}_0|\mathbf{x})||p(\mathbf{z}_0))] = \text{CE}(q(\mathbf{z}_0)||p(\mathbf{z}_0))$.

In order to study the variance of the training objective, we derive $\text{CE}(q(\mathbf{z}_0)||p(\mathbf{z}_0))$ analytically, assuming that both $q(\mathbf{z}_0) = p(\mathbf{z}_0) = \mathcal{N}(\mathbf{z}_0; \mathbf{0}, \mathbf{I})$. This is a reasonable simplification for our analysis because pretraining our LSGM model with a $\mathcal{N}(\mathbf{z}_0; \mathbf{0}, \mathbf{I})$ prior brings $q(\mathbf{z}_0)$ close to $\mathcal{N}(\mathbf{z}_0; \mathbf{0}, \mathbf{I})$ and our SGM prior is often dominated by the fixed Normal mixture component. Nevertheless, we empirically observe that the variance reduction techniques developed with this simplification still work well when $q(\mathbf{z}_0)$ and $p(\mathbf{z}_0)$ are not exactly $\mathcal{N}(\mathbf{z}_0; \mathbf{0}, \mathbf{I})$.

In this section, we start with presenting the mixed score parameterization for generic SDEs in App. B.1. Then, we discuss variance reduction with importance sampling for these generic SDEs in App. B.2. Finally, in App. B.3 and App. B.4, we focus on variance reduction of the VPSDEs and VESDEs, respectively, and we briefly discuss the Sub-VPSDE [2] in App. B.5.

## B.1 Generic Mixed Score Parameterization for Non-Variance Preserving SDEs

The mixed score parameterization uses the score that is obtained when dealing with Normal input data and just predicts an additional residual score. In the main text, we assume that the variance of the standard Normal data stays the same throughout the diffusion process, which is the case for VPSDEs. But the way Normal data diffuses depends generally on the underlying SDE and generic SDEs behave differently than the regular VPSDE in that regard.

Consider the generic forward SDEs in the form:

$$\mathrm{d}\mathbf{z} = f(t)\mathbf{z}\,\mathrm{d}t + g(t)\,\mathrm{d}\mathbf{w} \tag{18}$$

If our data distribution is standard Normal, i.e. $\mathbf{z}_0 \sim \mathcal{N}(\mathbf{z}_0; \mathbf{0}, \mathbf{I})$, using Eq. 13, we have

$$\mathring{\sigma}_t^2 := \frac{1}{\tilde{F}(t)}\left(\int_0^t \tilde{F}(s)g^2(s)ds + 1\right) = \frac{1}{\tilde{F}(t)}\left(\tilde{\sigma}_t^2 + 1\right) \tag{19}$$

with the definition $\tilde{\sigma}_t^2 := \int_0^t \tilde{F}(s)g^2(s)ds$. Hence, the score function at time $t$ is $\nabla_{\mathbf{z}_t} \log p(\mathbf{z}_t) = -\frac{\mathbf{z}_t}{\mathring{\sigma}_t^2}$. Using the geometric mixture $p(\mathbf{z}_t) \propto \mathcal{N}(\mathbf{z}_t; 0, \mathring{\sigma}_t^2)^{1-\alpha} p'_{\boldsymbol{\theta}}(\mathbf{z}_t)^\alpha$, we can generally define our mixed score parameterization as

$$\boldsymbol{\epsilon}_\theta(\mathbf{z}_t, t) := \frac{\sigma_t}{\mathring{\sigma}_t^2}(1 - \boldsymbol{\alpha}) \odot \mathbf{z}_t + \boldsymbol{\alpha} \odot \boldsymbol{\epsilon}'_\theta(\mathbf{z}_t, t). \tag{20}$$

In the case of VPSDEs, we have $\mathring{\sigma}_t^2 = 1$ which corresponds to the mixed score introduced in the main text.

**Remark:** It is worth noting that both $\mathring{\sigma}_t^2$ and $\sigma_t^2$ are solutions to the same differential equation in Eq. 13 with different initial conditions. It is easy to see that $\mathring{\sigma}_t^2 - \sigma_t^2 = (1 - \sigma_0^2)\tilde{F}(t)^{-1}$.

## B.2 Variance Reduction of Cross Entropy with Importance Sampling for Generic SDEs

Let's consider the cross entropy expression for $p(\mathbf{z}_0) = \mathcal{N}(\mathbf{z}_0, \mathbf{0}, \mathbf{I})$ and $q(\mathbf{z}_0) = \mathcal{N}(\mathbf{z}_0, \mathbf{0}, (1 - \sigma_0^2)\mathbf{I})$ where we have scaled down the variance of $q(\mathbf{z}_0)$ to $(1 - \sigma_0^2)$ to accommodate the fact that the diffusion process with initial variance $\sigma_0^2$ applies a perturbation with variance $\sigma_0^2$ in its initial step (hence, the marginal distribution at $t = 0$ is $\mathcal{N}(\mathbf{z}_0, \mathbf{0}, \mathbf{I})$ and we know that the optimal score is $\boldsymbol{\epsilon}_\theta(\mathbf{z}_t, t) = \frac{\sigma_t}{\mathring{\sigma}_t^2}\mathbf{z}_t$, i.e., the Normal component).

The cross entropy $\text{CE}(q(\mathbf{z}_0)||p(\mathbf{z}_0))$ with the optimal score $\boldsymbol{\epsilon}_\theta(\mathbf{z}_t, t) = \frac{\sigma_t}{\mathring{\sigma}_t^2}\mathbf{z}_t$ is:

$$\text{CE} - \text{const.} = \frac{1}{2}\int_\epsilon^1 \frac{g^2(t)}{\sigma_t^2}\mathbb{E}_{\mathbf{z}_0,\boldsymbol{\epsilon}}\left[||\boldsymbol{\epsilon} - \boldsymbol{\epsilon}_\theta(\mathbf{z}_t, t)||_2^2\right]dt \tag{21}$$

$$= \frac{1}{2}\int_\epsilon^1 \frac{g^2(t)}{\sigma_t^2}\mathbb{E}_{\mathbf{z}_0,\boldsymbol{\epsilon}}\left[||\boldsymbol{\epsilon} - \frac{\sigma_t}{\mathring{\sigma}_t^2}\mathbf{z}_t||_2^2\right]dt \tag{22}$$

$$= \frac{1}{2}\int_\epsilon^1 \frac{g^2(t)}{\sigma_t^2}\mathbb{E}_{\mathbf{z}_0,\boldsymbol{\epsilon}}\left[||\boldsymbol{\epsilon} - \frac{\sigma_t}{\mathring{\sigma}_t^2}(\tilde{F}(t)^{-\frac{1}{2}}\mathbf{z}_0 + \boldsymbol{\epsilon}\sigma_t)||_2^2\right]dt \tag{23}$$

$$= \frac{1}{2}\int_\epsilon^1 \frac{g^2(t)}{\sigma_t^2}\mathbb{E}_{\mathbf{z}_0,\boldsymbol{\epsilon}}\left[||\frac{\mathring{\sigma}_t^2 - \sigma_t^2}{\mathring{\sigma}_t^2}\boldsymbol{\epsilon} - \frac{\sigma_t}{\mathring{\sigma}_t^2}\tilde{F}(t)^{-\frac{1}{2}}\mathbf{z}_0||_2^2\right]dt \tag{24}$$

$$= \frac{1}{2}\int_\epsilon^1 \frac{g^2(t)}{\sigma_t^2}\left(\frac{(\mathring{\sigma}_t^2 - \sigma_t^2)^2}{(\mathring{\sigma}_t^2)^2}\mathbb{E}_{\boldsymbol{\epsilon}}\left[||\boldsymbol{\epsilon}||_2^2\right] + \frac{\sigma_t^2}{(\mathring{\sigma}_t^2)^2}\tilde{F}(t)^{-1}\mathbb{E}_{\mathbf{z}_0}\left[||\mathbf{z}_0||_2^2\right]\right)dt \tag{25}$$

$$= \frac{D}{2}\int_\epsilon^1 \frac{g^2(t)}{\sigma_t^2}\left(\frac{(\mathring{\sigma}_t^2 - \sigma_t^2)^2}{(\mathring{\sigma}_t^2)^2} + \frac{\sigma_t^2}{(\mathring{\sigma}_t^2)^2}\tilde{F}(t)^{-1}(1 - \sigma_0^2)\right)dt \tag{26}$$

$$= \frac{D}{2}\int_\epsilon^1 \frac{g^2(t)}{\sigma_t^2}\left(\frac{(\mathring{\sigma}_t^2 - \sigma_t^2)^2}{(\mathring{\sigma}_t^2)^2} + \frac{\sigma_t^2(\mathring{\sigma}_t^2 - \sigma_t^2)}{(\mathring{\sigma}_t^2)^2}\right)dt \tag{27}$$

$$= \frac{D}{2}\int_\epsilon^1 \frac{g^2(t)}{\sigma_t^2}dt - \frac{D}{2}\int_\epsilon^1 \frac{g^2(t)}{\mathring{\sigma}_t^2}dt \tag{28}$$

$$= \frac{D}{2}\int_\epsilon^1 \frac{\frac{d}{dt}\sigma_t^2 + 2f(t)\sigma_t^2}{\sigma_t^2}dt - \frac{D}{2}\int_\epsilon^1 \frac{\frac{d}{dt}\mathring{\sigma}_t^2 + 2f(t)\mathring{\sigma}_t^2}{\mathring{\sigma}_t^2}dt \tag{29}$$

$$= \frac{D}{2}\int_\epsilon^1 \frac{\frac{d}{dt}\sigma_t^2}{\sigma_t^2}dt - \frac{D}{2}\int_\epsilon^1 \frac{\frac{d}{dt}\mathring{\sigma}_t^2}{\mathring{\sigma}_t^2}dt \tag{30}$$

$$= D\frac{1-\epsilon}{2}\mathbb{E}_{t\sim\mathcal{U}[\epsilon,1]}\left[\frac{d}{dt}\log\left(\frac{\sigma_t^2}{\mathring{\sigma}_t^2}\right)\right] \tag{31}$$

$$= D\frac{1-\epsilon}{2}\mathbb{E}_{t\sim\mathcal{U}[\epsilon,1]}\left[\frac{d}{dt}\log\left(\frac{\tilde{\sigma}_t^2 + \sigma_0^2}{\tilde{\sigma}_t^2 + 1}\right)\right], \tag{32}$$

where in Eq. 23, we have used $\mathbf{z}_t = \tilde{F}(t)^{-\frac{1}{2}}\mathbf{z}_0 + \boldsymbol{\epsilon}\sigma_t$. In Eq. 25, we have used the fact that $\mathbf{z}_0$ and $\boldsymbol{\epsilon}$ are independent. In Eq. 27, we have used the identity $\mathring{\sigma}_t^2 - \sigma_t^2 = (1 - \sigma_0^2)\tilde{F}(t)^{-1}$. In Eq. 29, we have used $g^2(t) = \frac{d}{dt}\sigma_t^2 + 2f(t)\sigma_t^2$ from Eq. 13.

Therefore, the IW distribution with minimum variance for $\text{CE}(q(\mathbf{z}_0)||p(\mathbf{z}_0))$ is

$$r(t) \propto \frac{d}{dt}\log\left(\frac{\tilde{\sigma}_t^2 + \sigma_0^2}{\tilde{\sigma}_t^2 + 1}\right) \tag{33}$$

with normalization constant

$$\tilde{R} = \log\left(\left(\frac{\tilde{\sigma}_1^2 + \sigma_0^2}{\tilde{\sigma}_1^2 + 1}\right)\left(\frac{\tilde{\sigma}_\epsilon^2 + 1}{\tilde{\sigma}_\epsilon^2 + \sigma_0^2}\right)\right) \tag{34}$$

and CDF

$$R(t) = \frac{1}{\tilde{R}}\log\left(\left(\frac{\tilde{\sigma}_t^2 + \sigma_0^2}{\tilde{\sigma}_t^2 + 1}\right)\left(\frac{\tilde{\sigma}_\epsilon^2 + 1}{\tilde{\sigma}_\epsilon^2 + \sigma_0^2}\right)\right) \tag{35}$$

Hence, the inverse CDF is

$$t = \left(\tilde{\sigma}_t^2\right)^{inv}\left(\frac{\sigma_0^2 - \left(\frac{\tilde{\sigma}_\epsilon^2 + \sigma_0^2}{\tilde{\sigma}_\epsilon^2 + 1}\right)^{1-\rho}\left(\frac{\tilde{\sigma}_1^2 + \sigma_0^2}{\tilde{\sigma}_1^2 + 1}\right)^\rho}{\left(\frac{\tilde{\sigma}_\epsilon^2 + \sigma_0^2}{\tilde{\sigma}_\epsilon^2 + 1}\right)^{1-\rho}\left(\frac{\tilde{\sigma}_1^2 + \sigma_0^2}{\tilde{\sigma}_1^2 + 1}\right)^\rho - 1}\right) \tag{36}$$

Finally, the cross entropy objective with importance weighting becomes

$$\frac{1}{2}\int_\epsilon^1 \frac{g^2(t)}{\sigma_t^2}\mathbb{E}_{\mathbf{z}_0,\boldsymbol{\epsilon}}\left[||\boldsymbol{\epsilon}-\boldsymbol{\epsilon}_\theta(\mathbf{z}_t,t)||_2^2\right]dt = \frac{\tilde{R}}{2}\mathbb{E}_{t\sim r(t)}\left[\frac{1+\tilde{\sigma}_t^2}{1-\sigma_0^2}\mathbb{E}_{\mathbf{z}_0,\boldsymbol{\epsilon}}||\boldsymbol{\epsilon}-\boldsymbol{\epsilon}_\theta(\mathbf{z}_t,t)||_2^2\right] \quad (37)$$

$$= \frac{1}{2}\log\left(\left(\frac{\tilde{\sigma}_1^2+\sigma_0^2}{\tilde{\sigma}_1^2+1}\right)\left(\frac{\tilde{\sigma}_\epsilon^2+1}{\tilde{\sigma}_\epsilon^2+\sigma_0^2}\right)\right)\mathbb{E}_{t\sim r(t)}\left[\frac{1+\tilde{\sigma}_t^2}{1-\sigma_0^2}\mathbb{E}_{\mathbf{z}_0,\boldsymbol{\epsilon}}||\boldsymbol{\epsilon}-\boldsymbol{\epsilon}_\theta(\mathbf{z}_t,t)||_2^2\right] \quad (38)$$

The idea here is to write everything as a function of $\tilde{\sigma}_t^2 = \int_0^t \tilde{F}(s)g^2(s)ds$. We see that $\tilde{\sigma}_t^2$ is monotonically increasing for any $g(t)$ and $f(t)$; hence, it always has an inverse and inverse transform sampling is, in principle, always possible. However, we should pick $g(t)$ and $f(t)$ such that $\tilde{\sigma}_t^2$ and its inverse are also analytically tractable to avoid dealing with numerical methods.

### B.3 VPSDE

Consider the simple forward diffusion process in the form:

$$d\mathbf{z} = -\frac{1}{2}\beta(t)\mathbf{z}dt + \sqrt{\beta(t)}d\mathbf{w} \quad (39)$$

which corresponds to the VPSDE from Song et al. [2]. The appealing characteristic of this diffusion model is that if $\mathbf{z}_0 \sim \mathcal{N}(\mathbf{z}_0;\mathbf{0},\mathbf{I})$, intermediate $\mathbf{z}(t)$ will also have a standard Normal distribution and its variance is constant (i.e., $\frac{d}{dt}\mathring{\sigma}_t^2 = 0$). In the original VPSDE, $\beta(t)$ is defined by a linear function $\beta(t) = \beta_0 + (\beta_1 - \beta_0)t$ that interpolates between $[\beta_0, \beta_1]$.

### B.3.1 Variance Reduction for Likelihood Weighting (Geometric VPSDE)

Our analysis in App. B.2, Eq. 30 shows that the cross entropy can be expressed as:

$$\text{CE}(q(\mathbf{z}_0)||p(\mathbf{z}_0)) - \text{const} = \frac{D}{2}\int_\epsilon^1 \frac{\frac{d}{dt}\sigma_t^2}{\sigma_t^2}dt - \frac{D}{2}\int_\epsilon^1 \frac{\frac{d}{dt}\mathring{\sigma}_t^2}{\mathring{\sigma}_t^2}dt \quad (40)$$

$$= \frac{D}{2}\int_\epsilon^1 \frac{\frac{d}{dt}\sigma_t^2}{\sigma_t^2}dt \quad (41)$$

$$= D\frac{1-\epsilon}{2}\mathbb{E}_{t\sim\mathcal{U}[\epsilon,1]}\left[\frac{\frac{d}{dt}\sigma_t^2}{\sigma_t^2}\right] \quad (42)$$

where for the VPSDE we have used $\frac{d}{dt}\mathring{\sigma}_t^2 = 0$.

A sample-based estimation of this expectation has a low variance if $\frac{1}{\sigma_t^2}\frac{d\sigma_t^2}{dt}$ is constant for all $t \in [0,1]$. By solving the ODE $\frac{1}{\sigma_t^2}\frac{d\sigma_t^2}{dt} = const.$, we can see that a log-linear noise schedule of the form $\sigma_t^2 = \sigma_{\min}^2(\frac{\sigma_{\max}^2}{\sigma_{\min}^2})^t$ satisfies this condition, with $t \in [0,1]$, $0 < \sigma_{\min}^2 < \sigma_{\max}^2 < 1$, and $\sigma_{\min}^2 = \sigma_0^2$.

Using Eq. 13, we can find an expression for $\beta(t)$ that generates such noise schedule:

$$\beta(t) = \frac{1}{1-\sigma_t^2}\frac{d\sigma_t^2}{dt} = \frac{\sigma_t^2}{1-\sigma_t^2}\log(\frac{\sigma_{\max}^2}{\sigma_{\min}^2}) = \frac{\sigma_{\min}^2(\frac{\sigma_{\max}^2}{\sigma_{\min}^2})^t}{1-\sigma_{\min}^2(\frac{\sigma_{\max}^2}{\sigma_{\min}^2})^t}\log(\frac{\sigma_{\max}^2}{\sigma_{\min}^2}) \quad (43)$$

We call a VPSDE with $\beta(t)$ defined as above a *geometric VPSDE*. For small $\sigma_{\min}^2$ and $\sigma_{\max}^2$ close to 1, all inputs diffuse closely towards the standard Normal prior at $t = 1$. In that regard, notice that our geometric VPSDE is well-behaved with positive $\beta(t)$ only within the relevant interval $t \in [0,1]$ and for $0 < \sigma_{\min}^2 < \sigma_{\max}^2 < 1$. These conditions also imply $\sigma_t^2 < 1$ for all $t \in [0,1]$. This is expected for any VPSDE. We can approach unit variance arbitrarily closely but not reach it exactly.

Importantly, our geometric VPSDE is different from the "variance-exploding" SDE (VESDE), proposed by Song et al. [5] (also see App. C). The VESDE leverages a SDE in which the variance grows in an almost unbounded way, while the mean of the input distribution stays constant. Because

of this, the hyperparameters of the VESDE must be chosen carefully in a data-dependent manner [38], which can be problematic in our case (see discussion in App. B.4). Furthermore, Song et al. also found that the VESDE does not perform well when used with probability flow-based sampling [2]. In contrast, our geometric VPSDE combines the variance preserving behavior (i.e. standard Normal input data remains standard Normal throughout the diffusion process; all individual inputs diffuse towards standard Normal prior) of the VPSDE with the geometric growth of the variance in the diffusion process, which was first used in the VESDE.

Finally, for the geometric VPSDE we also have that $\frac{\partial}{\partial t}\text{CE}(q(\mathbf{z}_t)||p(\mathbf{z}_t)) = const.$ for Normal input data. Hence, data is encoded "as continuously as possible" throughout the diffusion process. This is in line with the arguments made by Song et al. in [38]. We hypothesize that this is particularly beneficial towards learning models with strong likelihood or NELBO performance. Indeed, in our experiments we observe the geometric VPSDE to perform best on this metric.

### B.3.2 Variance Reduction for Likelihood Weighting (Importance Sampling)

Above, we have assumed that we sample from a uniform distribution for $t$ and we have defined $\beta(t)$ and $\sigma_t^2$ such that the variance of a Monte-Carlo estimation of the expectation is minimum. Another approach for improving the sample-based estimate of the expectation is to keep $\beta(t)$ and $\sigma_t^2$ unchanged and to use importance sampling such that the variance of the estimate is minimum.

Using importance sampling, we can rewrite the expectation in Eq. 42 as:

$$\mathbb{E}_{t\sim\mathcal{U}[\epsilon,1]}\left[\frac{1}{\sigma_t^2}\frac{d\sigma_t^2}{dt}\right] = \mathbb{E}_{t\sim r(t)}\left[\frac{1}{r(t)}\frac{1}{\sigma_t^2}\frac{d\sigma_t^2}{dt}\right] \tag{44}$$

where $r(t)$ is a proposal distribution. The theory of importance sampling [28] shows that $r(t) \propto \frac{1}{\sigma_t^2}\frac{d\sigma_t^2}{dt} = \frac{d\log\sigma_t^2}{dt}$ will have the smallest variance. In order to use this proposal distribution, we require (i) sampling from $r(t)$ and (ii) evaluating the objective using this importance sampling technique.

**Sampling from $r(t)$ by inverse transform sampling:** It's easy to see that the normalization constant for $r(t)$ is $\int_\epsilon^1 \frac{d\log\sigma_t^2}{dt}dt = \log\sigma_1^2 - \log\sigma_\epsilon^2$. Thus, the PDF $r(t)$ is:

$$r(t) = \frac{1}{\log\sigma_1^2 - \log\sigma_\epsilon^2}\frac{1}{\sigma_t^2}\frac{d\sigma_t^2}{dt} = \frac{\beta(t)(1-\sigma_t^2)}{(\log\sigma_1^2 - \log\sigma_\epsilon^2)\sigma_t^2} \tag{45}$$

We can derive inverse transform sampling by deriving the inverse CDF:

$$R(t) = \frac{\log\frac{\sigma_t^2}{\sigma_\epsilon^2}}{\log\frac{\sigma_1^2}{\sigma_\epsilon^2}} = \rho \Rightarrow \frac{\sigma_t^2}{\sigma_\epsilon^2} = \left(\frac{\sigma_1^2}{\sigma_\epsilon^2}\right)^\rho \Rightarrow t = \text{var}^{-1}\left(\left(\sigma_1^2\right)^\rho\left(\sigma_\epsilon^2\right)^{1-\rho}\right) \tag{46}$$

where $\text{var}^{-1}$ is the inverse of $\sigma_t^2$.

**Importance Weighted Objective:** The cross entropy is then written as (ignoring the constants here):

$$\frac{1}{2}\int_\epsilon^1 \frac{\beta(t)}{\sigma_t^2}\mathbb{E}_{\mathbf{z}_0,\boldsymbol{\epsilon}}\left[||\boldsymbol{\epsilon} - \boldsymbol{\epsilon}_\theta(\mathbf{z}_t,t)||_2^2\right]dt = \frac{1}{2}\mathbb{E}_{t\sim r(t)}\left[\frac{(\log\sigma_1^2 - \log\sigma_\epsilon^2)}{(1-\sigma_t^2)}\mathbb{E}_{\mathbf{z}_0,\boldsymbol{\epsilon}}||\boldsymbol{\epsilon} - \boldsymbol{\epsilon}_\theta(\mathbf{z}_t,t)||_2^2\right] \tag{47}$$

### B.3.3 Variance Reduction for Unweighted Objective

Using a similar derivation as in App. B.2, we can show that for the unweighted objective for $p(\mathbf{z}_0) = \mathcal{N}(\mathbf{z}_0,\mathbf{0},\mathbf{I})$ and $q(\mathbf{z}_0) = \mathcal{N}(\mathbf{z}_0,\mathbf{0},(1-\sigma_0^2)\mathbf{I})$, we have

$$\int_\epsilon^1 \mathbb{E}_{\mathbf{z}_0,\boldsymbol{\epsilon}}\left[||\boldsymbol{\epsilon} - \boldsymbol{\epsilon}_\theta(\mathbf{z}_t,t)||_2^2\right]dt = \frac{D}{2}\int_\epsilon^1\left(\frac{(\mathring{\sigma}_t^2 - \sigma_t^2)^2}{(\mathring{\sigma}_t^2)^2} + \frac{\sigma_t^2(\mathring{\sigma}_t^2 - \sigma_t^2)}{(\mathring{\sigma}_t^2)^2}\right)dt \tag{48}$$

$$= D\frac{1-\epsilon}{2}\mathbb{E}_{t\sim\mathcal{U}[\epsilon,1]}\left[1-\sigma_t^2\right] \tag{49}$$

$$= D\frac{1-\epsilon}{2}\mathbb{E}_{t\sim r(t)}\left[\frac{1-\sigma_t^2}{r(t)}\right] \tag{50}$$

with proposal distribution $r(t) \propto 1 - \sigma_t^2$. Recall that in the VPSDE with linear $\beta(t) = \beta_0 + (\beta_1 - \beta_0)t$, we have

$$1 - \sigma_t^2 = (1 - \sigma_0^2)e^{-\int_0^t \beta(s)ds} = (1 - \sigma_0^2)e^{-\beta_0 t - (\beta_1 - \beta_0)\frac{t^2}{2}} \tag{51}$$

Hence, the normalization constant of $r(t)$ is

$$\tilde{R} = \int_\epsilon^1 (1 - \sigma_0^2)e^{-\beta_0 t - (\beta_1 - \beta_0)\frac{t^2}{2}} dt \tag{52}$$

$$= \underbrace{(1 - \sigma_0^2)e^{\frac{1}{2}\frac{\beta_0}{\beta_1 - \beta_0}}\sqrt{\frac{\pi}{2(\beta_1 - \beta_0)}}}_{:= A_{\tilde{R}}} \left[ \operatorname{erf}\left( \sqrt{\frac{\beta_1 - \beta_0}{2}} \left[ 1 + \frac{\beta_0}{\beta_1 - \beta_0} \right] \right) - \operatorname{erf}\left( \sqrt{\frac{\beta_1 - \beta_0}{2}} \left[ \epsilon + \frac{\beta_0}{\beta_1 - \beta_0} \right] \right) \right]$$

$$\tag{53}$$

Similarly, we can write the CDF of $r(t)$ as

$$R(t) = \frac{A_{\tilde{R}}}{\tilde{R}} \left[ \operatorname{erf}\left( \sqrt{\frac{\beta_1 - \beta_0}{2}} \left[ t + \frac{\beta_0}{\beta_1 - \beta_0} \right] \right) - \operatorname{erf}\left( \sqrt{\frac{\beta_1 - \beta_0}{2}} \left[ \epsilon + \frac{\beta_0}{\beta_1 - \beta_0} \right] \right) \right] \tag{54}$$

solving $\rho = R(t)$ for $t$ then results in

$$t = \sqrt{\frac{2}{\beta_1 - \beta_0}} \operatorname{erfinv}\left( \frac{\rho \tilde{R}}{A_{\tilde{R}}} + \operatorname{erf}\left( \sqrt{\frac{\beta_1 - \beta_0}{2}} \left[ \epsilon + \frac{\beta_0}{\beta_1 - \beta_0} \right] \right) \right) - \frac{\beta_0}{\beta_1 - \beta_0} \tag{55}$$

**Importance Weighted Objective:**

$$\int_\epsilon^1 \mathbb{E}_{\mathbf{z}_0, \boldsymbol{\epsilon}} \left[ ||\boldsymbol{\epsilon} - \boldsymbol{\epsilon}_\theta(\mathbf{z}_t, t)||_2^2 \right] dt = \mathbb{E}_{t \sim r(t)} \left[ \frac{\tilde{R}}{(1 - \sigma_t^2)} \mathbb{E}_{\mathbf{z}_0, \boldsymbol{\epsilon}} ||\boldsymbol{\epsilon} - \boldsymbol{\epsilon}_\theta(\mathbf{z}_t, t)||_2^2 \right] \tag{56}$$

### B.3.4 Variance Reduction for Reweighted Objective

For the reweighted mechanism, we drop only $\sigma_t^2$ from the cross entropy objective but we keep $g^2(t) = \beta(t)$. Using a similar derivation in App. B.2, we can show that unweighted objective for $p(\mathbf{z}_0) = \mathcal{N}(\mathbf{z}_0, \mathbf{0}, \mathbf{I})$ and $q(\mathbf{z}_0) = \mathcal{N}(\mathbf{z}_0, \mathbf{0}, (1 - \sigma_0^2)\mathbf{I})$, we have

$$\int_\epsilon^1 \beta(t)\mathbb{E}_{\mathbf{z}_0, \boldsymbol{\epsilon}} \left[ ||\boldsymbol{\epsilon} - \boldsymbol{\epsilon}_\theta(\mathbf{z}_t, t)||_2^2 \right] dt = D\frac{1 - \epsilon}{2} \mathbb{E}_{t \sim \mathcal{U}[\epsilon, 1]} \left[ \frac{d\sigma_t^2}{dt} \right] = D\frac{1 - \epsilon}{2} \mathbb{E}_{t \sim r(t)} \left[ \frac{\frac{d\sigma_t^2}{dt}}{r(t)} \right] \tag{57}$$

with proposal distribution $r(t) \propto \frac{d\sigma_t^2}{dt} = \beta(t)(1 - \sigma_t^2)$.

In this case, we have the following proposal $r(t)$, its CDF $R(t)$ and inverse CDF $R^{-1}(\rho)$:

$$r(t) = \frac{\beta(t)(1 - \sigma_t^2)}{\sigma_1^2 - \sigma_\epsilon^2}, \quad R(t) = \frac{\sigma_t^2 - \sigma_\epsilon^2}{\sigma_1^2 - \sigma_\epsilon^2}, \quad t = R^{-1}(\rho) = \operatorname{var}^{-1}((1 - \rho)\sigma_\epsilon^2 + \rho\sigma_1^2) \tag{58}$$

Note that usually $\sigma_\epsilon^2 \gtrapprox 0$ and $\sigma_1^2 \lessapprox 1$. In that case, the inverse CDF can be thought of as $R^{-1}(\rho) \approx \operatorname{var}^{-1}(\rho)$.

**Importance Weighted Objective:**

$$\frac{1}{2}\int_\epsilon^1 \beta(t)\mathbb{E}_{\mathbf{z}_0, \boldsymbol{\epsilon}} \left[ ||\boldsymbol{\epsilon} - \boldsymbol{\epsilon}_\theta(\mathbf{z}_t, t)||_2^2 \right] dt = \frac{1}{2}\mathbb{E}_{t \sim r(t)} \left[ \frac{(\sigma_1^2 - \sigma_\epsilon^2)}{(1 - \sigma_t^2)} \mathbb{E}_{\mathbf{z}_0, \boldsymbol{\epsilon}} ||\boldsymbol{\epsilon} - \boldsymbol{\epsilon}_\theta(\mathbf{z}_t, t)||_2^2 \right] \tag{59}$$

**Remark:** It is worth noting that the derivation of the importance sampling distribution for the reweighted objective does not make any assumption on the form of $\beta(t)$. Thus, the IS distribution can be formed for any VPSDE when training with the reweighted objective, including the original VPSDE with linear $\beta(t)$ and also our new geometric VPSDE.

### B.4 VESDE

The VESDE [2] is defined by:

$$d\mathbf{z} = \sqrt{\frac{d}{dt}\sigma(t)^2}d\mathbf{w} \tag{60}$$

$$= \sqrt{\sigma_{\min}^2 \log\left(\frac{\sigma_{\max}^2}{\sigma_{\min}^2}\right)\left(\frac{\sigma_{\max}^2}{\sigma_{\min}^2}\right)^t}d\mathbf{w} \tag{61}$$

with $\sigma(t)^2 = \sigma_{\min}^2\left(\frac{\sigma_{\max}^2}{\sigma_{\min}^2}\right)^t$.

Solving the Fokker-Planck equation for input distribution $\mathcal{N}(\mu_0, \sigma_0^2)$ results in

$$\mu_t = \mu_0; \qquad \sigma_t^2 = \sigma_0^2 - \sigma_{\min}^2 + \sigma_{\min}^2\left(\frac{\sigma_{\max}^2}{\sigma_{\min}^2}\right)^t \tag{62}$$

Typical values for $\sigma_{\min}^2$ and $\sigma_{\max}^2$ are $\sigma_{\min}^2 = 0.01^2$ and $\sigma_{\max}^2 = 50^2$ (CIFAR10). Usually, we use $\sigma_{\min}^2 = \sigma_0^2$.

Note that when the input data is distributed as $\mathbf{z}_0 \sim \mathcal{N}(\mathbf{z}_0; \mathbf{0}, \mathbf{I})$, the variance at time $t$ in VESDE is given by:

$$\mathring{\sigma}_t^2 = 1 - \sigma_{\min}^2 + \sigma_{\min}^2\left(\frac{\sigma_{\max}^2}{\sigma_{\min}^2}\right)^t \tag{63}$$

Note that $\sigma_{\max}^2$ is typically very large and chosen empirically based on the scale of the data [38]. However, this is tricky in our case, as the role of the data is played by the latent space encodings, which themselves are changing during training. We did briefly experiment with the VESDE and calculated $\sigma_{\max}^2$ as suggested in [38] using the encodings after the VAE pre-training stage. However, these experiments were not successful and we suffered from significant training instabilities, even with variance reduction techniques. Therefore, we did not further explore this direction.

Nevertheless, our proposed variance reduction techniques via importance sampling can be derived also for the VESDE. Hence, for completeness, they are shown below.

#### B.4.1 Variance Reduction for Likelihood Weighting

Let's have a closer look at the likelihood objective when using the VESDE for modeling the standard Normal data. Following similar arguments as in previous sections, we have $\mathbf{z}_0 \sim \mathcal{N}(\mathbf{z}_0; \mathbf{0}, (1 - \sigma_{\min}^2)\mathbf{I})$. With the optimal score $\boldsymbol{\epsilon}_\theta(\mathbf{z}_t, t) = \frac{\sigma_t}{\mathring{\sigma}_t^2}\mathbf{z}_t$ (i.e., the Normal component), we have the following expression for $\mathrm{CE}(q(\mathbf{z}_0)||p(\mathbf{z}_0))$ from Eq. 30:

$$\frac{1}{2}\int_\epsilon^1 \frac{g^2(t)}{\sigma_t^2}\mathbb{E}_{\boldsymbol{\mu}_0,\boldsymbol{\epsilon}}\left[||\boldsymbol{\epsilon} - \boldsymbol{\epsilon}_\theta(\mathbf{z}_t, t)||_2^2\right]dt = \frac{D}{2}\int_\epsilon^1 \frac{\frac{d}{dt}\sigma_t^2}{\sigma_t^2}dt - \frac{D}{2}\int_\epsilon^1 \frac{\frac{d}{dt}\mathring{\sigma}_t^2}{\mathring{\sigma}_t^2}dt = \tag{64}$$

$$\frac{D}{2}\int_\epsilon^1 \left[\frac{\frac{d}{dt}\sigma_t^2}{\sigma_t^2} - \frac{\frac{d}{dt}\mathring{\sigma}_t^2}{\mathring{\sigma}_t^2}\right]dt = D\frac{1-\epsilon}{2}\mathbb{E}_{t\sim\mathcal{U}[\epsilon,1]}\left[\frac{\frac{d}{dt}\sigma_t^2}{\sigma_t^2} - \frac{\frac{d}{dt}\mathring{\sigma}_t^2}{\mathring{\sigma}_t^2}\right] \tag{65}$$

*Since the term inside the expectation is not constant in t, the VESDE does not result in an objective with naturally minimal variance, opposed to our proposed geometric VPSDE.*

We derive an importance sampling scheme with a proposal distribution

$$r(t) \propto \frac{1}{\sigma_t^2}\frac{d\sigma_t^2}{dt} - \frac{1}{\mathring{\sigma}_t^2}\frac{d\mathring{\sigma}_t^2}{dt} = \log\left(\frac{\sigma_{\max}^2}{\sigma_{\min}^2}\right)\left(1 - \frac{\sigma_{\min}^2\left(\frac{\sigma_{\max}^2}{\sigma_{\min}^2}\right)^t}{1 - \sigma_{\min}^2 + \sigma_{\min}^2\left(\frac{\sigma_{\max}^2}{\sigma_{\min}^2}\right)^t}\right) \tag{66}$$

Note that the quantity above is always positive as $\frac{\sigma_{\min}^2 \left(\frac{\sigma_{\max}^2}{\sigma_{\min}^2}\right)^t}{1-\sigma_{\min}^2+\sigma_{\min}^2\left(\frac{\sigma_{\max}^2}{\sigma_{\min}^2}\right)^t} \le 1$ with $\sigma_{\min}^2 < 1$. In this case

the normalization constant of $r(t)$ is $\tilde{R} = \log\left(\frac{\mathring{\sigma}_\epsilon^2}{\sigma_\epsilon^2}\frac{\sigma_{\max}^2}{\mathring{\sigma}_1^2}\right)$ and the CDF is:

$$R(t) = \frac{1}{\tilde{R}}\left[\log\sigma_t^2 - \log\sigma_\epsilon^2 + \log\mathring{\sigma}_\epsilon^2 - \log\mathring{\sigma}_t^2\right] = \frac{1}{\tilde{R}}\log\left(\frac{\mathring{\sigma}_\epsilon^2\sigma_t^2}{\mathring{\sigma}_t^2\sigma_\epsilon^2}\right) \tag{67}$$

And the inverse CDF is:

$$t = \mathring{\text{var}}^{-1}\left(\frac{1-\sigma_{\min}^2}{1-\left(\frac{\sigma_\epsilon^2}{\mathring{\sigma}_\epsilon^2}\right)^{1-\rho}\left(\frac{\sigma_{\max}^2}{\mathring{\sigma}_1^2}\right)^\rho}\right) \tag{68}$$

where $\mathring{\text{var}}^{-1}$ is the inverse of $\mathring{\sigma}_t^2$.

So, the objective with importance sampling is then:

$$\frac{1}{2}\int_\epsilon^1 \frac{g^2(t)}{\sigma_t^2}\mathbb{E}_{\mathbf{z}_0,\boldsymbol{\epsilon}}\left[||\boldsymbol{\epsilon}-\boldsymbol{\epsilon}_\theta(\mathbf{z}_t,t)||_2^2\right]dt = \frac{1}{2}\mathbb{E}_{t\sim r(t)}\left[\log\left(\frac{\mathring{\sigma}_\epsilon^2}{\sigma_\epsilon^2}\frac{\sigma_{\max}^2}{\mathring{\sigma}_1^2}\right)\frac{\mathring{\sigma}_t^2}{1-\sigma_{\min}^2}\mathbb{E}_{\mathbf{z}_0,\boldsymbol{\epsilon}}||\boldsymbol{\epsilon}-\boldsymbol{\epsilon}_\theta(\mathbf{z}_t,t)||_2^2\right]$$

In contrast to the VESDE, the geometric VPSDE combines the geometric progression in diffusion variance directly with minimal variance in the objective by design. Furthermore, it is simpler to set up, because we can always choose $\sigma_{\max}^2 \sim 1$ for the geometric VPSDE and do not have to use a data-specific $\sigma_{\max}^2$ as proposed by [38].

### B.4.2 Variance Reduction for Unweighted Objective

When we drop all "prefactors" in the objective, the importance sampling distribution stays the same as above, since $\frac{g^2(t)}{\sigma_t^2}$ is constant in $t$. The objective becomes:

$$\int_\epsilon^1 \mathbb{E}_{\mathbf{z}_0,\boldsymbol{\epsilon}}\left[||\boldsymbol{\epsilon}-\boldsymbol{\epsilon}_\theta(\mathbf{z}_t,t)||_2^2\right]dt = \mathbb{E}_{t\sim r(t)}\left[\frac{\log\left(\frac{\mathring{\sigma}_\epsilon^2}{\sigma_\epsilon^2}\frac{\sigma_{\max}^2}{\mathring{\sigma}_1^2}\right)}{\log\left(\frac{\sigma_{\max}^2}{\sigma_{\min}^2}\right)}\frac{\mathring{\sigma}_t^2}{1-\sigma_{\min}^2}\mathbb{E}_{\mathbf{z}_0,\boldsymbol{\epsilon}}||\boldsymbol{\epsilon}-\boldsymbol{\epsilon}_\theta(\mathbf{z}_t,t)||_2^2\right] \tag{69}$$

### B.4.3 Variance Reduction for Reweighted Objective

To define the importance sampling for the reweighted objective by $\sigma_t^2$, we use the fact that $\frac{d\sigma_t^2}{dt} = \frac{d\mathring{\sigma}_t^2}{dt}$ in VESDEs. Using a similar derivation as in App. B.2, we show:

$$\frac{1}{2}\int_\epsilon^1 g^2(t)\mathbb{E}_{\mathbf{z}_0,\boldsymbol{\epsilon}}\left[||\boldsymbol{\epsilon}-\boldsymbol{\epsilon}_\theta(\mathbf{z}_t,t)||_2^2\right]dt = \frac{D}{2}\int_\epsilon^1 \frac{d\sigma_t^2}{dt}dt - \frac{D}{2}\int_\epsilon^1 \frac{d\mathring{\sigma}_t^2}{dt}\frac{\sigma_t^2}{\mathring{\sigma}_t^2}dt \tag{70}$$

$$= \frac{D}{2}\int_\epsilon^1 \frac{d\mathring{\sigma}_t^2}{dt}\left(\frac{\mathring{\sigma}_t^2-\sigma_t^2}{\mathring{\sigma}_t^2}\right)dt \tag{71}$$

$$= \frac{D(1-\sigma_0^2)}{2}\int_\epsilon^1 \frac{1}{\mathring{\sigma}_t^2}\frac{d\mathring{\sigma}_t^2}{dt}dt \tag{72}$$

Thus, the optimal proposal for reweighted objective and the inverse CDF are:

$$r(t) \sim \frac{1}{\mathring{\sigma}_t^2}\frac{d\mathring{\sigma}_t^2}{dt} \Rightarrow r(t) = \frac{1}{\log(\frac{\mathring{\sigma}_1^2}{\mathring{\sigma}_\epsilon^2})}\frac{1}{\mathring{\sigma}_t^2}\frac{d\mathring{\sigma}_t^2}{dt} \Rightarrow R(t) = \frac{\log(\frac{\mathring{\sigma}_t^2}{\mathring{\sigma}_\epsilon^2})}{\log(\frac{\mathring{\sigma}_1^2}{\mathring{\sigma}_\epsilon^2})} \Rightarrow t = \mathring{\text{var}}^{-1}\left((\mathring{\sigma}_\epsilon^2)^{1-\rho}(\mathring{\sigma}_1^2)^\rho\right) \tag{73}$$

So, the reweighted objective with importance sampling is:

$$\frac{1}{2}\int_\epsilon^1 g^2(t)\mathbb{E}_{\boldsymbol{\mu}_0,\boldsymbol{\epsilon}}\left[||\boldsymbol{\epsilon}-\boldsymbol{\epsilon}_\theta(\mathbf{z}_t,t)||_2^2\right]dt = \frac{1}{2}\mathbb{E}_{t\sim r(t)}\left[\log\left(\frac{\mathring{\sigma}_1^2}{\mathring{\sigma}_\epsilon^2}\right)\mathring{\sigma}_t^2\mathbb{E}_{\boldsymbol{\mu}_0,\boldsymbol{\epsilon}}||\boldsymbol{\epsilon}-\boldsymbol{\epsilon}_\theta(\mathbf{z}_t,t)||_2^2\right] \quad (74)$$

Note that in practice, we can safely set $\epsilon = 0$ as initial $\sigma_0^2$ is non-zero in the VESDE.

### B.5 Sub-VPSDE

Song et al. also proposed the Sub-VPSDE [2]. It is defined as:

$$d\mathbf{z} = -\frac{1}{2}\beta(t)\mathbf{z}dt + \sqrt{\beta(t)\left(1-e^{-2\int_0^t\beta(s)ds}\right)}d\mathbf{w} \quad (75)$$

with the same linear $\beta(t)$ as for the regular VPSDE.

Solving the Fokker-Planck equation for input distribution $\mathcal{N}(\mu_0,\sigma_0^2)$ at $t=0$ results in

$$\mu_t = e^{-\frac{1}{2}\int_0^t\beta(s)ds}\mu_0; \qquad \sigma_t^2 = \left(1.0-e^{-\int_0^t\beta(s)ds}\right)^2 + \sigma_0^2\,e^{-\int_0^t\beta(s)ds} \quad (76)$$

Deriving importance sampling distributions for variance reduction for the Sub-VPSDE can be more complicated than for the VPSDE, Geometric VPSDE, and VESDE and we did not investigate this in detail. However, for the same linear $\beta(t)$ the Sub-VPSDE is close to the VPSDE, only slightly reducing the the variance $\sigma_t^2$ of the diffusion process distribution for small $t$. This suggests that the IS distribution derived using the regular VPSDE will likely also significantly reduce the variance of the objective due to $t$-sampling of the Sub-VPSDE, just not as optimally as theoretically possible. In Fig. 6, we show the training NELBO of an LSGM trained on CIFAR-10 with $w_{\text{ll}}$-weighting using the Sub-VPSDE. We show the NELBO both for uniform $t$ sampling as well as for $t$ sampling from the IS distribution that was originally derived for the regular VPSDE with the same $\beta(t)$ (the experi-

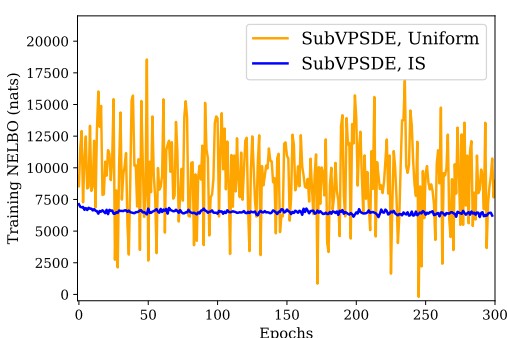

Figure 6: Variance reduction of the sample-based estimate of the training objective for the Sub-VPSDE, using an IS distribution derived from the regular VPSDE.

ment and model setup is otherwise the same as the one for the ablation study on SDEs, weighting mechanisms and variance reduction). We indeed observe a significantly reduced training objective variance. We were consequently able to train large LSGM models in a stable manner using the Sub-VPSDE with VPSDE-based IS. However, the strongest generative performance in either NLL or FID was not achieved using the Sub-VPSDE, but with the Geometric VPSDE or regular VPSDE. For that reason, we did not focus on the Sub-VPSDE in our main experiments. However, a generative modeling performance comparison of the VPSDE vs. Sub-VPSDE in a smaller LSGM model is presented in App. H.4.

## C Expressions for the Normal Transition Kernel

In our derivations of the Normal transition kernel $q(\mathbf{z}_t|\mathbf{z}_0)$, we only considered the general case in Eq. 12 and Eq. 13. However, the expression for $q(\mathbf{z}_t|\mathbf{z}_0)$ can be further simplified for different SDEs that are considered in this paper. For completeness, we provide the expressions for $q(\mathbf{z}_t|\mathbf{z}_0)$ below:

$$q(\mathbf{z}_t|\mathbf{z}_0) = \begin{cases} \mathcal{N}\left(\mathbf{z}_t; e^{-\frac{1}{2}\int_0^t\beta(s)ds}\mathbf{z}_0, \left[1-(1-\sigma_0^2)e^{-\int_0^t\beta(s)ds}\right]\mathbf{I}\right) & \text{VPSDE (linear } \beta(t)) \\ \mathcal{N}\left(\mathbf{z}_t; \sqrt{\frac{1-\sigma_{\min}^2(\frac{\sigma_{\max}^2}{\sigma_{\min}^2})^t}{1-\sigma_{\min}^2}}\mathbf{z}_0, \sigma_{\min}^2(\frac{\sigma_{\max}^2}{\sigma_{\min}^2})^t\mathbf{I}\right) & \text{Geometric VPSDE} \\ \mathcal{N}\left(\mathbf{z}_t; \mathbf{z}_0, \sigma_{\min}^2(\frac{\sigma_{\max}^2}{\sigma_{\min}^2})^t\mathbf{I}\right) & \text{VESDE} \end{cases} \quad (77)$$

In both VESDE and Geometric VPSDE, the initial variance $\sigma_0^2$ is denoted by $\sigma_{\min}^2 > 0$. These diffusion processes start from a slightly perturbed version of the data at $t = 0$. In VESDE, $\sigma_{\max}^2$ by definition is large (as the name variance exploding SDE suggests) and it is set based on the scale of the data [38]. In contrast, $\sigma_{\max}^2$ in the Geometric VPSDE does not depend on the scale of the data and it is set to $\sigma_{\max}^2 \approx 1$. In the VPSDE, the initial variance is denoted by the hyperparameter $\sigma_0^2$. In contrast to VESDE and Geometric VPSDE, we often set the initial variance to zero in VPSDE, meaning that the diffusion process models the data distribution exactly at $t = 0$. However, using the VPSDE with $\sigma_0^2 = 0$ comes at the cost of not being able to sample $t$ in the full interval $[0, 1]$ during training and also prevents us from solving the probability flow ODE all the way to zero during sampling [2].

## D  Probability Flow ODE

In LSGM, to sample from our SGM prior in latent space and to estimate NELBOs, we follow Song et al. [2] and build on the connection between SDEs and ODEs. We use black-box ODE solvers to solve the probability flow ODE. Here, we briefly recap this approach.

All SDEs used in this paper can be written in the general form

$$d\mathbf{z} = \mathbf{f}(\mathbf{z}, t)dt + g(t)d\mathbf{w}$$

The reverse of this diffusion process is also a diffusion process running backwards in time [105, 2], defined by

$$d\mathbf{z} = \left[ \mathbf{f}(\mathbf{z}, t) - g^2(t)\nabla_{\mathbf{z}_t} \log q(\mathbf{z}_t) \right] dt + g(t)d\bar{\mathbf{w}},$$

where $d\bar{\mathbf{w}}$ denotes a standard Wiener process going backwards in time, $dt$ now represents a negative infinitesimal time increment, and $\nabla_{\mathbf{z}_t} \log q(\mathbf{z}_t)$ is the score function of the diffusion process distribution at time $t$. Interestingly, Song et al. have shown that there is a corresponding ODE that generates the same marginal probability distributions $q(\mathbf{z}_t)$ when acting upon the same prior distribution $q(\mathbf{z}_1)$. It is given by

$$d\mathbf{z} = \left[ \mathbf{f}(\mathbf{z}, t) - \frac{g^2(t)}{2}\nabla_{\mathbf{z}_t} \log q(\mathbf{z}_t) \right] dt$$

and usually called the *probability flow ODE*. This connects score-based generative models using diffusion processes to continuous Normalizing flows, which are based on ODEs [73, 106]. Note that in practice $\nabla_{\mathbf{z}_t} \log q(\mathbf{z}_t)$ is approximated by a learnt model. Therefore, the generative distributions defined by the ODE and SDE above are formally not exactly equivalent when inserting this learnt model for the score function expression. Nevertheless, they often achieve quite similar performance in practice [2]. This aspect is discussed in detail in concurrent work by Song et al. [4].

We can use the above ODE for efficient sampling of the model via black-box ODE solvers. Specifically, we can draw samples from the standard Normal prior distribution at $t = 1$ and then solve this ODE towards $t = 0$. In fact, this is how we perform sampling from the latent SGM prior in our paper. Similarly, we can also use this ODE to calculate the probability of samples under this generative process using the instantaneous change of variables formula (see [73, 106] for details). We rely on this for calculating the probability of latent space samples under the score-based prior in LSGM. Note that this involves calculating the trace of the Jacobian of the ODE function. This is usually approximated via Hutchinson's trace estimator, which is unbiased but has a certain variance (also see discussion in Sec. F).

This approach is applicable similarly for all diffusion processes and SDEs considered in this paper.

## E  Converting VAE with Hierarchical Normal Prior to Standard Normal Prior

Converting a VAE with hierarchical prior to a standard Normal prior can be done using a simple change of variables. Consider a VAE with hierarchical encoder $q(\mathbf{z}|\mathbf{x}) = \prod_l q(\mathbf{z}_l|\mathbf{z}_{<l}, \mathbf{x})$ and

hierarchical prior $p(\mathbf{z}) = \prod_l p(\mathbf{z}_l|\mathbf{z}_{<l})$ where $\mathbf{z} = \{\mathbf{z}_l\}_{l=1}^L$ represent all latent variables and:

$$p(\mathbf{z}_l|\mathbf{z}_{<l}) = \mathcal{N}(\mathbf{z}_l; \boldsymbol{\mu}_l(\mathbf{z}_{<l}), \sigma_l^2(\mathbf{z}_{<l})\mathbf{I}) \tag{78}$$

$$q(\mathbf{z}_l|\mathbf{z}_{<l}, \mathbf{x}) = \mathcal{N}(\mathbf{z}_l; \boldsymbol{\mu}_l'(\mathbf{z}_{<l}, \mathbf{x}), \sigma_l'^2(\mathbf{z}_{<l}, \mathbf{x})\mathbf{I}) \tag{79}$$

where for simplicity we have assumed that the variance is shared for all the components. We can reparameterize the latent variables by introducing $\boldsymbol{\epsilon}_l = \frac{\mathbf{z}_l - \boldsymbol{\mu}_l(\mathbf{z}_{<l})}{\sigma_l(\mathbf{z}_{<l})}$. With this reparameterization, the equivalent VAE is:

$$p(\boldsymbol{\epsilon}_l) = \mathcal{N}(\boldsymbol{\epsilon}_l; \mathbf{0}, \mathbf{I}) \tag{80}$$

$$q(\boldsymbol{\epsilon}_l|\boldsymbol{\epsilon}_{<l}, \mathbf{x}) = \mathcal{N}(\boldsymbol{\epsilon}_l; \frac{\boldsymbol{\mu}_l'(\mathbf{z}_{<l}, \mathbf{x}) - \boldsymbol{\mu}_l(\mathbf{z}_1)}{\sigma_l(\mathbf{z}_{<l})}, \frac{\sigma_l'^2(\mathbf{z}_{<l}, \mathbf{x})}{\sigma_l^2(\mathbf{z}_{<l})}\mathbf{I}), \tag{81}$$

$$\tag{82}$$

where $\mathbf{z}_l = \boldsymbol{\mu}_l(\mathbf{z}_{<l}) + \sigma_l(\mathbf{z}_{<l})\boldsymbol{\epsilon}_l$. In this equivalent parameterization, we can consider $\boldsymbol{\epsilon}_l$ as latent variables with a standard Normal prior.

### E.1  Converting NVAE Prior to Standard Normal Prior

In NVAE [20], the prior has the same hierarchical form as in Eq. 78. However, the authors observe that the *residual parameterization* of the encoder often improves the generative performance. In this parameterization, with a small modification, the encoder is defined by:

$$q(\mathbf{z}_l|\mathbf{z}_{<l}, \mathbf{x}) = \mathcal{N}(\mathbf{z}_l; \boldsymbol{\mu}_l(\mathbf{z}_{<l}) + \sigma_l(\mathbf{z}_{<l})\Delta\boldsymbol{\mu}_l'(\mathbf{z}_{<l}, \mathbf{x}), \sigma_l^2(\mathbf{z}_{<l})\Delta\sigma_l'^2(\mathbf{z}_{<l}, \mathbf{x})\mathbf{I}), \tag{83}$$

where the encoder is tasked to predict the residual parameters $\Delta\boldsymbol{\mu}_l'(\mathbf{z}_{<l}, \mathbf{x})$ and $\Delta\sigma_l'^2(\mathbf{z}_{<l}, \mathbf{x})$. Using the same reparameterization as above ($\boldsymbol{\epsilon}_l = \frac{\mathbf{z}_l - \boldsymbol{\mu}_l(\mathbf{z}_{<l})}{\sigma_l(\mathbf{z}_{<l})}$), we have the equivalent VAE in the form:

$$p(\boldsymbol{\epsilon}_l) = \mathcal{N}(\boldsymbol{\epsilon}_l; \mathbf{0}, \mathbf{I}) \tag{84}$$

$$q(\boldsymbol{\epsilon}_l|\boldsymbol{\epsilon}_{<l}, \mathbf{x}) = \mathcal{N}(\boldsymbol{\epsilon}_l; \Delta\boldsymbol{\mu}_l'(\mathbf{z}_{<l}, \mathbf{x}), \Delta\sigma_l'^2(\mathbf{z}_{<l}, \mathbf{x})\mathbf{I}), \tag{85}$$

where $\mathbf{z}_l = \boldsymbol{\mu}_l(\mathbf{z}_{<l}) + \sigma_l(\mathbf{z}_{<l})\boldsymbol{\epsilon}_l$. In other words, the residual parameterization of encoder, introduced in NVAE, predicts the mean and variance for the $\boldsymbol{\epsilon}_l$ distributions directly.

## F  Bias in Importance Weighted Estimation of Log-Likelihood

A common approach for estimating test log-likelihood in VAEs is to use the importance weighted bound on log-likelihood [74]. In LSGM, we have access to an unbiased but stochastic estimation of the prior likelihood $\log p(\mathbf{z}_0)$ which we obtain using the probability flow ODE [2]. The stochasticity in the estimation comes from Hutchinson's trick [106]. In VAEs, the test log-likelihood is estimated using importance weighted (IW) estimation [74]:

$$\mathbb{E}_{\mathbf{z}^{(1)}, \ldots, \mathbf{z}^{(K)} \sim q(\mathbf{z}|\mathbf{x})}[\log(\frac{1}{K}\sum_{k=1}^K \exp(w^{(k)}))] \quad \text{where} \quad w^{(k)} = \log p(\mathbf{z}^{(k)}) + \log p(\mathbf{x}|\mathbf{z}^{(k)}) - \log q(\mathbf{z}^{(k)}|\mathbf{x}) \tag{86}$$

which is a statistical lower bound on $\log p(\mathbf{x})$.

In this section, we provide an informal analysis that shows that IW estimation with $K > 1$ can overestimate the log-likelihood when $\log p(z)$ is measured with an unbiased estimator with variance $\sigma^2$. In our analysis we assume that $\sigma^2$ is small and we use Taylor expansion to study how the IW bound varies. Under our analysis, we observe that the bias has $O(\sigma^2)$ and it can be minimized by ensuring that $\sigma^2$ is sufficiently small.

Consider the Taylor expansion around $\mathbf{w}$ up to second order of the function $\log \sum \exp(\mathbf{w}) = \log \sum_k e^{w_i}$ where $\mathbf{w} = \{w^{(k)}\}_{k=1}^K$ ($\log \sum \exp : \mathbb{R}^K \to \mathbb{R}$). With $\boldsymbol{\epsilon} \sim \mathcal{N}(\boldsymbol{\epsilon}, \mathbf{0}, \mathbf{I})$ and assuming that $\sigma^2$ is sufficiently small so that all terms beyond second order contribute negligibly, we have:

$$\mathbb{E}_{\boldsymbol{\epsilon}}[\log \sum \exp(\mathbf{w} + \sigma\boldsymbol{\epsilon})] \approx \log \sum \exp(\mathbf{w}) + \sigma\underbrace{\mathbb{E}_{\boldsymbol{\epsilon}}[\boldsymbol{\epsilon}^T]}_{\mathbf{0}}\nabla_{\mathbf{w}}\log\sum\exp(\mathbf{w}) + \sigma^2\underbrace{\mathbb{E}_{\boldsymbol{\epsilon}}[\boldsymbol{\epsilon}^T\mathbf{H}\boldsymbol{\epsilon}]}_{\text{trace}(\mathbf{H})} \tag{87}$$

where $\mathbf{H}$ is the Hessian matrix for the $\log \sum \exp$ function at $\mathbf{w}$. Note that the gradient $\nabla_{\mathbf{w}} \log \sum \exp(\mathbf{w}) = \frac{e^{w_i}}{\sum_j e^{w_j}}$ is the softmax function and $\text{trace}(\mathbf{H}) = \sum_i \frac{e^{w_i}}{\sum_j e^{w_j}} \left(1 - \frac{e^{w_i}}{\sum_j e^{w_j}}\right) \leq 1$. Thus, we have:

$$\mathbb{E}_{\boldsymbol{\epsilon}}[\log \sum \exp(\mathbf{w} + \sigma \boldsymbol{\epsilon})] \lessgtr \log \sum \exp(\mathbf{w}) + \sigma^2 \qquad (88)$$

So, when the importance weights $\mathbf{w} = \{w^{(k)}\}_{k=1}^K$ are estimated with sufficiently small variance $\sigma^2$, the bias is proportional to the variance of this estimate.

In our experiments, we observe that the variance of the $\log p(\mathbf{z}_0)$ estimate is not small enough to obtain a reliable estimate of test likelihood using the importance weighted bound. One way to reduce the variance is to use many randomly sampled noise vectors in Hutchinson's trick. However, this makes NLL estimation computationally too expensive. Fortunately, when evaluating NELBO (which corresponds to $K = 1$ here), the NELBO estimate is unbiased and its variance is small because of averaging across big test datasets (with often 10k samples). For example, on MNIST the standard deviation of our $\log p(\mathbf{z}_0)$ estimate is 0.36 nat, while the standard deviation of NELBO is 0.07 nat.

# G   Additional Implementation Details

All hyperparameters for our main models are provided in Tab. 7.

## G.1   VAE Backbone

The VAE backbone for all LSGM models is NVAE [20][7], one of the best-performing VAEs in the literature. It has a hierarchical latent space with group-wise autoregressive latent variable dependencies and it leverages residual neural networks (for architecture details see [20]). It uses depth-wise separable convolutions in the decoder. Although both the approximate posterior and the prior are hierarchical in its original version, we can reparametrize the prior and write it as a product of independent Normal distributions (see Sec. E).

The VAE's most important hyperparameters include the number of latent variable groups and their spatial resolution, the channel depth of the latent variables, the number of residual cells per group, and the number of channels in the convolutions in the residual cells. Furthermore, when training the VAE during the first stage we are using KL annealing and KL balancing, as described in [20]. For some models, we complete KL annealing during the pre-training stage, while for other models we found it beneficial to anneal only up to a KL-weight $\beta_{\text{KL}} < 1.0$ in the ELBO during the first stage and complete KL annealing during the main end-to-end LSGM training stage. This provides additional flexibility in learning an expressive distribution in latent space during the second training stage, as it prevents more latent variables from becoming inactive while the prior is being trained gradually. However, when using a very large backbone VAE together with an SGM objective that does not correspond to maximum likelihood training, i.e. $w_{\text{un}}$- or $w_{\text{re}}$-weighting, we empirically observe that this approach can also hurt NLL, while slightly improving FID (see *CIFAR10 (best FID)* model).

Note that the VAE Backbone performance for CIFAR10 reported in Tab. 2 in the main paper corresponds to the 20-group backbone VAE (trained to full KL-weight $\beta_{\text{KL}} = 1.0$) from the *CIFAR10 (balanced)* LSGM model (see hyperparameter Tab. 7).

**Image Decoders:** Since SGMs [2] assume that the data is continuous, they rely on uniform de-quantization when measuring data likelihood. However, in LSGM, we rely on decoders designed specifically for images with discrete intensity values. On color images, we use mixtures of discretized logistics [82], and on binary images, we use Bernoulli distributions. These decoder distributions are both available from the NVAE implementation.

## G.2   Latent SGM Prior

Our denoising networks for the latent SGM prior are based on the NCSN++ architecture from Song et al. [2], adapted such that the model ingests and predicts tensors according to the VAE's latent variable dimensions. We vary hyperparameters such as the number of residual cells per spatial resolution level

---

[7] https://github.com/NVlabs/NVAE (NVIDIA Source Code License)

and the number of channels in convolutions. Note that all our models use $0.2$ dropout in the SGM prior. Some of our models use upsampling and downsampling operations with anti-aliasing based on Finite Impulse Response (FIR) [107], following Song et al. [2].

NVAE has a hierarchical latent structure. For small image datasets including CIFAR-10, MNIST and OMNIGLOT all the latent variables have the same spatial dimensions. Thus, the diffusion process input $\mathbf{z}_0$ is constructed by concatenating the latent variables from all groups in the channel dimension. Our NVAE backbone on the CelebA-HQ-256 dataset comes with multiple spatial resolutions in latent groups. In this case, we only feed the smallest resolution groups to the SGM prior and assume that the remaining groups have a standard Normal distribution.

### G.3 Training Details

To optimize our models, we are mostly following the previous literature. The VAE's encoder and decoder networks are trained using an Adamax optimizer [108], following NVAE [20]. In the second stage, the whole model is trained with an Adam optimizer [108] and we perform learning rate annealing for the VAE network optimization, while we keep the learning rate constant when optimizing the SGM prior parameters. At test time, we use an exponential moving average (EMA) of the parameters of the SGM prior with $0.9999$ EMA decay rate, following [1, 2]. Note that, when using the VPSDE with linear $\beta(t)$, we are also generally following [1, 2] and use $\beta_0 = 0.1$ and $\beta_1 = 20.0$. We did not observe any benefits in using the EMA parameters for the VAE networks.

### G.4 Evaluation Details

For evaluation, we are drawing samples and calculating log-likelihoods using the probability flow ODE, leveraging black-box ODE solvers, following [73, 106, 2]. Similar to [2], we are using an RK45 ODE solver [109], based on `scipy`, using the `torchdiffeq` interface [8]. Integration cutoffs close to zero and ODE solver error tolerances used for evaluation are indicated in Tab. 7 (for example, for the VPSDE with linear $\beta(t)$ we usually use $\sigma_0^2 = 0$ and therefore have that $\sigma_t^2$ goes to 0 at $t = 0$, hence preventing us from integrating the probability flow ODE all the way to exactly 0. This was handled similarly by Song et al. [2]).

Following the conventions established by previous work [88, 3, 1, 5], when evaluating our main models we compute FID at frequent intervals during training and report FID and NLL at the minimum observed FID.

Vahdat and Kautz in NVAE [20] observe that setting the batch normalization (BN) layers to train mode during sampling (i.e., using batch statistics for normalization instead of moving average statistics) improves sample quality. We similarly observe that setting BN layers to train mode improves sample quality by about 1 FID score on the CelebA-HQ-256 dataset, but it does not affect performance on the CIFAR-10 dataset. In contrast to NVAE, we do not change the temperature of the prior during sampling, as we observe that it hurts generation quality.

### G.5 Ablation Experiments

Here we provide additional details and discussions about the ablation experiments performed in the paper.

#### G.5.1 Ablation: SDEs, Objective Weighting Mechanisms and Variance Reduction

The models that were used for the ablation experiment on SDEs, objective weighting mechanisms and variance reduction and produced the results in Tab. 6 in the main paper use an overall similar setup as the *CIFAR10 (best NLL)* one, with a few exceptions: They are trained only for 1000 epochs and evaluation always happens using the checkpoint at the end of training. Furthermore, the total batchsize over all GPUs is reduced from 256 to 128. Additionally, only 2 instead of 8 cells per residual are used in the latent SGM prior networks. Finally, the VAE's KL term is annealed all the way to $\beta_{\text{KL}} = 1.0$ during the first training stage for these experiments. All other hyperparameters correspond to the *CIFAR10 (best NLL)* setup, except those that are explicitly varied as part of the ablation study and mentioned in Tab. 6 in the paper.

---

[8] https://github.com/rtqichen/torchdiffeq (MIT License)

Table 7: Hyperparameters for our main models. We use the same notations and abbreviations as in Tab. 6 in main paper.

| Hyperparameter | CIFAR10 (best FID) | CIFAR10 (balanced) | CIFAR10 (best NLL) | CelebA-HQ-256 (best quantitative) | CelebA-HQ-256 (best qualitative) | OMNIGLOT | MNIST |
|---|---|---|---|---|---|---|---|
| **VAE Backbone** | | | | | | | |
| # normalizing flows | 0 | 0 | 2 | 2 | 2 | 0 | 0 |
| # latent variable scales | 1 | 1 | 1 | 3 | 2 | 1 | 1 |
| # groups in each scale | 20 | 20 | 4 | 8 | 10 | 3 | 2 |
| spatial dims. of $\mathbf{z}$ in each scale | $16^2$ | $16^2$ | $16^2$ | $128^2, 64^2, 32^2$ | $128^2, 64^2$ | $16^2$ | $8^2$ |
| # channel in $\mathbf{z}$ | 9 | 9 | 45 | 20 | 20 | 20 | 20 |
| # initial channels in enc. | 128 | 128 | 256 | 64 | 64 | 64 | 64 |
| # residual cells per group | 2 | 2 | 3 | 2 | 2 | 3 | 1 |
| NVAE's spectral reg. $\lambda$ | $10^{-2}$ | $10^{-2}$ | $10^{-2}$ | $3 \times 10^{-2}$ | $3 \times 10^{-2}$ | $10^{-2}$ | $10^{-2}$ |
| **Training (VAE pre-training)** | | | | | | | |
| # epochs | 400 | 600 | 400 | 200 | 200 | 200 | 200 |
| learning rate VAE | $10^{-2}$ | $10^{-2}$ | $10^{-2}$ | $10^{-2}$ | $10^{-2}$ | $10^{-2}$ | $10^{-2}$ |
| batch size per GPU | 32 | 32 | 64 | 4 | 4 | 64 | 100 |
| # GPUs | 8 | 8 | 4 | 16 | 16 | 2 | 2 |
| KL annealing to | $\beta_{\mathrm{KL}}{=}0.7$ | $\beta_{\mathrm{KL}}{=}1.0$ | $\beta_{\mathrm{KL}}{=}0.7$ | $\beta_{\mathrm{KL}}{=}1.0$ | $\beta_{\mathrm{KL}}{=}1.0$ | $\beta_{\mathrm{KL}}{=}1.0$ | $\beta_{\mathrm{KL}}{=}0.7$ |
| **Latent SGM Prior** | | | | | | | |
| # number of scales | 3 | 3 | 3 | 4 | 5 | 3 | 2 |
| # residual cells per scale | 8 | 8 | 8 | 8 | 8 | 8 | 8 |
| # conv. channels at each scale | $[512]\times3$ | $[512]\times3$ | $[512]\times3$ | $256, [512]\times3$ | $[320]\times2, [640]\times3$ | $[256]\times3$ | $[256]\times2$ |
| use FIR [107] | yes | yes | yes | yes | yes | no | no |
| **Training (Main LSGM training)** | | | | | | | |
| # epochs | 1875 | 1875 | 1875 | 1000 | 2000 | 1500 | 800 |
| learning rate VAE | $10^{-4}$ | $10^{-4}$ | $10^{-4}$ | $10^{-4}$ | - | $10^{-4}$ | $10^{-4}$ |
| learning rate SGM prior | $10^{-4}$ | $10^{-4}$ | $10^{-4}$ | $10^{-4}$ | $10^{-4}$ | $3 \times 10^{-4}$ | $3 \times 10^{-4}$ |
| batch size per GPU | 16 | 16 | 16 | 4 | 8 | 32 | 32 |
| # GPUs | 16 | 16 | 16 | 16 | 16 | 4 | 4 |
| KL annealing | continued | no | continued | no | no | continued | continued |
| SDE | VPSDE | VPSDE | Geo. VPSDE | VPSDE | VPSDE | VPSDE | VPSDE |
| $\sigma_0^2$ ($= \sigma_{\min}^2$ for Geo. VPSDE) | 0.0 | 0.0 | $3 \times 10^{-5}$ | 0.0 | 0.0 | 0.0 | 0.0 |
| $\sigma_{\max}^2$ (only for Geo. VPSDE) | - | - | 0.999 | - | - | - | - |
| $t$-sampling cutoff during training | 0.01 | 0.01 | 0.0 | 0.01 | 0.01 | 0.01 | 0.01 |
| SGM prior weighting mechanism | $w_{\mathrm{un}}$ | $w_{\mathrm{un}}$ | $w_{\mathrm{ll}}$ | $w_{\mathrm{re}}$ | $w_{\mathrm{re}}$ | $w_{\mathrm{ll}}$ | $w_{\mathrm{ll}}$ |
| t-sampling approach (SGM-obj.) | $r_{\mathrm{un}}(t)$ | $r_{\mathrm{un}}(t)$ | $\mathcal{U}[0,1]$ | $r_{\mathrm{re}}(t)$ | $r_{\mathrm{re}}(t)$ | $r_{\mathrm{ll}}(t)$ | $r_{\mathrm{ll}}(t)$ |
| t-sampling approach (q-obj.) | *rew.* | *rew.* | *rew.* | $r_{\mathrm{ll}}(t)$ | - | *rew.* | *rew.* |
| **Evaluation** | | | | | | | |
| ODE solver integration cutoff | $10^{-6}$ | $10^{-6}$ | $10^{-6}$ | $10^{-5}$ | $10^{-5}$ | $10^{-5}$ | $10^{-5}$ |
| ODE solver error tolerance | $10^{-5}$ | $10^{-5}$ | $10^{-5}$ | $10^{-5}$ | $10^{-5}$ | $10^{-5}$ | $10^{-5}$ |

As discussed in the main paper, the results of this ablation study overall validate that importance sampling is important to stabilize training, that the $w_{\mathrm{ll}}$-weighting mechanism as well as our novel geometric VPSDE are well suited for training towards strong likelihood, and that the $w_{\mathrm{un}}$- and $w_{\mathrm{re}}$-weighting mechanisms tend to produce better FIDs. Although these trends generally hold, it is noteworthy that not all results translate perfectly to our large models that we used to produce our main results. For instance, the setting with $w_{\mathrm{re}}$-weighting and no importance sampling for the SGM objective, which produced the best FID in Tab. 6 (main paper), is generally unstable for our bigger models, in line with our observation that IS is usually necessary to stabilize training. The stable training run for this setting in Tab. 6 can be considered an outlier.

Furthermore, for CIFAR10 we obtained our very best FID results using the VPSDE, $w_{\mathrm{un}}$-weighting, IS, and sample reweighting for the $q$-objective, while for the slightly smaller models used for the results in Tab. 6, there is no difference between using sample reweighting and drawing a separate batch $t$ with $r_{\mathrm{ll}}(t)$ for training $q$ for this case (see Tab. 6 main paper, VPSDE, $w_{\mathrm{un}}$, $r_{\mathrm{un}}(t)$ fields). Also, CelebA-HQ-256 behaves slightly different for the large models in that the VPSDE with $w_{\mathrm{re}}$-weighting and sampling a separate batch $t$ with $r_{\mathrm{ll}}(t)$ for $q$-training performed best by a small margin (see hyperparameter Tab. 7).

### G.5.2 Ablation: End-to-End Training

The model used for the results on the ablation study regarding end-to-end training vs. fully separate VAE and SGM prior training is the same one as used for the ablation study on SDEs, objective weighting mechanisms and variance reduction above, evaluated in a similar way. For this experiment, we used the VPSDE, $w_{\mathrm{un}}$-objective weighting, IS for $t$ with $r_{\mathrm{un}}(t)$ when training the SGM prior, and we did draw a second batch $t$ with $r_{\mathrm{ll}}(t)$ for training $q$ (only relevant for the end-to-end training setup).

### G.5.3 Ablation: Mixing Normal and Neural Score Functions

The model used for the ablation study on mixing Normal and neural score functions is again similar to the one used for the other ablations with the exception that the underlying VAE has only a single latent variable group, which makes it much smaller and removes all hierarchical dependencies between latent variables. We tried training multiple models with larger backbone VAEs, but they were generally unstable when trained without our mixed score parametrization, which only highlights its importance. As for the previous ablation, for this experiment we used the VPSDE, $w_{\mathrm{un}}$-objective weighting, IS for $t$ with $r_{\mathrm{un}}(t)$ when training the SGM prior, and we did draw a second batch $t$ with $r_{\mathrm{ll}}(t)$ for training $q$.

### G.6 Training Algorithms

To unambiguously clarify how we train our LSGMs, we summarized the training procedures in three different algorithms for different situations:

1. *Likelihood training with IS.* In this case, the SGM prior and the encoder share the same weighted likelihood objective and do not need to be updated separately.

2. *Un/Reweighted training with separate IS of $t$ for SGM-objective and $q$-objective.* Here, the SGM prior and the encoder need to be updated with different weightings, because the encoder always needs to be trained using the weighted (maximum likelihood) objective. We draw separate batches $t$ using separate IS distribution for the two differently weighted objectives (i.e. last term in Eq. 8 from main paper vs. Eq. 9).

3. *Un/Reweighted training with IS of $t$ for the SGM-objective and reweighting for the $q$-objective.* What this means is that when training the encoder with the score-based cross entropy term (last term in Eq. 8 from main paper), we are using an importance sampling distribution that was actually tailored to un- or reweighted training for the SGM objective (Eq. 9 from main paper) and therefore isn't optimal for the weighted (maximum likelihood) objective necessary for encoder training. However, if we nevertheless use the same importance sampling distribution, we do not need to draw a second batch of $t$ for encoder training. In practice, this boils down to different (re-)weighting factors in the cross entropy term (see Algorithm 3).

For efficiency comparison between approaches (2) and (3), we observe that (3) consumes more memory than (2) in general but it can be faster due to the shared computation for the denoising step. Due to the memory limitations, we use (2) on large image datasets. Note that the choice between (2) and (3) may affect generative performance as we empirically observed in our experiments.

---

**Algorithm 1** Likelihood training with IS

---

**Input:** data $\mathbf{x}$, parameters $\{\boldsymbol{\theta}, \boldsymbol{\phi}, \boldsymbol{\psi}\}$
Draw $\mathbf{z}_0 \sim q_{\boldsymbol{\phi}}(\mathbf{z}_0|\mathbf{x})$ using encoder.
Draw $t \sim r_{\mathrm{ll}}(t)$ with IS distribution of likelihood weighting (Sec. B).
Calculate $\boldsymbol{\mu}_t(\mathbf{z}_0)$ and $\sigma_t^2$ according to SDE.
Draw $\mathbf{z}_t \sim q(\mathbf{z}_t|\mathbf{z}_0)$ using $\mathbf{z}_t = \boldsymbol{\mu}_t(\mathbf{z}_0) + \sigma_t^2\boldsymbol{\epsilon}$ where $\boldsymbol{\epsilon} \sim \mathcal{N}(\boldsymbol{\epsilon}, \mathbf{0}, \mathbf{I})$.
Calculate score $\boldsymbol{\epsilon}_{\boldsymbol{\theta}}(\mathbf{z}_t, t) = \sigma_t(1 - \boldsymbol{\alpha}) \odot \mathbf{z}_t + \boldsymbol{\alpha} \odot \boldsymbol{\epsilon}'_{\boldsymbol{\theta}}(\mathbf{z}_t, t)$.
Calculate cross entropy $\mathrm{CE}(q_{\boldsymbol{\phi}}(\mathbf{z}_0|\mathbf{x})||p_{\boldsymbol{\theta}}(\mathbf{z}_0)) \approx \frac{1}{r_{\mathrm{ll}}(t)}\frac{w_{\mathrm{ll}}(t)}{2}||\boldsymbol{\epsilon} - \boldsymbol{\epsilon}_{\boldsymbol{\theta}}(\mathbf{z}_t, t)||_2^2$.
Calculate objective $\mathcal{L}(\mathbf{x}, \boldsymbol{\theta}, \boldsymbol{\phi}, \boldsymbol{\psi}) = -\log p_{\boldsymbol{\psi}}(\mathbf{x}|\mathbf{z}_0) + \log q_{\boldsymbol{\phi}}(\mathbf{z}_0|\mathbf{x}) + \mathrm{CE}(q_{\boldsymbol{\phi}}(\mathbf{z}_0|\mathbf{x})||p_{\boldsymbol{\theta}}(\mathbf{z}_0))$.
Update all parameters $\{\boldsymbol{\theta}, \boldsymbol{\phi}, \boldsymbol{\psi}\}$ by minimizing $\mathcal{L}(\mathbf{x}, \boldsymbol{\theta}, \boldsymbol{\phi}, \boldsymbol{\psi})$.

---

**Algorithm 2** Un/Reweighted training with separate IS of $t$

---

**Input:** data $\mathbf{x}$, parameters $\{\boldsymbol{\theta}, \boldsymbol{\phi}, \boldsymbol{\psi}\}$
Draw $\mathbf{z}_0 \sim q_{\boldsymbol{\phi}}(\mathbf{z}_0|\mathbf{x})$ using encoder.

$\triangleright$ `Update SGM prior`
Draw $t \sim r_{\mathrm{un/re}}(t)$ with IS distribution for un/reweighted objective (Sec. B).
Calculate $\boldsymbol{\mu}_t(\mathbf{z}_0)$ and $\sigma_t^2$ according to SDE.
Draw $\mathbf{z}_t \sim q(\mathbf{z}_t|\mathbf{z}_0)$ using $\mathbf{z}_t = \boldsymbol{\mu}_t(\mathbf{z}_0) + \sigma_t^2 \boldsymbol{\epsilon}$ where $\boldsymbol{\epsilon} \sim \mathcal{N}(\boldsymbol{\epsilon}, \mathbf{0}, \mathbf{I})$.
Calculate score $\boldsymbol{\epsilon}_{\boldsymbol{\theta}}(\mathbf{z}_t, t) = \sigma_t(1 - \boldsymbol{\alpha}) \odot \mathbf{z}_t + \boldsymbol{\alpha} \odot \boldsymbol{\epsilon}'_{\boldsymbol{\theta}}(\mathbf{z}_t, t)$.
Calculate objective $\mathcal{L}(\boldsymbol{\theta}) \approx \frac{1}{r_{\mathrm{un/re}}(t)} \frac{w_{\mathrm{un/re}}(t)}{2} ||\boldsymbol{\epsilon} - \boldsymbol{\epsilon}_{\boldsymbol{\theta}}(\mathbf{z}_t, t)||_2^2$.
Update SGM prior parameters $\boldsymbol{\theta}$ by minimizing $\mathcal{L}(\boldsymbol{\theta})$.

$\triangleright$ `Update VAE Encoder and Decoder with new` $t$ `sample`
Draw $t \sim r_{\mathrm{ll}}(t)$ with IS distribution for likelihood weighting (Sec. B).
Calculate $\boldsymbol{\mu}_t(\mathbf{z}_0)$ and $\sigma_t^2$ according to SDE.
Draw $\mathbf{z}_t \sim q(\mathbf{z}_t|\mathbf{z}_0)$ using $\mathbf{z}_t = \boldsymbol{\mu}_t(\mathbf{z}_0) + \sigma_t^2 \boldsymbol{\epsilon}$ where $\boldsymbol{\epsilon} \sim \mathcal{N}(\boldsymbol{\epsilon}, \mathbf{0}, \mathbf{I})$.
Calculate score $\boldsymbol{\epsilon}_{\boldsymbol{\theta}}(\mathbf{z}_t, t) = \sigma_t(1 - \boldsymbol{\alpha}) \odot \mathbf{z}_t + \boldsymbol{\alpha} \odot \boldsymbol{\epsilon}'_{\boldsymbol{\theta}}(\mathbf{z}_t, t)$.
Calculate cross entropy $\mathrm{CE}(q_{\boldsymbol{\phi}}(\mathbf{z}_0|\mathbf{x})||p_{\boldsymbol{\theta}}(\mathbf{z}_0)) \approx \frac{1}{r_{\mathrm{ll}}(t)} \frac{w_{\mathrm{ll}}(t)}{2} ||\boldsymbol{\epsilon} - \boldsymbol{\epsilon}_{\boldsymbol{\theta}}(\mathbf{z}_t, t)||_2^2$.
Calculate objective $\mathcal{L}(\mathbf{x}, \boldsymbol{\phi}, \boldsymbol{\psi}) = -\log p_{\boldsymbol{\psi}}(\mathbf{x}|\mathbf{z}_0) + \log q_{\boldsymbol{\phi}}(\mathbf{z}_0|\mathbf{x}) + \mathrm{CE}(q_{\boldsymbol{\phi}}(\mathbf{z}_0|\mathbf{x})||p_{\boldsymbol{\theta}}(\mathbf{z}_0))$.
Update VAE parameters $\{\boldsymbol{\phi}, \boldsymbol{\psi}\}$ by minimizing $\mathcal{L}(\mathbf{x}, \boldsymbol{\phi}, \boldsymbol{\psi})$.

---

**Algorithm 3** Un/Reweighted training with IS of $t$ for the SGM objective

---

**Input:** data $\mathbf{x}$, parameters $\{\boldsymbol{\theta}, \boldsymbol{\phi}, \boldsymbol{\psi}\}$
Draw $\mathbf{z}_0 \sim q_{\boldsymbol{\phi}}(\mathbf{z}_0|\mathbf{x})$ using encoder.
Draw $t \sim r_{\mathrm{un/re}}(t)$ with IS distribution for un/reweighted objective (Sec. B).
Calculate $\boldsymbol{\mu}_t(\mathbf{z}_0)$ and $\sigma_t^2$ according to SDE.
Draw $\mathbf{z}_t \sim q(\mathbf{z}_t|\mathbf{z}_0)$ using $\mathbf{z}_t = \boldsymbol{\mu}_t(\mathbf{z}_0) + \sigma_t^2 \boldsymbol{\epsilon}$ where $\boldsymbol{\epsilon} \sim \mathcal{N}(\boldsymbol{\epsilon}, \mathbf{0}, \mathbf{I})$.
Calculate score $\boldsymbol{\epsilon}_{\boldsymbol{\theta}}(\mathbf{z}_t, t) = \sigma_t(1 - \boldsymbol{\alpha}) \odot \mathbf{z}_t + \boldsymbol{\alpha} \odot \boldsymbol{\epsilon}'_{\boldsymbol{\theta}}(\mathbf{z}_t, t)$.
Compute $\mathcal{L}_{DSM} := ||\boldsymbol{\epsilon} - \boldsymbol{\epsilon}_{\boldsymbol{\theta}}(\mathbf{z}_t, t)||_2^2$

$\triangleright$ `SGM prior loss`
Calculate objective $\mathcal{L}(\boldsymbol{\theta}) \approx \frac{1}{r_{\mathrm{un/re}}(t)} \frac{w_{\mathrm{un/re}}(t)}{2} \mathcal{L}_{DSM}$.

$\triangleright$ `VAE Encoder and Decoder loss computed with the same` $t$ `sample`
Calculate cross entropy $\mathrm{CE}(q_{\boldsymbol{\phi}}(\mathbf{z}_0|\mathbf{x})||p_{\boldsymbol{\theta}}(\mathbf{z}_0)) \approx \frac{1}{r_{\mathrm{un/re}}(t)} \frac{w_{\mathrm{ll}}(t)}{2} \mathcal{L}_{DSM}$.
Calculate objective $\mathcal{L}(\mathbf{x}, \boldsymbol{\phi}, \boldsymbol{\psi}) = -\log p_{\boldsymbol{\psi}}(\mathbf{x}|\mathbf{z}_0) + \log q_{\boldsymbol{\phi}}(\mathbf{z}_0|\mathbf{x}) + \mathrm{CE}(q_{\boldsymbol{\phi}}(\mathbf{z}_0|\mathbf{x})||p_{\boldsymbol{\theta}}(\mathbf{z}_0))$.

$\triangleright$ `Update all parameters`
Update SGM prior parameters $\boldsymbol{\theta}$ by minimizing $\mathcal{L}(\boldsymbol{\theta})$.
Update VAE parameters $\{\boldsymbol{\phi}, \boldsymbol{\psi}\}$ by minimizing $\mathcal{L}(\mathbf{x}, \boldsymbol{\phi}, \boldsymbol{\psi})$.

---

### G.7 Computational Resources

In total, the research project consumed $\approx 350,000$ GPU hours, which translates to an electricity consumption of about $\approx 50$ MWh. We used an in-house GPU cluster of V100 NVIDIA GPUs.

## H Additional Experiments

### H.1 Additional Samples

In this section, we provide additional samples generated by our models for CIFAR-10 in Fig. 7, and CelebA-256-HQ in Fig. 8.

Table 8: Experiment with a small VAE architecture on dynamically binarized MNIST.

| Method | NELBO ↓ (nats) |
|---|---|
| Small VAE [24] | 84.08±0.10 |
| Small VAE + inverse autoregressive flow [24] | 80.80±0.07 |
| Our small VAE | 83.85 |
| Our LSGM w/ small VAE | **79.23** |

Table 9: Number of function evaluations (NFE) of ODE solver during probability flow-based latent SGM prior sampling and corresponding sampling time for our main CIFAR-10 models. Sampling was done in batches of size 16 using a single Titan V GPU. Results are averaged over 20 sampling runs. See Tab. 2 in main text for generative performance metrics.

| Method | NFE ↓ | Sampling Time ↓ |
|---|---|---|
| LSGM (FID) | 138 | 11.07 sec. |
| LSGM (NLL) | 120 | 9.58 sec. |
| LSGM (balanced) | 128 | 10.26 sec. |

## H.2 MNIST: Small VAE Experiment

Here, we examine our LSGM on a small VAE architecture. We specifically follow [24] and build a small VAE in the NVAE codebase. In particular, the model does not have hierarchical latent variables, but only a single latent variable group with a total of 64 latent variables. Encoder and decoder consist of small ResNets with 6 residual cells in total (every two cells there is a down- or up-sampling operation, so we have 3 blocks with 2 residual cells per block). The experiments are done on dynamically binarized MNIST. As we can see in Table 8, our implementation of the VAE obtains a similar test NELBO as [24]. However, our LSGM improves the NELBO by almost 4.6 nats. This simple experiment shows that we can even obtain good generative performance with our LSGM using small VAE architectures.

## H.3 CIFAR-10: Neural Network Evaluations during Sampling

In Tab. 9, we report the number of neural network evaluations performed by the ODE solver during sampling from our CIFAR-10 models. ODE solver error tolerance is $10^{-5}$ and time integration cutoff is $10^{-6}$. CIFAR-10 is a highly diverse and more multimodal dataset, compared to CelebA-HQ-256. Because of that, the latent SGM prior that is learnt is more complex, requiring more function evaluations.

## H.4 CIFAR-10: Sub-VPSDE vs. VPSDE

In App. B.5 we discussed how variance reduction techniques derived based on the VPSDE can also help reducing the variance of the sample-based estimate of the training objective when using the Sub-VPSDE in the latent space SGM. Here, we perform a quantitative comparison between the VPSDE and the Sub-VPSDE, following the same experimental setup and using the same models as for the ablation study on SDEs, objective weighting mechanisms, and variance reduction (experiment details in App. G.5.1). The results are reported in Tab. 10. We find that the VPSDE generally performs slightly better in FID, while we observed little difference in NELBO in these experiments. Importantly, the Sub-VPSDE also did not outperform our novel geometric VPSDE in NELBO. We also see that the combination of Sub-VPSDE with $w_{re}$-weighting performs poorly. Consequently, we did not explore the Sub-VPSDE further in our main experiments.

## H.5 CelebA-HQ-256: Different ODE Solver Error Tolerances

In Fig. 9, we visualize CelebA-HQ-256 samples from our LSGM model for varying ODE solver error tolerances.

Table 10: Comparing the VPSDE and Sub-VPSDE in LSGM. For detailed explanations of abbreviations in the table, see Tab. 6 in main paper. Note that importance sampling distributions are generally based on derivations with the VPSDE, even when using the Sub-VPSDE, as discussed in App. B.5.

| SGM-obj.-weighting | | $w_{ll}$ | $w_{un}$ | $w_{re}$ |
|---|---|---|---|---|
| $t$-sampling (SGM-obj.) | | $r_{ll}(t)$ | $r_{un}(t)$ | $r_{re}(t)$ |
| $t$-sampling (q-obj.) | | *rew.* | $r_{ll}(t)$ | $r_{ll}(t)$ |
| **VPSDE** | **FID**↓ | 8.00 | 5.39 | 6.19 |
| | **NELBO**↓ | 2.97 | 2.98 | 2.99 |
| **Sub-VPSDE** | **FID**↓ | 8.46 | 5.73 | 19.10 |
| | **NELBO**↓ | 2.97 | 2.97 | 3.04 |

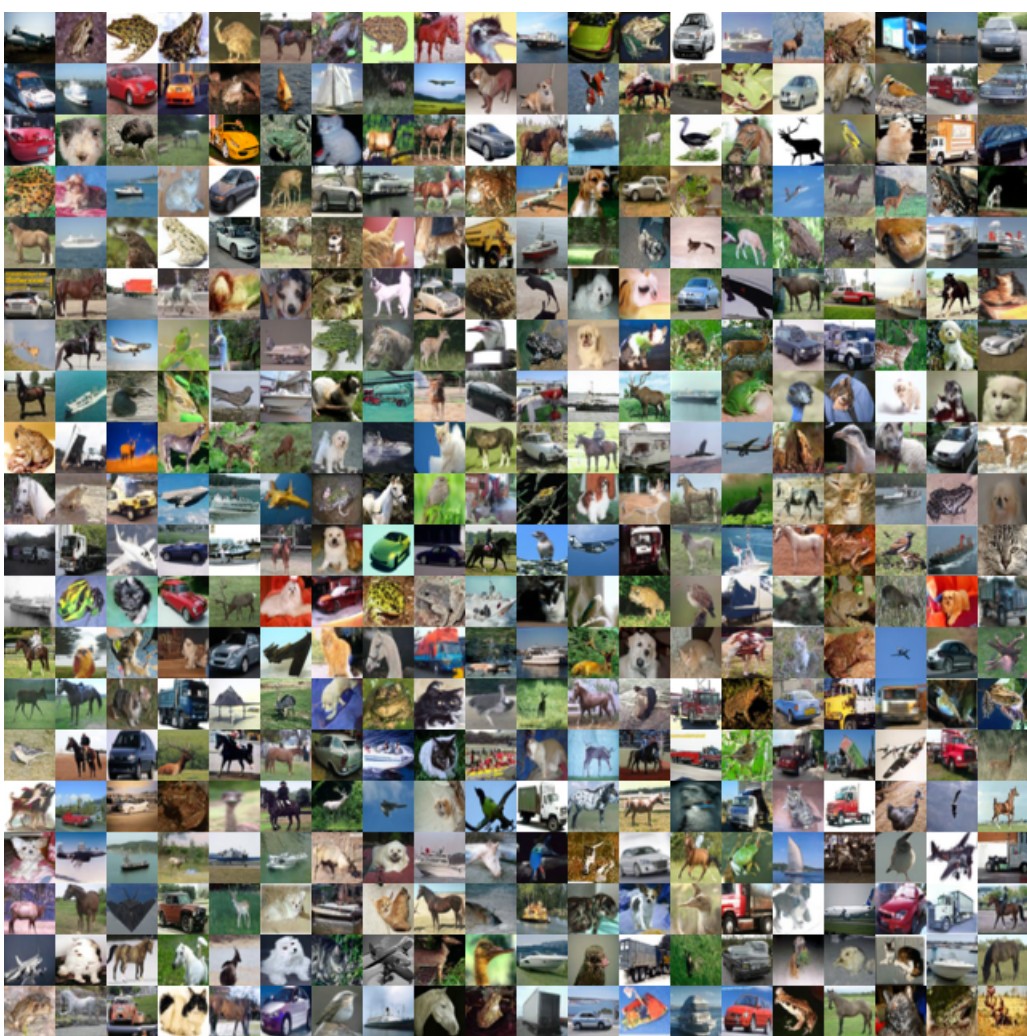

Figure 7: Additional uncurated samples generated by LSGM on the CIFAR-10 dataset (best FID model). Sampling in the latent space is done using the probability flow ODE.

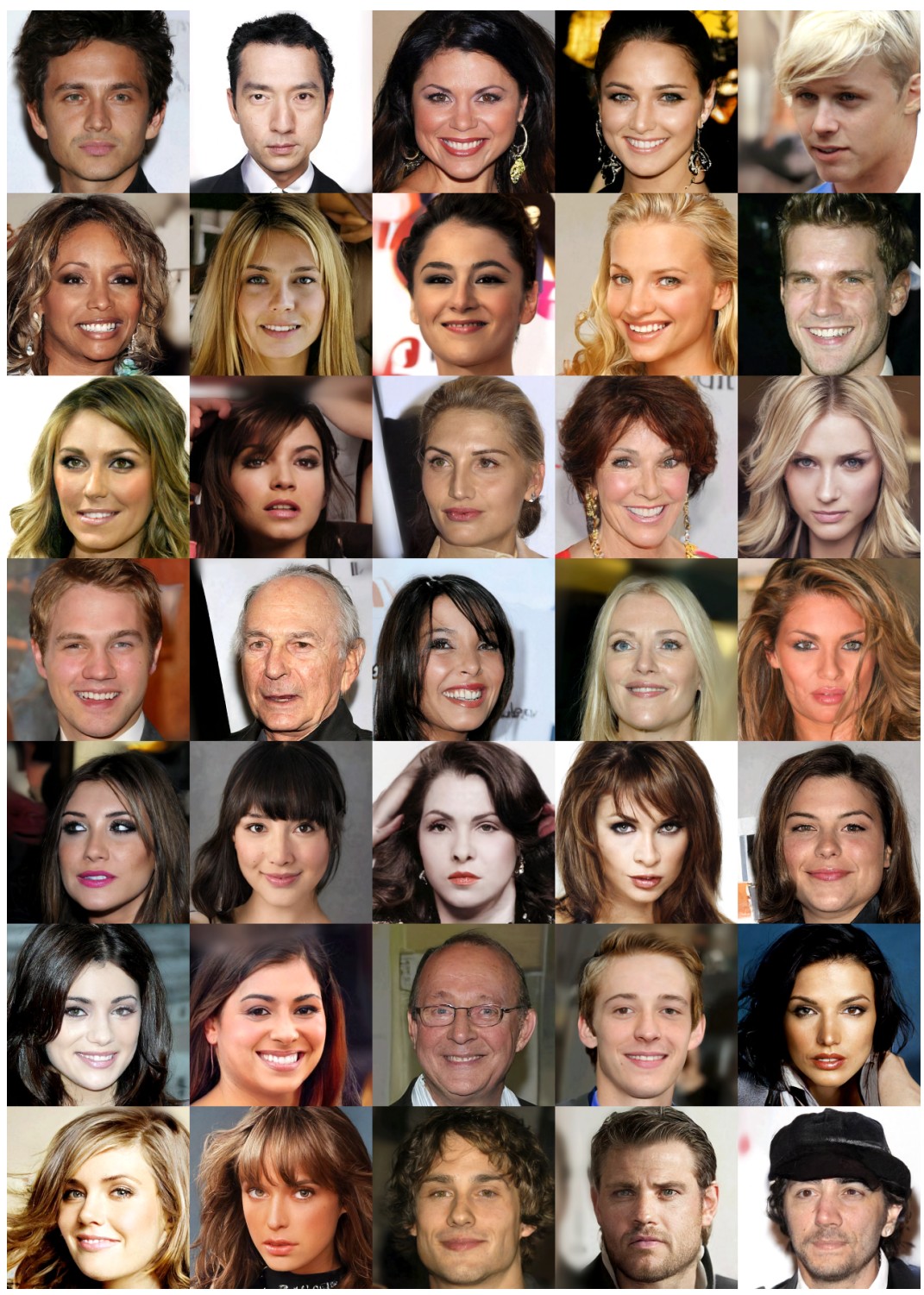

Figure 8: Additional uncurated samples generated by LSGM on the CelebA-HQ-256 dataset. Sampling in the latent space is done using the probability flow ODE.

### H.6 CelebA-HQ-256: Ancestral Sampling

For our experiments in this paper, we use the probability flow ODE to sample from the model. However, on CelebA-HQ-256, we observe that ancestral sampling [2, 1, 27] from the prior instead of solving the probability flow ODE often generates much higher quality samples. However, the FID score is slightly worse for this approach. In Fig. 10, Fig. 11, and Fig. 12, we visualize samples generated with different numbers of steps in ancestral sampling.

### H.7 CelebA-HQ-256: Sampling from VAE Backbone vs. LSGM

For the quantitative results on the CelebA-HQ-256 dataset in the main text, we use an LSGM with spatial dimension of 32×32 for the latent variables in the SGM prior. However, for the qualitative results we used an LSGM with the prior spatial dimension of 64×64. The 32×32 dimensional model achieves a better FID score compared to the 64×64 dimensional model (FID 7.22 vs. 8.53) and sampling from it is much faster (2.7 sec. vs. 39.9 sec.). However, the visual quality of the samples is slightly worse. In this section, we visualize samples generated by the 32×32 dimensional model as well as the VAE backbone for this model. In this experiment, the VAE backbone is fully trained. Samples from our VAE backbone are visualized in Fig. 13 and for our 32×32 dimensional LSGM in Fig. 14.

### H.8 Evolution Samples on the ODE and SDE Reverse Generative Process

In Fig. 15, we visualize the evolution of the latent variables under both the reverse generative SDE and also the probability flow ODE. We are decoding the intermediate latent samples along the reverse-time generative process via the decoder to pixel space.

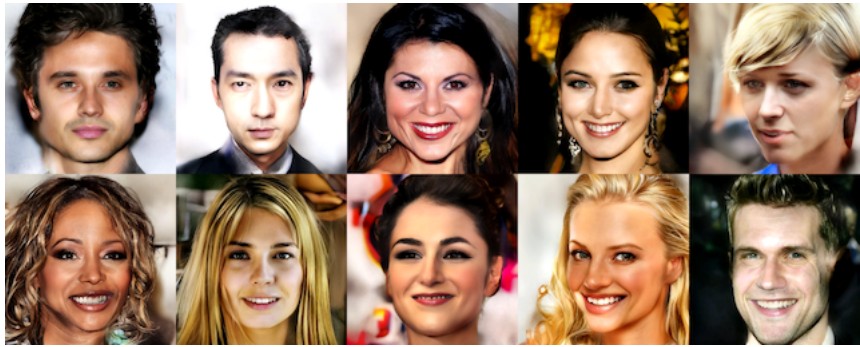

(a) ODE solver error tolerance $10^{-2}$

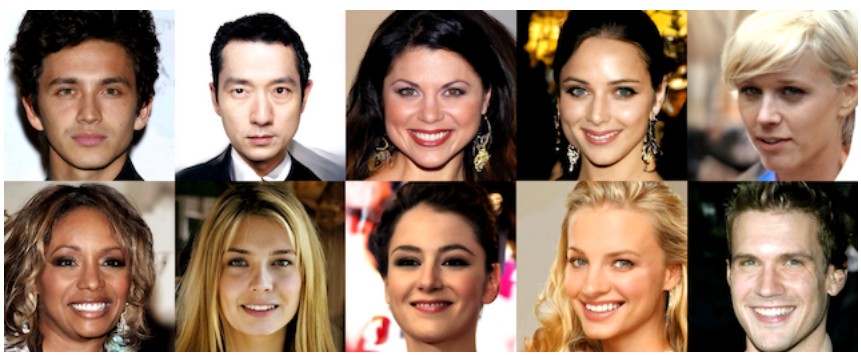

(b) ODE solver error tolerance $10^{-3}$

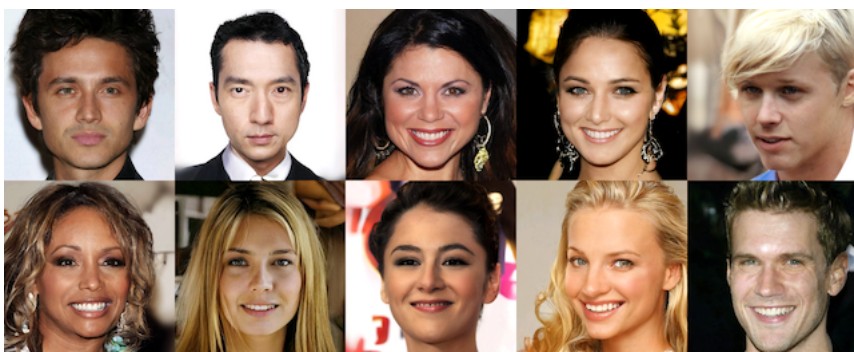

(c) ODE solver error tolerance $10^{-4}$

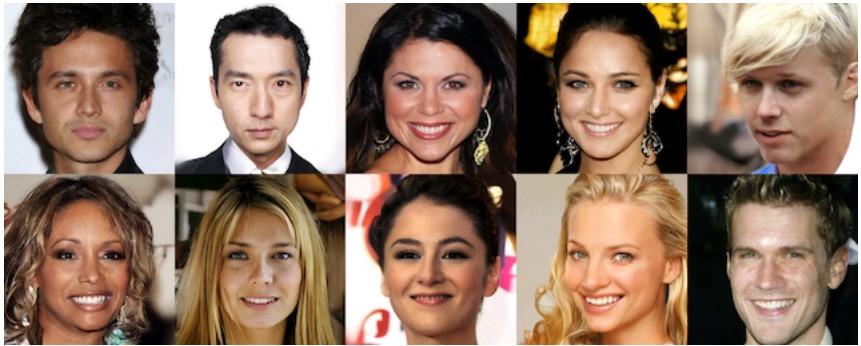

(d) ODE solver error tolerance $10^{-5}$

Figure 9: The effect of ODE solver error tolerance on the quality of samples. In contrast to the original SGM [2] where high error tolerance results in pixelated images (see Fig. 3 in [2]), in our case high error tolerances create low-frequency artifacts. Reducing the error tolerance improves subtle details slightly.

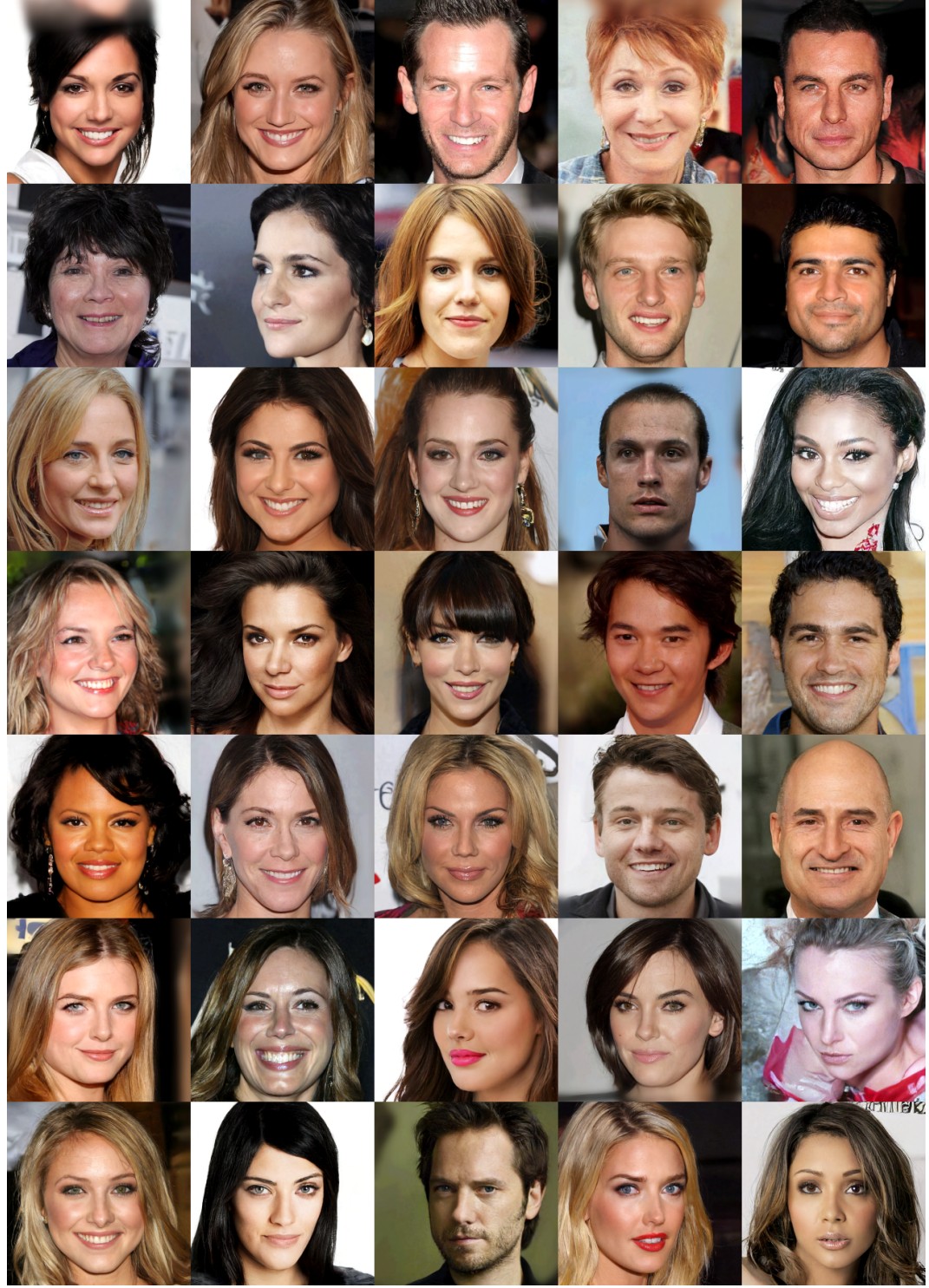

Figure 10: Uncurated samples generated by LSGM on the CelebA-HQ-256 dataset using 200-step ancestral sampling for the prior.

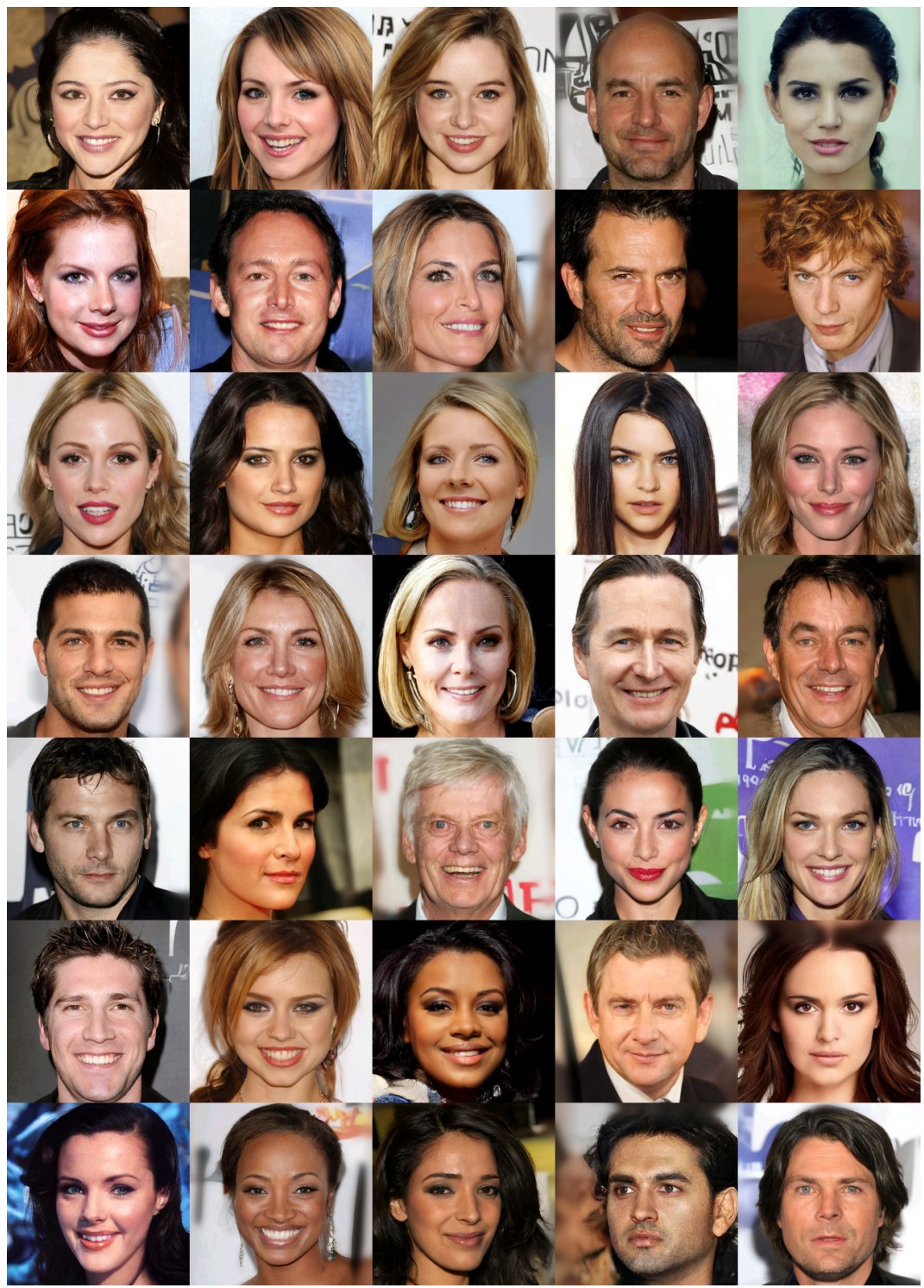

Figure 11: Uncurated samples generated by LSGM on the CelebA-HQ-256 dataset using 1000-step ancestral sampling for the prior.

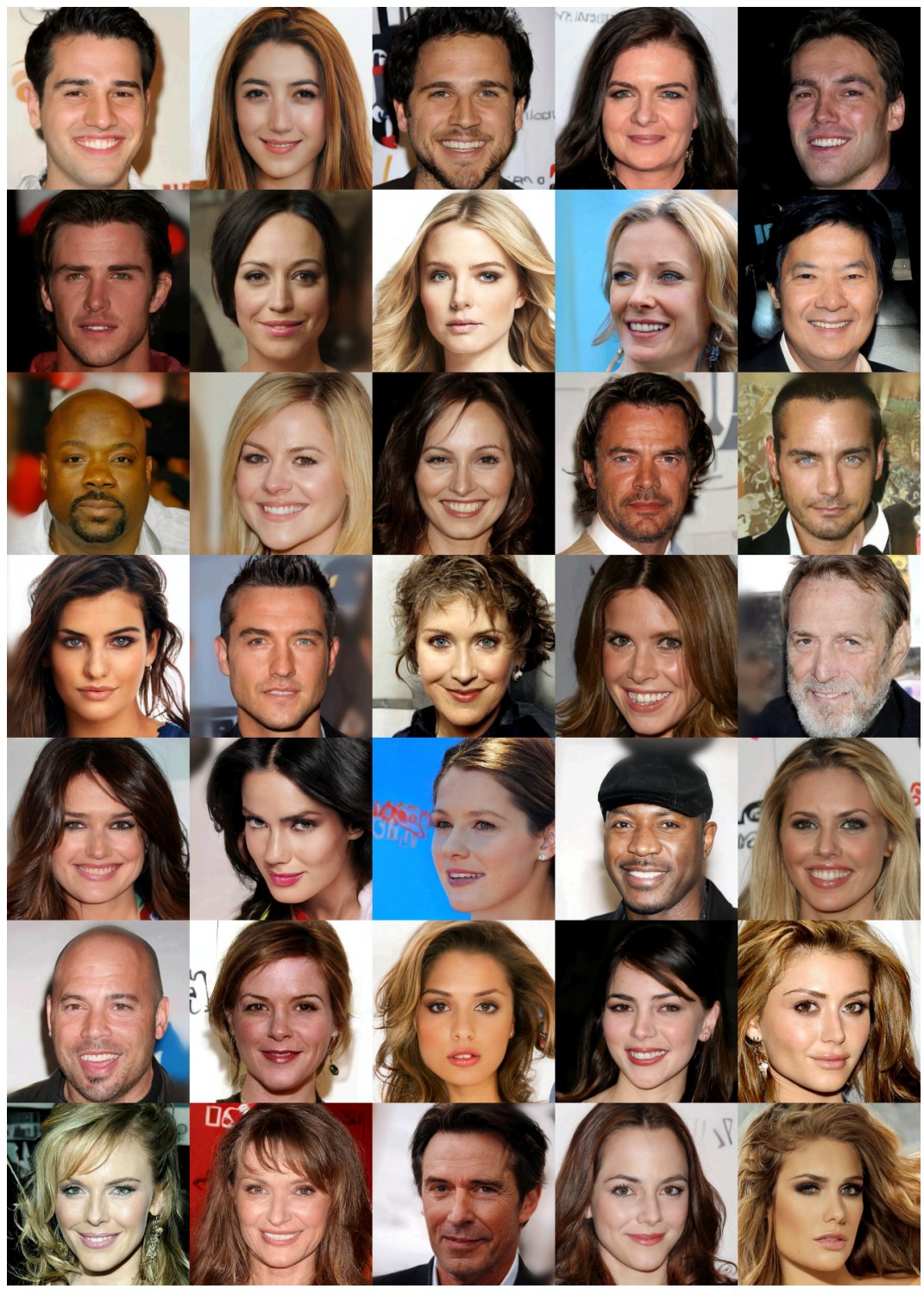

Figure 12: Additional uncurated samples generated by LSGM on the CelebA-HQ-256 dataset using 1000-step ancestral sampling.

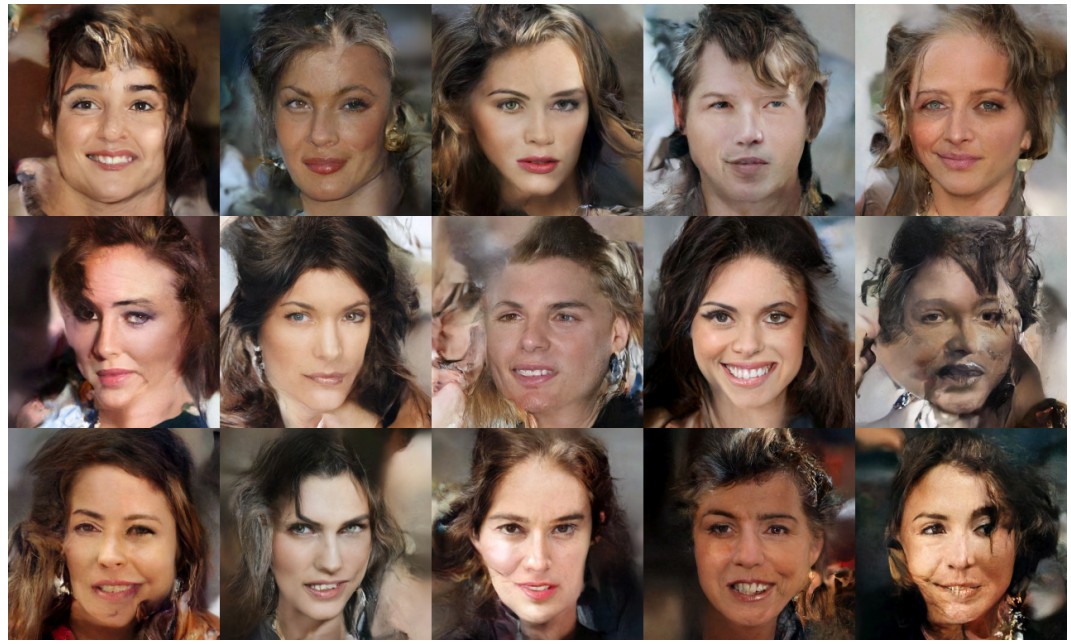

Figure 13: Uncurated samples generated by our VAE backbone without changing the temperature of the prior. The poor quality of the samples from the VAE backbone is partially due to the large spatial dimensions of the latent space in which long-range correlations are not encoded well.

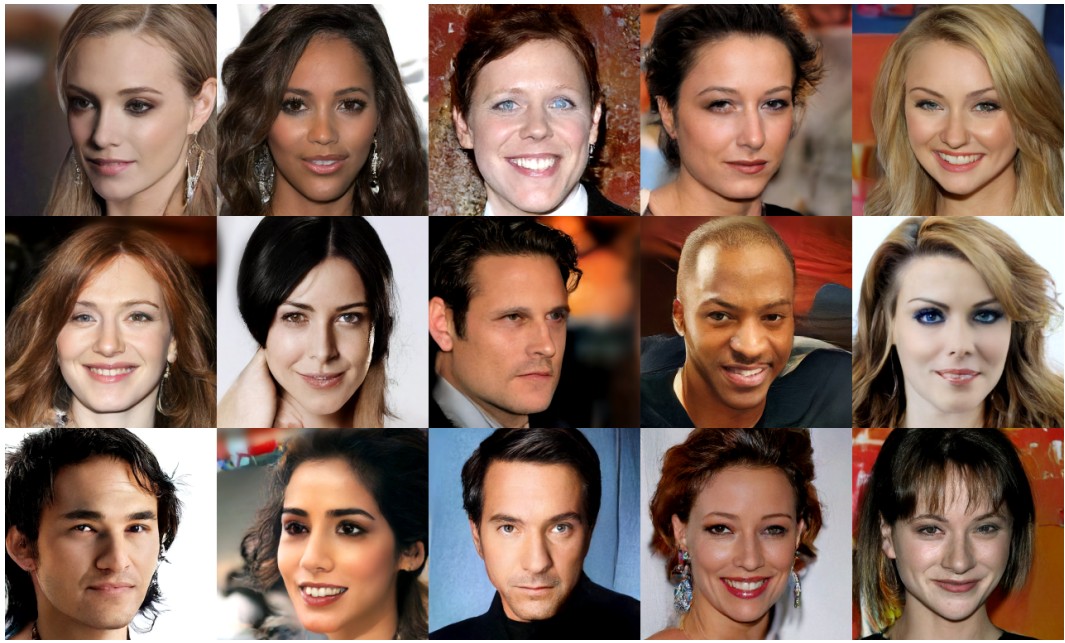

Figure 14: Uncurated samples generated by LSGM with the SGM prior applied to the latent variables of 32×32 spatial dimensions, on the CelebA-HQ-256 dataset. Sampling in the latent space is done using the probability flow ODE.

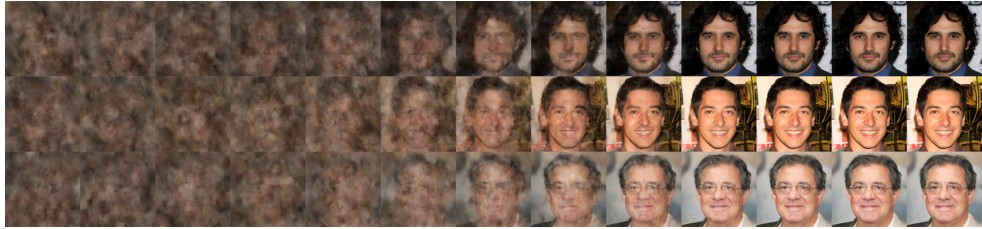

(a) Evolution of latent variables under the SDE

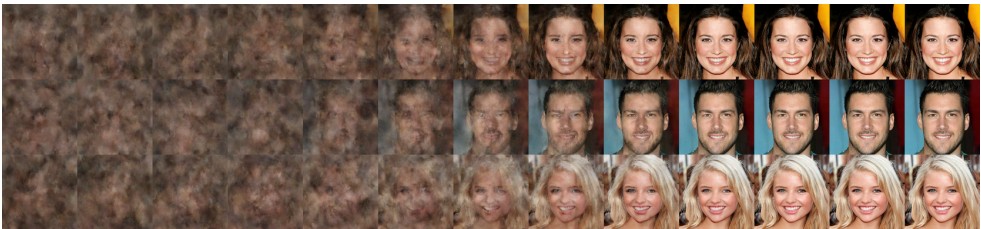

(b) Evolution of latent variables under the SDE

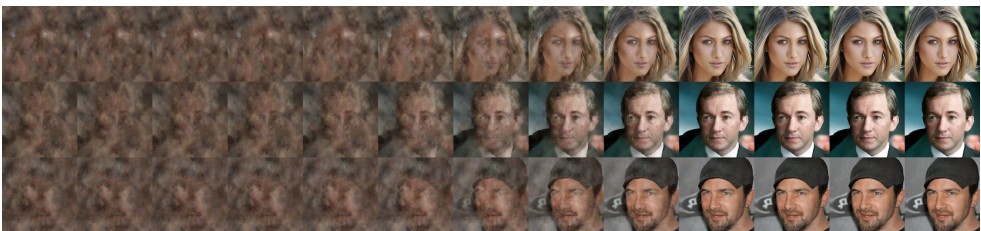

(c) Evolution of latent variables under the ODE

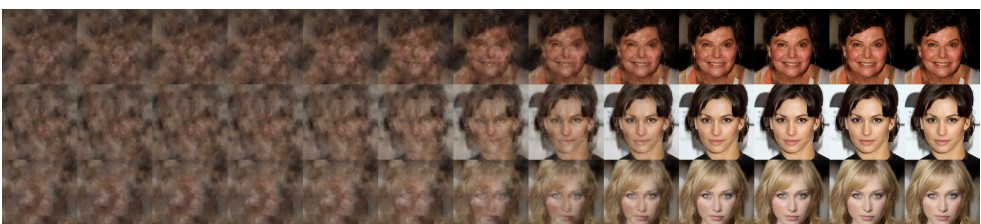

(d) Evolution of latent variables under the ODE

Figure 15: We visualize the evolution of the latent variables under both the reverse generative SDE (a-b) and also the probability flow ODE (c-d). Specifically, we feed latent variables from different stages along the generative denoising diffusion process to the decoder to map them back to image space. The 13 different images in each row correspond to the times $t = [1.0, 0.9, 0.8, 0.7, 0.6, 0.5, 0.4, 0.3, 0.2, 0.1, 0.05, 0.01, 10^{-5}]$ along the reverse denoising diffusion process. The evolution of the images is noticeably different from diffusion models that are run directly in pixel space (see, for example, Fig. 1 in [2]).