# OpenReview forum: "Score-based Generative Modeling in Latent Space"
_NeurIPS.cc/2021/Conference — NeurIPS 2021 Poster_

### Official Review · Reviewer_ipL2 · 2021-07-15

**Rating:** 8
**Confidence:** 4

**Summary:**

The paper at hand proposes end-to-end training of score-based generative models in the latent space of a variational autoencoder. The VAE is pre-trained using a normal prior which is then replaced by the score-based generative model that is then jointly trained with the VAE. The paper also introduces a novel training objective based on the cross entropy between the encoder distribution and the SGM prior. Two techniques for variance reduction of the loss function are discussed.


**Limitations And Societal Impact:**

Yes.

**Main Review:**

REASONS FOR SCORE:

- The idea of applying the score based model to the latent space of a VAE is novel, as far as I can see.
- The experimental evaluation yields promising results on 5-bit CIFAR10, CelebA and binarized MNIST and OMNIGLOT: the proposed LSGM outperforms a variety of generative models in terms of test likelihood and FID-score (although only by a very small margin). The value of the proposed model lies within the sampling time which is decreased from 44.6 min to to 3.91 minutes for a batch of 16 images, which is remarkable.
- The choice of models that are compared is, in my opinion, complete and the experimental evaluation is overall very convincing. Only [1] achieve slightly better results on CelebA, however the paper appeared on arxiv after the NeurIPS deadline. In Table 3 the authors report an FID score of 10.70 on CelebA for [2], however in Table 3 of [2] an FID of 10.2 is reported (again this last version was uploaded after the deadline).
- The structure of the paper is reasonable. Although the paper is very technical, the authors succeeded at clearly explaining their approach. A short background section introduces the relevant concepts for readers that are not familiar with score-based generative models. During reading the paper, I spotted no typos.
- The authors justify their choices by an ablation study where they analyze the effect of SDEs, training objectives, weighting mechanisms, and variance reduction techniques. As a small point of criticism: I had a hard time understanding Table 6. The authors should spend some time refactoring the table and its very long caption.

CONCLUSION:

Overall, I would reccomend to accept this submission into NeurIPS 2021. The paper is well-written and addresses the efficiency issues of score-based generative models. The experimental results on the benchmark datasets are promissing.

MINOR REMARKS:

- Paragraph captions not always in title-case (for example: "Implementation details")


REFERENCES:

[1] Kim, Dongjun, et al. "Score Matching Model for Unbounded Data Score." arXiv preprint arXiv:2106.05527 (2021).

[2] Esser, Patrick, Robin Rombach, and Bjorn Ommer. "Taming transformers for high-resolution image synthesis." arXiv preprint arXiv:2012.09841 (2021)

**Time Spent Reviewing:**

3

---

> ### Author Response · Authors · 2021-08-10
> **Initial Author Response**
>
> Thank you for pointing out the recent updates on [1] and [2]. We are going to add a small discussion around the concurrent works on denoising diffusion models. As you suggested, we will update Table 6 in our final camera-ready version.
>
> We also would like to thank you for the thoughtful and constructive feedback. If you have any further questions, concerns, or comments, please feel free to let us know during the discussion period. **Otherwise, we would appreciate it if you considered raising your final rating.**

---

> > ### Comment · Reviewer_ipL2 · 2021-08-25
> > **Reply to rebuttal**
> >
> > Thank you for your response! I updated my score to 8.

---

### Official Review · Reviewer_Gyo1 · 2021-07-15

**Rating:** 8
**Confidence:** 4

**Summary:**

The paper proposes Latent Score-based Generative Model (LSGM) which introduces a score-based prior in the Variational Autoencoder (VAE) framework. In contrast to previous score-based models that operate in the data space, LSGM uses a score-based model in the latent space. A sample from a base distribution (standard Gaussian) is denoised using the score-based prior in the latent space and is then mapped to the data space using a decoder. The authors further discuss how the various terms in the ELBO can be computed to train the model in an end-to-end fashion without requiring the time-dependent marginal score function. Multiple variance reduction techniques and training tricks have also been proposed for the resulting objective function. Both quantitative and qualitative results demonstrate the ability of the model to generate high fidelity images.

**Ethical Concerns:**

No.

**Limitations And Societal Impact:**

Yes.

**Main Review:**

### Significance

This paper makes a significant contribution both to the fields of score-based models and VAEs with more expressive priors. Severals well-motivated training stabilization techniques have been proposed which may prove helpful for the community. The paper also improves the state of the art in terms of sample quality (FID) for VAEs.

### Clarity

The paper is well-written and organized. There are a fews places where inclusion of details in the main text would improve the clarity of the paper:

i) Line 148: The parameterization of score function using $\epsilon_\theta$.
ii) Line 263: Differences between the three models.

### Technical quality

The paper is technically sound. The empirical results are fairly comprehensive and evaluate LSGM on various datasets against state-of-the-art deep generative models. Ablation studies presented help elucidate the contribution of different components of the proposed model.

### Originality/Relation to prior work

The paper presents a novel combination of score-based modeling with VAEs: a score-based prior distribution is introduced in the VAE framework. The relation to prior-work has been discussed sufficiently.

The proposed work is also related to recent works [e.g., 1, 2, 3] that propose techniques to improve samples from deep generative models (particularly GANs). In these works, latent vectors are first diffused to a "better" point and then decoded to generate a sample. This is similar to LSGM where the latent vectors are denoised using a score-based model. A discussion of relation to these works would improve the related works section.

[1] Tanaka, Akinori. "Discriminator optimal transport." arXiv preprint arXiv:1910.06832 (2019).

[2] Ansari, Abdul Fatir, Ming Liang Ang, and Harold Soh. "Refining deep generative models via discriminator gradient flow." arXiv preprint arXiv:2012.00780 (2020).

[3] Che, Tong, et al. "Your GAN is secretly an energy-based model and you should use discriminator driven latent sampling." arXiv preprint arXiv:2003.06060 (2020).

### Additional comments/Questions

- Did the authors investigate what final value of $\alpha$ is learned by the model? Specifically, is there an "optimal" mixing coefficient?
- In the ablation study the authors mention that the mixture formulation is particularly important for the sample quality. Do the authors have any insights on why this is the case?

-------
Post Rebuttal: Thank you for the clarification on $\alpha$ and the mixture formulation. I believe that a detailed discussion of the mixture formulation will improve the manuscript.

**Time Spent Reviewing:**

8

---

> ### Author Response · Authors · 2021-08-10
> **Initial Author Response**
>
> **Additional details and reference:**
>
> Thank you for pointing out the lines in which we could improve the presentation. We will update these lines in our final version. Also, thank you for pointing out [1, 2, 3]. We will discuss the direction of improving GANs using latent space models in our section that discusses related works.
>
> **The final value of $\alpha$:**
>
> In our experiments, $\alpha$ often converges to a very small value. For example, on CelebA-HQ-256, we observe that max($\alpha$) < 0.02. When we examine our mixed score function, we observe that our estimated score function is close to a Normal score function for most of the reverse diffusion process. We are not aware of an optimal mixing coefficient but in our experiments, we observe that when a latent variable is inactive (i.e., the posterior collapse problem), $\alpha$ for that latent variable is converged to zero.
>
> **Ablation study on mixed score formulation and sample quality:**
>
> The mixed score formulation provides a great inductive bias when learning the score function. In particular, once the diffusion process has added lots of noise to the data, the marginal distribution is close to a Normal one, for which we know the score function analytically, and we only need to learn a correction. However, learning that we are close to Normal without such an inductive bias purely from samples is extremely difficult in high dimensions due to the curse of dimensionality (for Cifar10, the latent space has dimensions up to 16x16x180 for our large models). Furthermore, due to our importance sampling schemes, we tend to oversample small $t$, rather than large $t$ (see Figure 3 in the paper). Therefore, we speculate that the model is simply not able to learn an accurate score function along the full range of times $t$ from limited samples without the mixed score formulation. However, sampling high-quality images requires an accurate score function estimate for all $t$. Since this seems to be not achievable without the mixed score, sample quality suffers significantly. On the other hand, the log-likelihood of samples is highly sensitive to local image statistics and primarily determined at small $t$, which are usually oversampled in importance sampling. It is plausible that we are still able to learn a reasonable estimate of the score function for these small $t$ even without the mixed score formulation, such that log-likelihood suffers much less than sample quality as estimated by FID scores.
>
> That being said, this is an interesting question that could be analyzed further. We would like to thank you for pointing out this observation.
>
> **Final Remark:** Thank you for providing thoughtful and constructive feedback. We hope that we were able to answer your questions and concerns above. If you have any further questions, concerns or comments, please feel free to let us know during the discussion period.

---

### Official Review · Reviewer_N7bW · 2021-07-16

**Rating:** 6
**Confidence:** 4

**Summary:**

The paper proposes to learn a flexible VAE prior using score-based generative model. Contrary to the existing score-based model that builds on high-dim pixel space, the paper applies the score-based model directly on the latent space which is typically low-dimensional.  The effectiveness of the model has been verified using generative FID score, negative log-likelihood as well as various ablation studies.

**Limitations And Societal Impact:**

The paper mentions its limitation and broader impact.

**Main Review:**

Originality:

 (+/-) The model presented is relatively intuitive and seems not appeared in the recent literature. However, the idea that uses some recently well-performed diffusion/score-based model as a flexible prior of VAE is straightforward and appears less surprising.

Quality:

 (+) The authors put efforts on designing various ablation studies to explore their model and conduct quite a lot comparisons with existing baselines. The overall generative performance looks good. I also appreciate the authors provide comprehensive supplementary materials for additional proof, details and results.

 (-) Though the FID score is good, but the vae and Sgm backbone structure can be complicated, and some places need better described in the main text. I understand the powerful backbone structures are used to target SOTA performance. But from the model/methodology side, it might be better to also focus on the "basic" backbone structures on which quite a few existing generative models build (e.g., the vae backbone with a few convolution-relu layers). Does the paper try such basic backbone models in the experiments? In the tab 2& 3, Ours (VAE backbone) refers to the non-NVAE structure? What structure exactly? Also, does all the results obtained by pre-training the NVAE first OR jointly train both VAE and SGM? Would be more interesting if the latter.  I would adjust my initial scores based on the feedback of the authors on above issues.

Clarity:

(+) overall, the paper is easy to follow and presentation is clear despite a few places about training procedures as listed above.

Significance:

 (+/-) The idea is intuitive and could be easy to use/adapt for the follow-up works, but the idea itself is not so surprising, and cannot be considered transformative.



**Time Spent Reviewing:**

3

---

> ### Author Response · Authors · 2021-08-10
> **Initial Author Response**
>
> **Comment on the novelty of LSGM:**
>
> LSGM is the first work to derive a simple expression for the cross-entropy term that allows training both score-based prior and approximate posterior (encoder) at the same time in an end-to-end fashion in the VAE framework. Prior works, including Song et al. [2], assumed that the data-generating distribution is fixed. In contrast, in LSGM the distribution of the latent variables is also optimized during training which introduces new technical challenges as discussed in Sec 3.1. Additionally, LSGM is equipped with a novel mixed parameterization of the score function (Sec 3.2), and novel importance sampling distributions (Sec 3.4) that are derived analytically. We believe that these techniques will be adopted even by practitioners that train score-based generative models in the data space. Also note that our ablation studies demonstrate that these technical contributions are crucial for the strong performance that we observe in our experiments.
>
> **The choice of architectures:**
>
> *Architecture:* our goal in this paper is not to design new VAE and SGM network architectures. That is why we simply build our LSGM using NVAE [20] as the VAE backbone and NCSN++ [2] as the SGM backbone. This allows us to have a fair comparison with these recent state-of-the-art works. It also enables us to examine our method’s scalability to large generative models.
>
> *VAE backbone in Tables 2 & 3:* We rely on NVAE as our VAE backbone. However, the original NVAE paper uses an aggressively large number of hierarchical groups in the latent space. In our initial experiments, we observed that our LSGM often performs well even with a smaller number of groups. That’s why we differentiate between our VAE backbone (i.e. NVAE with a smaller number of groups) and the original NVAE models that were published by Vahdat and Kautz. Please check Appendix G and Table 7 in the Supplementary Material for the exact number of groups and additional architecture details.
>
> In Tables 2 & 3, our "VAE Backbone" refers to the VAE models that were used in this paper and that are fully trained to convergence without introducing the SGM prior. "LSGM" refers to our model including the same VAE backbone and the SGM prior, trained jointly in an end-to-end fashion.
>
> *Smaller Architecture:* Thank you for this excellent suggestion. Below, we train a small VAE model similar to Kingma et al. [a], which is similar to a simple VAE that has been widely used in the literature. In particular, the model does not have hierarchical latent variables, but only a single latent variable group with a total of 64 latent variables. Encoder and decoder consist of small ResNets with 6 residual cells in total (every two cells there is a down- or up-sampling operation, so we have 3 blocks with 2 residual cells per block). The experiments are done on dynamically binarized MNIST, a widely used benchmark for small VAEs. As we can see in the Table below, our implementation of the VAE obtains a similar test ELBO as [a]. However, our LSGM improves the ELBO by almost 4.6 nats. This simple experiment shows that we can even obtain good generative performance with our LSGM using small VAE architectures. We will add this additional experiment to the final version of the paper.
>
> | Model      | ELBO (nats) |
> | ----------- | ----------- |
> | Small VAE [a]      |  -84.08 ($\pm$ 0.10)       |
> | Small VAE + inverse autoregressive flow [a]   |  -80.80 ($\pm$ 0.07)       |
> | Our small VAE | -83.85 |
> |Our LSGM w/ small VAE |**-79.23**|
>
> [a] Kingma et al., Improved Variational Inference with Inverse Autoregressive Flow, NeurIPS 2016.
>
> **Final Remark:** We appreciate your thoughtful and detailed feedback. We are in the process of organizing our code for the public release, which hopefully will facilitate further adoption of LSGM to new problem domains by the research community. We hope that we were able to answer your questions and concerns. If you have any further questions, concerns, or comments, please feel free to let us know during the discussion period and we will be very happy to address them. **Otherwise, we would appreciate it if you considered raising your final rating.** Thank you very much.

---

> > ### Comment · Reviewer_N7bW · 2021-08-21
> > **Reply to rebuttal**
> >
> > Thanks for the clarification. The newly added ELBO results are well appreciated.
> >
> > Do authors happen to have the experimental results (using similar simple VAE structure or more simpler feedforward non-residual network structures) on Cifar10 (in terms of FID)? Any comments on that? Also, it's good to know the whole framework is jointly trained, since NVAE backbone is quite large, so any insights on the training complexity or computation efficiently?
> >
> > Thanks

---

> > > ### Author Response · Authors · 2021-08-24
> > > **RE: Reply to rebuttal**
> > >
> > > We did not experiment with this particular VAE architecture on the CIFAR10 dataset. However, in the very early stages of our experimentation, we examined a small single-latent-variable-group NVAE on CIFAR10. It is worth noting that the expressivity and computational cost of NVAE are controlled by the number of hierarchical latent variable groups and the original NVAE uses 30 groups on CIFAR10. Using this one-group NVAE backbone (an extremely small VAE), our early-stage LSGM obtained the FID score of 12.88 with likelihood loss weighting (not optimal for FID), which is by far better than previous VAEs (the original NVAE with 30 groups achieves 23.49 FID).
> > >
> > > Later experiments indicated that increasing the number of latent groups improves both FID and test log-likelihood (which was expected given previous works on hierarchical VAEs). However, as we mentioned earlier, the performance saturated with a smaller number of groups, and we did not observe further improvements with the aggressively large number of groups that were used in the original NVAE. Because of the smaller NVAE backbone, we could train LSGM end-to-end (including both NVAE backbone and SGM prior) using similar computational resources that were used for training the very large original NVAE. For example, on CelebA-HQ-256, LSGM uses 24 latent variable groups and it is trained with 4 images per GPU on 32-GB V100 GPUs. In contrast, the original NVAE used 36 groups and it was also trained with the same number of images per GPU. The actual training time per iteration is also approximately the same for both LSGM and the original NVAE on this dataset.
> > >
> > > We hope that we were able to answer your questions and concerns. If you have any further questions, please feel free to let us know and we will be very happy to address them. Otherwise, we would appreciate it if you considered raising your paper rating. We strongly believe that LSGM has a lot of potential in many different application domains and we are working hard to release our source code to facilitate research in this space.

---

### Official Review · Reviewer_2Ubw · 2021-07-16

**Rating:** 7
**Confidence:** 3

**Summary:**

The paper proposes a generative model based on the VAE framework with a sophisticated prior / inference scheme based on denoising score-matching. Authors develop extensive machinery to adapt related ideas from score-based generation in the observed space to latent space. This greatly improves sampling time while preserving and sometimes improving sample quality and/or likelihood of data. I believe the paper is an interesting addition to the existing collection of generative modelling techniques.

**Limitations And Societal Impact:**

No issues.

**Main Review:**

Novelty:

To my knowledge, the method is novel. There is a certain intersection with some concurrent NeurIPS submissions though which will can be reflected by authors when preparing the final version of the paper.

Clarity:

The paper is generally well-written but due a large amount of mathematical material, the writing is dense and there are some details that I would to be discussed more.

1) As I understand, the exceptionally good results with modelling binary images are explained by the fact that score-matching in the latent space resulted into very tight variational approximation and, hence, good gradients for training the decoder. Would it be possible to quantify this directly by measuring the KL divergence for the LSGM and the same model trained with standard amortized inference techniques? Or there are even better ways to further illustrate this?
2) I couldn't find detailed information on how the structure of the latent space. As I understand, it's a lower resolution image-like structure. If so, can authors visualize evolution of the latent variable under the SDE and what reconstructions does it produce?
3) Clearly, effectiveness of LSGMs depends on the choice of the latent space, and I would like to get, again, a more detailed discussion around this. How do the performance metrics depend on its size and structure (image-like vs flat vector)?

Quality:

I didn't read proofs for the presented theorems, but I followed equations in the main text and they made sense to me.

Significance:

I believe LSGMs are a valuable contribution because it can enable faster generation and tighter variational approximations.

I think the potential impact can be further improved if similarly better results are obtained with decoder architectures othen than NVAE which is arguably very specially structured in handling latent variables.

**Time Spent Reviewing:**

5

---

> ### Author Response · Authors · 2021-08-10
> **Initial Author Response**
>
> **Tightness of the variational bound:**
>
> Your hypothesis regarding the tightness of the variational bound is most likely correct for our models. This is because the SGM prior is so expressive that it can very accurately model the aggregate posterior distribution. Thereby, in LSGM we avoid the “holes-in-the-prior” problem that plagues basically all previous VAEs and we are able to obtain very high-quality results when sampling from the model (the “holes-in-the-prior” problem is discussed in detail, for example, in [a,b]). Ultimately, we expect that this also leads to a very small gap between the approximate posterior and true posterior. However, measuring the variational gap requires measuring the difference between the ELBO and the data log-likelihood. The latter is usually estimated using importance sampling with the approximate posterior as proposal distribution. However, in our experiments, we observed that measuring the data log-likelihood reliably is problematic as we discussed in App. F (due to the bias in the importance weighted estimation of log-likelihood when the prior log-likelihood term is based on a stochastic estimate [Hutchinson’s trace estimator]).
>
> **Structure of the latent variables:**
>
> As you correctly pointed out, LSGM’s latent space is constructed using lower resolution variables with image-like spatial structure, following NVAE. Note that these latent variables are organized into groups with hierarchical dependencies as in NVAE. In the CelebA-HQ-256 models, we additionally have varying spatial resolutions (or *scales*) for the groups. The specific number of scales and their dimensions, the number of hierarchical groups per scale, and the number of latent variables in the channel dimension per group are presented in Table 1 in the Appendix. When feeding the latent variables to the SGM prior to model their joint distribution, we simply concatenate the variables from the different groups in the channel dimensions and feed the complete high-dimensional tensor to the SGM prior. For example, for Cifar10 where all latent variables have the same resolution, this results in a tensor with 16x16x180 dimensions. For the CelebA-HQ-256 models, we only concatenate the variables from the lowest-resolution groups and only model their distribution with the SGM prior.
>
>
> **Visualization of the evolution of latent variables under the SDE:**
>
> Thank you for suggesting this interesting experiment. You can find the visualizations of the evolution of the latent variables under both the reverse generative SDE and also the probability flow ODE at this [link](https://drive.google.com/file/d/1nuLmjUA_lx2d6NkSDyAwpwlfot_lMNRD/view) (anonymized). Specifically, we fed latent variables from different stages along the generative denoising diffusion process to the decoder to map them back to image space. The 13 different images for each row correspond to the times $t=[1.0, 0.9, 0.8, 0.7, 0.6, 0.5, 0.4, 0.3, 0.2, 0.1, 0.05, 0.01, 10^{-5}]$ along the reverse denoising diffusion process. Note how the evolution of the images is noticeably different from diffusion models that are run directly in pixel space (see, for example, Fig. 1 in [2]). We will include these visualizations in our final version of the paper.
>
>
> **Image-like vs. flat latent space:**
>
> In the LSGM formulation, we do not make any assumptions about the structure of the latent space. Theoretically, LSGM can be applied to latent variables that are arranged in any arbitrary way. In particular, LSGM can be applied for both image-like 2D and flat 1D latent variables. We believe that the performance of 2D vs. 1D latent spaces highly depends on the underlying VAE structure and the inductive biases of the neural networks representing denoising diffusions.
>
> Our goal in this paper was not to design new network architectures for generative models. That is why in our experiments we focused on the latent spaces arranged in 2D because current state-of-the-art VAEs (including NVAE [20] and VDVAE [21]) only consist of such latent spaces. Additionally, latent variables arranged in a 2D image-like fashion can be easily fed to NCSN++ [2], which is designed for 2D images. This structure allows us to have a fair comparison against both NVAE [20] and original SGM [2]. We believe that LSGM’s strong performance is also partially due to the strong inductive biases that are embedded in the design of NVAE and NCSN++.
>
> **Final Remark:** Thank you for providing thoughtful and constructive feedback. We will update our final version with the concurrent NeurIPS submissions on score-based generative models as you suggested. We hope that we could answer your questions and concerns above. If you have any further questions, concerns, or comments please feel free to let us know during the discussion period.
>
> [a] Bauer and Mnih, Resampled Priors for Variational Autoencoders. AISTATS, 2019.
>
> [b] Aneja et al., NCP-VAE: Variational Autoencoders with Noise Contrastive Priors, arXiv:2010,02917, 2020.

---

### Decision · Program_Chairs · 2021-09-27

**Decision:**

Accept (Poster)

**Comment:**

The reviewers generally agreed that this was a novel and exciting contribution. The author-reviewer discussion further supported the paper's quality.